# Quantitative convergence of trained single layer neural networks to Gaussian processes

**Eloy Mosig García**
Department of Mathematics
University of Pisa
Largo Bruno Pontecorvo, 5, 56127 Pisa PI, Italia
`eloy.mosig@phd.unipi.it`

**Andrea Agazzi**
Department of Mathematics and Statistics
University of Bern
Alpeneggstrasse 22, 3012 Bern
`andrea.agazzi@unibe.ch`

**Dario Trevisan**
Department of Mathematics
University of Pisa
Largo Bruno Pontecorvo, 5, 56127 Pisa PI, Italia
`dario.trevisan@unipi.it`

## Abstract

In this paper, we study the quantitative convergence of shallow neural networks trained via gradient descent to their associated Gaussian processes in the infinite-width limit. While previous work has established qualitative convergence under broad settings, precise, finite-width estimates remain limited, particularly during training. We provide explicit upper bounds on the quadratic Wasserstein distance between the network output and its Gaussian approximation at any training time $t \geq 0$, demonstrating polynomial decay with network width. Our results quantify how architectural parameters, such as width and input dimension, influence convergence, and how training dynamics affect the approximation error.

## 1 Introduction

Deep neural networks have achieved remarkable success across a wide range of tasks in computer vision, natural language processing, and scientific computing, often surpassing traditional models by large margins LeCun et al. [2015], Goodfellow et al. [2016]. This empirical progress has sparked substantial interest in understanding the theoretical principles underlying their behavior, particularly in the overparameterized regime, where the number of parameters is larger than the one of training samples.

A major avenue of theoretical investigation in this sense focuses on studying the properties of neural networks in the infinite-width limit. For instance, when the network's parameters at initialization are sampled from a Gaussian distribution, it was shown Neal [1996], de G. Matthews et al. [2018] that the network's output converges to a Gaussian process as width tends to infinity, providing a tractable framework for theoretical analysis.

This perspective was significantly extended by the introduction of the Neural Tangent Kernel (NTK) framework Jacot et al. [2018], allowing to characterize the training dynamics of infinitely wide neural networks under gradient descent in function space. In this limit, the network evolves approximately linearly around its initialization, and training can be understood as kernel regression with a fixed kernel which depends on the the architecture only, and is evaluated on input data. This linearization dramatically simplifies the analysis of generalization and convergence, and has led to a large body of theoretical work on the expressivity and limitations of infinitely wide networks.

39th Conference on Neural Information Processing Systems (NeurIPS 2025).

However, the practical relevance of NTK-based analyses hinges on the accuracy of their approximation at finite width. While existing literature has established qualitative convergence of wide neural networks in the NTK regime to Gaussian processes at positive training time Lee et al. [2020], rigorous quantitative results — providing explicit finite-width error bounds — remain scarce. This gap limits the applicability of NTK theory to realistic settings where network width is large but finite.

Indeed, quantitative convergence guarantees are crucial to bridge theory and practice. They allow one to bound the discrepancy between the predictions of a finite-width network and its infinite-width NTK counterpart, thus enabling quantitative uncertainty quantification estimates and the safe deployment of theoretical insights to real-world architectures. Moreover, such estimates reveal how network width, depth, initialization, and training hyperparameters impact the validity of linear approximations during training. These insights are essential for developing principled training strategies and for diagnosing when the NTK regime offers a reliable approximation, or when nonlinear effects beyond NTK must be taken into account.

## 1.1 Our contributions

This paper provides rigorous quantitative estimates for the convergence of trained shallow neural networks towards their Gaussian process counterparts, measured in terms of quadratic Wasserstein distances. Specifically, we extend previously established convergence bounds at initialization obtained by Basteri and Trevisan [2024], Favaro et al. [2025] and Trevisan [2023] to arbitrary positive training times. Our results deliver explicit convergence rates that decay polynomially with network width, clearly delineating how the approximation error evolves during training.

Concretely, we demonstrate that the distance of the distribution of a shallow neural network's output trained via gradient descent to its Gaussian process approximation at any training time $t > 0$ satisfies explicit quantitative bounds dependent on network width. Our main theorem (3.4) shows that for any test point $x$, under mild assumptions on the hidden layer width $n_1$ and on the regularity of the activation function we have:

$$\mathcal{W}_2^2(f_t(x), G_t(x)) = \mathcal{O}\left(\frac{\log n_1}{n_1}\right).$$

We also address long-term training dynamics explicitly, characterizing convergence rates as training time diverges. Indeed, the above result continues to hold on timescales $t$ growing polynomially in the network width $n_1$, as discussed in Remark 3.5.

These results significantly refine prior qualitative statements, providing actionable quantitative guidelines on how network parameters and training duration determine the extent to which finite networks emulate their infinite-width limits.

## 1.2 Related work

The convergence of randomly initialized neural networks to Gaussian processes in the infinite-width limit has been a foundational result in the theory of neural networks. This phenomenon was first suggested by Neal [1996] and later formalized for deep architectures by de G. Matthews et al. [2018]. The key insight that this correspondence extends beyond initialization was introduced by Jacot et al. [2018], who demonstrated that training dynamics in the infinite-width limit are governed by the so-called Neural Tangent Kernel (NTK), a deterministic kernel that linearizes the training trajectory. This sparked significant interest in the use of kernel methods to analyze deep learning models.

Following these developments, several works studied the convergence of finite-width neural networks to their limiting Gaussian processes. In particular, Lee et al. [2020] established that gradient descent dynamics in the NTK regime converge to those of a linearized model. More recently, the works of Basteri and Trevisan [2024] and Trevisan [2023] provided quantitative convergence rates at initialization, measured in Wasserstein distance, which laid the groundwork for a more refined understanding of the finite-width behavior of neural networks. Moreover, in the also recent work by Favaro et al. [2025], additional quantitative results were obtained for total variation and convex distances. Complementary to these works, Bordino et al. [2025] used second-order Poincaré inequalities to derive QCLTs for Gaussian neural networks, obtaining a general but suboptimal convergence rate compared to the optimal $n^{-1}$.

However, these results were largely confined to the initialization regime. To this day, extensions to the full training trajectory remained limited, with few works addressing how approximation errors evolve over time or depend on architectural features such as width and depth. The present work builds on this gap by extending the quantitative convergence discussed above to trained networks, providing explicit bounds on the Wasserstein distance between the network output and the associated Gaussian process for any positive training time.

From a spectral perspective, the NTK's conditioning plays a central role in understanding convergence rates and generalization. Lower bounds on the smallest eigenvalue of the empirical NTK have been derived under various conditions. For instance, Karhadkar et al. [2024] and Bombari et al. [2022] provide sharp bounds in the context of ReLU and smooth activation functions, respectively.

Additionaly, Carvalho et al. [2025] showed that under very mild assumptions on the non-linearity and non-proportionality of the training data, the analytic NTK is not degenerate. These results are essential for establishing the stability of the gradient flow and, hence, for deriving quantitative convergence guarantees.

Our results are closely related to the work of de G. Matthews et al. [2018], who proved weak convergence of fully-connected BNNs at initialization to a Gaussian process under the metric $\rho(f, f') = \sum_{i \in \mathbb{N}} 2^{-i} \min\{1, |f(x_i) - f'(x_i)|\}$, defined on a countable input set. In our setting, the input set is finite; considering the restriction $\rho_F$, it follows that convergence in $\mathcal{W}_2$ implies convergence in $\rho_F$. de G. Matthews et al. [2018] also analyzed convergence under the maximum mean discrepancy (MMD). While MMD is not generally controlled by Wasserstein distances, connections have been established via regularized OT divergences (Feydy et al. [2019], Nietert et al. [2021]). Moreover, Vayer and Gribonval [2023] identified conditions on the RKHS kernel $k_H$ under which $MMD \lesssim \mathcal{W}_2$. Consequently, our bounds also imply MMD convergence under these conditions. The metric $\rho_F$ which offers a notion of pointwise convergence and is oblivious of the tails of the distributions, which helps stablish the results in de G. Matthews et al. [2018]. On the other hand, $\mathcal{W}_2$ captures the geometric structure and scaling of the output space. Finally, while de G. Matthews et al. [2018] address the more general setting deep networks, our analysis focuses on the shallow case, yielding new quantitative rates which improve previous ones in our setting.

A foundational stream of research has shown that, under sufficient overparameterization, gradient-based training of neural networks converges to a global minimum. Seminal results by Du et al. [2019] and Arora et al. [2019] established that for wide two-layer networks with ReLU activation, the empirical NTK remains well-conditioned, enabling convergence via kernel regression. Subsequent advances generalized these results to deep architectures in different directions, such as Allen-Zhu et al. [2019], Zou and Gu [2019], Sankararaman et al. [2020], Wu et al. [2019], Wei et al. [2019], Zou et al. [2020], which provide guarantees that hold with high probability over parameter initalization. These works reinforce that in the NTK regime, the network trajectory stays close to its linearization around initialization. Our contributions align with this body of work and further extend this literature by providing novel finite-sample quantitative bounds on the Wasserstein-2 distance between neural network outputs and their Gaussian process approximations.

## 1.3 Structure of the paper

Section 2 introduces our notation and mathematical preliminaries. In Section 3, we present our primary theoretical contributions, including our main quantitative convergence theorem. The key technical proofs and intermediate results are outlined succinctly, referring the reader to the relevant lemmas in the Supplementary Material. Numerical experiments validating our theoretical predictions appear in Section 4. Section 5 discusses implications and future research directions.

## 2 Notation

In the following, given a matrix $A \in \mathbb{R}^{p \times q}$ we will denote by $\|A\|$ its Frobenius norm and by $\|A\|_{op}$ its operator norm. $A_{i\_} \in \mathbb{R}^q$ will denote the $i$-th row of $A$ and $A_{\_j} \in \mathbb{R}^p$ will denote the $j$-th column of $A$, for $1 \le i \le p$ and $1 \le j \le q$. The symbol $\cdot$ denotes the usual matrix product. $\sigma_{\min}(A), \sigma_{\max}(A)$ are the smallest and largest singular value of $A$; and if $p = q$, $\lambda_{\min}(A)$ and $\lambda_{\max}(A)$ denote the smallest and largest eigenvalue of $A$, respectively. For any vector-valued function $f$, $f(z)_u$ denotes

the $u$-th coordinate of $f(z)$. For any Polish metric space $X$, $\mathcal{P}(X)$ will denote the space of Borel probability measures on $X$.

## 2.1 Shallow neural networks and associated Gaussian process

We consider a fully connected, shallow (i.e. single hidden layer) neural network of *width* $n_1$ and input dimension $n_0$. We assume the output dimension $n_2$ to be equal to 1 for simplicity. The output of the neural network as a function of its parameters is given by:

$$f(x;\theta) = \frac{1}{\sqrt{n_1}}\Phi\left(\frac{1}{\sqrt{n_0}}x\theta^{(0)}\right)\theta^{(1)} \in \mathbb{R},$$

where $\theta^{(0)} \in \mathbb{R}^{n_0 \times n_1}$ and $\theta^{(1)} \in \mathbb{R}^{n_1}$ denote the inner and outer (respectively) *weights* or *parameters*, $\Phi(z)$ is the *activation function*, which acts entrywise on its input, and $x \in \mathbb{R}^{n_0}$ from now on denotes a test input. Note that our model implicitly covers neural networks with biases $b^{(0)} \in \mathbb{R}^{n_1}, b^{(1)} \in \mathbb{R}$, by substituting the input $x$ with $(x,1)$, using the parameters $\tilde{\theta}^{(0)} = (\theta^{(0)}, b^{(0)}) \in \mathbb{R}^{(n_0+1)\times n_1}, \tilde{\theta}^{(1)} = (\theta^{(1)}, b^{(1)}) \in \mathbb{R}^{n_1+1}$ and using the activation function $\tilde{\Phi}(z) = (\Phi(z), 1)$. We will denote by $N = n_0 n_1 + n_1$ the total dimension of the parameters. For any ordered set of inputs $X = (x_1, \ldots, x_d) \in \mathbb{R}^{n_0 \times d}$ we will use the notation $f(X;\theta) = (f(x_1;\theta), \ldots, f(x_d;\theta)) \in \mathbb{R}^{n_0 \times d}$. In what follows, parameters $\theta_{ij}^{(0)}, \theta_j^{(1)}$, for $1 \le i \le n_0$ and $1 \le j \le n_1$, are drawn independent and identically distributed (i.i.d.) from standard Gaussian random variables at initialization.

We will denote by $h_i$ the *preactivation* of $i$-th hidden neuron, for $1 \le i \le n_1$:

$$h_i(x;\theta) = \frac{1}{\sqrt{n_0}}x(\theta^{(0)})_{\_i} \in \mathbb{R}.$$

Now we introduce the *Gaussian approximation* $G$ of the neural network $f$ as the centered Gaussian process associated to the covariance operator $\mathcal{K}$ given by:

$$\tilde{\mathcal{K}}(x, x') = \frac{1}{n_0}x^\top x',$$

$$\mathcal{K}(x, x') = \mathbb{E}_{(u,v)\sim\mathcal{N}(0,\mathcal{T}(x,x'))}[\Phi(u)\Phi(v)],$$

with

$$\mathcal{T}(x, x') = \begin{pmatrix} \tilde{\mathcal{K}}(x,x) & \tilde{\mathcal{K}}(x,x') \\ \tilde{\mathcal{K}}(x',x) & \tilde{\mathcal{K}}(x',x') \end{pmatrix}.$$

Explicit convergence rates for the Gaussian approximation at initialization can be found in Basteri and Trevisan [2024], Favaro et al. [2025].

## 2.2 Training, empirical NTK and limiting kernel

Let $\mathcal{D} = \{(x_i, y_i)\}_{i=1}^n \subset \mathbb{R}^{n_0} \times \mathbb{R}$ be a given dataset. Denote by $\mathcal{X} = (x_1, \ldots, x_n) \in \mathbb{R}^{n_0 \times n}$ the vector of training inputs, and by $y = (y_1, \ldots, y_n) \in \mathbb{R}^n$ the vector of outputs. From now on, we will consider the parameters $\theta = \theta_t$ to evolve on training time. Consider the empirical risk for the mean squared error loss:

$$\mathcal{R}_\mathcal{D}[f, \theta] = \frac{1}{2}(f(\mathcal{X};\theta) - y)^\top(f(\mathcal{X};\theta) - y).$$

Continuous time gradient descent with respect to this objective function yields the following dynamics for the parameters and the network:

$$\frac{\partial}{\partial t}\theta_t = -\nabla_\theta \mathcal{R}_\mathcal{D}[f, \theta_t] = -\nabla_\theta f(\mathcal{X};\theta_t)(f(\mathcal{X};\theta_t) - y),$$

$$\frac{\partial}{\partial t}f(\mathcal{X};\theta) = -\nabla_\theta f(\mathcal{X};\theta_t)^\top \nabla_\theta f(\mathcal{X};\theta_t)(f(\mathcal{X};\theta_t) - y), \tag{2.1}$$

$$\frac{\partial}{\partial t}f(x;\theta) = -\nabla_\theta f(x;\theta_t)^\top \nabla_\theta f(\mathcal{X};\theta_t)(f(\mathcal{X};\theta_t) - y).$$

In Lee et al. [2020] the authors showed that the dynamics in (2.1) converge to those of a linearized network, which we now introduce.

**Definition 2.1.** Given a neural network $f$ we define its associated *linearized network*:

$$f^{\text{lin}}(x; \theta_t) = f(x; \theta_0) + \nabla_\theta f(x; \theta_0)|_{\theta=\theta_0}\omega_t$$

with the change of parameters $\omega_t = \theta_t - \theta_0$. In the following, we will also consider the *linearized gradient flow*, given by

$$\frac{\partial}{\partial t}\overline{\theta}_t = -\nabla_\theta f(\mathcal{X}; \theta_0) \cdot (f^{\text{lin}}(\mathcal{X}; \overline{\theta}_t) - y).$$

The linearized network is known to approximate arbitrarily well the real training dynamics in the wide limit under some stability conditions and positive-definiteness of the NTK (see Theorem 5.1 in Bartlett et al. [2021] and Theorem 2.2 in Chizat et al. [2019]). In the Supplementary Material we prove an analogue result (Proposition B.9) adapted to our setting, In particular, our result applies to shallow networks with inner and outer weights and for the MSE loss without scaling over the number of training points, as opposed to the cited results.

An alternative formulation of this asymptotic linearzation phenomenon is provided in Jacot et al. [2018], where the neural tangent kernel (NTK) was introduced. The authors showed that under Gaussian initialization, the dynamics (2.1) are governed by this operator which we now define in our setting. Define the *empirical kernel*, or *NTK* $k \colon \mathbb{R}^{n_0} \times \mathbb{R}^{n_0} \times \mathbb{R}^N \to \mathbb{R}$ as:

$$k(x, x'; \theta) = \nabla_\theta f(x; \theta)\nabla_\theta f(x'; \theta)^\top,$$

where $\theta \in \mathbb{R}^N$ denotes the matrix of *parameters*. The empirical kernel at the hidden layer can also be defined as a function of $\mathbb{R}^{n_0} \times \mathbb{R}^{n_0} \times \mathbb{R}^N \to \mathbb{R}^{n_1 \times n_1}$:

$$\tilde{k}_{uv}(x, x'; \theta) = \nabla_{\theta^{(0)}} h(x; \theta)_u \nabla_{\theta^{(0)}} h(x'; \theta)_v^\top, \quad 1 \leq u, v \leq n_1,$$

During training, the parameter $\theta_t$ will be omitted when it is clear from the context and we will simply write $k_t(x, x') = k_t(x, x'; \theta_t)$ and $\tilde{k}_t(x, x') = \tilde{k}_t(x, x'; \theta_t)$.

The convergence of the NTK to this limiting kernel was first proven by the authors of Jacot et al. [2018]. Define the *analytical*, or *limiting kernel* $k_\infty$ as follows:

$$k_\infty(x, x') = \mathcal{K}(x, x') + \tilde{\mathcal{K}}(x, x')\mathbb{E}_{(u,v)\sim\mathcal{N}(0,\mathcal{T}(x,x'))}[\Phi'(u)\Phi'(v)],$$

with $\mathcal{T}(x, x') = \begin{pmatrix} \tilde{\mathcal{K}}(x, x) & \tilde{\mathcal{K}}(x, x') \\ \tilde{\mathcal{K}}(x', x) & \tilde{\mathcal{K}}(x', x') \end{pmatrix}$. From now on, let $k_\infty = k_\infty(\mathcal{X}, \mathcal{X})$ denote the limiting kernel valued on the training set.

Note that, in the linearized regime, Equation (2.1) can be solved analytically. In effect, letting $k_{\mathcal{X}\mathcal{X}} = \nabla_\theta f(\mathcal{X}; \theta_0)\nabla_\theta f_0(\mathcal{X}; \theta_0)^\top$ and $k_{x\mathcal{X}} = \nabla_\theta f(x; \theta_0)\nabla_\theta f(\mathcal{X}; \theta_0)^\top$, the gradient flow equations (2.1) for $f^{\text{lin}}$ can be rewritten as:

$$\frac{\partial}{\partial t}\overline{\theta}_t = -\nabla_\theta f(\mathcal{X}; \theta_0)^\top (f^{\text{lin}}(\mathcal{X}; \overline{\theta}_t) - y),$$

$$\frac{\partial}{\partial t}f^{\text{lin}}(\mathcal{X}; \overline{\theta}_t) = -k_{\mathcal{X}\mathcal{X}}(f^{\text{lin}}(\mathcal{X}; \overline{\theta}_t) - y), \quad (2.2)$$

$$\frac{\partial}{\partial t}f^{\text{lin}}(x; \overline{\theta}_t) = -k_{x\mathcal{X}}(f^{\text{lin}}(\mathcal{X}; \overline{\theta}_t) - y).$$

The inverse of matrix $k_{\mathcal{X}\mathcal{X}}$, which is random in the initialization of the network and may not be positive definite for some $\theta$, appears in the solution of Equation (2.2). Thus we introduce the following auxiliary operator:

**Definition 2.2.** For any symmetric, invertible matrix $B \in \mathbb{R}^{n\times n}$ and for any $t > 0$, define the $n \times n$ real matrix

$$I_t(B) = (\mathbb{1}_n - e^{-Bt})B^{-1}.$$

Note that $I_t(B)$ is invertible and symmetric since the matrix exponential of $B$ is positive definite. The operator $I_t$ can be extended to general symmetric matrices in the following way: define, for each $a \in \mathbb{R}$,

$$I_t(a) = \int_0^t e^{-as}ds = \begin{cases} \frac{1-e^{-at}}{a} & \text{if } a \neq 0, \\ t & \text{if } a = 0. \end{cases}$$

Let $B \in \mathbb{R}^{n \times n}$ be symmetric, not necesarily non-degenerate. Consider the eigenvalue decomposition of $B$, $B = UDU^\top$ with $D = \mathrm{diag}(a_1, \ldots a_n)$ and $U$ orthogonal, then put $I_t(B) = U\mathrm{diag}(I_t(a_1), \ldots I_t(a_n))U^\top$.

Lemma A.1 shows some properties and well-posedness of the operator $I_t$ defined in Definition 2.2. The matrix $I_t(B)$ can be thought of as the analytic continuation of the matrix function $(\mathbb{1}_n - e^{-Bt})B^{-1}$, for any $B \in \mathbb{R}^{n \times n}$ symmetric. With this definition, the solution to (2.2) reads:

$$f^{\mathrm{lin}}(\mathcal{X}; \overline{\theta}_t) = \exp(-k_{\mathcal{X}\mathcal{X}}t)f(\mathcal{X}; \theta_0) + (\mathbb{1}_n - \exp(-k_{\mathcal{X}\mathcal{X}}t))y, \tag{2.3}$$

$$f^{\mathrm{lin}}(x; \overline{\theta}_t) = f(x; \theta_0) - k_{x\mathcal{X}}I_t(k_{\mathcal{X}\mathcal{X}})(f(x; \theta_0) - y). \tag{2.4}$$

The computation leading to this expression is contained in Supplementary Material A.

In Lee et al. [2020], the authors showed that when $f$ is linear in its parameters, its output distribution at a test point $x \in \mathbb{R}^{n_0}$ converges weakly, as $n_1$ diverges, to a Gaussian process $G_t$ with mean and covariance given by:

$$\mu_t(x) = k_\infty(x, \mathcal{X})I_t(k_\infty)y,$$
$$\Sigma_t(x, x') = \mathcal{K}(x, x') - \mathcal{K}(x, \mathcal{X})I_t(k_\infty)k_\infty(\mathcal{X}, x') - k_\infty(x, \mathcal{X})I_t(k_\infty)\mathcal{K}(\mathcal{X}, x') \tag{2.5}$$
$$+ k_\infty(x, \mathcal{X})I_t(k_\infty)\mathcal{K}(\mathcal{X}, \mathcal{X})I_t(k_\infty)k_\infty(\mathcal{X}, x'),$$

for every positive training time $t$. For the sake of completeness we included a proof for the above formula in Supplementary Material A.1.

The solution of (2.2) completely characterizes the dynamics of the Gaussian process $G_t$ for any time $t \geq 0$, as a consequence of the Central Limit Theorem. In the rest of the paper we will assume that $k_\infty(\mathcal{X}, \mathcal{X})$ is positive definite. This is a mild assumption and holds in a very general setting. Indeed, the authors of Carvalho et al. [2025] showed that when the training data is in general position in $\mathbb{R}^{n_0}$ and $\Phi$ is not a polynomial the smallest eigenvalue of $k_\infty(\mathcal{X}, \mathcal{X})$ is strictly greater than zero.

## 3 Assumptions and main result

In this section we state our assumptions and main result.

**Assumption 1.** The parameters $\theta_{ij}^{(0)}, \theta_j^{(1)}$, for $1 \leq i \leq n_0$ and $1 \leq j \leq n_1$, are drawn independent and identically distributed (i.i.d.) from standard Gaussian random variables at initialization.

**Assumption 2.** The limiting kernel $k_\infty(\mathcal{X}, \mathcal{X})$ is positive definite.

**Assumption 3.** $\Phi$ and $\Phi'$ are Lipschitz continuous and bounded.

**Assumption 4.** For some fixed $r \geq 5$ the following inequality holds:

$$\frac{4\|\mathcal{X}\|(\sqrt{5}\|\Phi\|_\infty + \|y\|)}{\sqrt{n_1 n_0}}\left(\|\Phi'\|_\infty + \mathrm{Lip}\Phi + \frac{\|\mathcal{X}\|\mathrm{Lip}\Phi'\sqrt{r \log n_1}}{\sqrt{n_0}}\right) < \lambda_{\min}^\infty,$$

*Remark* 3.1. Note that these are rather mild assumptions. Assumption 2 is mild and holds in a very general setting. Indeed, the authors of Carvalho et al. [2025] showed that when the training data is in general position in $\mathbb{R}^{n_0}$ and $\Phi$ is not a polynomial the smallest eigenvalue of $k_\infty(\mathcal{X}, \mathcal{X})$ is strictly greater than zero. Assumption 3 is standard in literature and is satisfied by most activation functions, such as the sigmoid function and other logistic activations, hyperbolic tangent, Gaussian activation or sinusoid, among others. The ReLu family is a notable exception, although we expect our result to also hold in that case. As for Assumption 4, notice that the left hand side tends to zero as $\min\{n_1, n_0\}$ grows. This implies that our hypothesis is satisfied for sufficiently large $n_0$ or $n_1$. In particular, it holds for sufficiently overparametrized networks.

*Remark* 3.2. Assumption 4 may appear somewhat artificial, so we provide an informal intuition on its use. This assumption is needed to control the fluctuations of $\|k_0 - k_\infty\|$ with the (deterministic) smallest eigenvalue of the limiting kernel. This enables the use of Proposition B.9, which provides a quenched estimation of the $L^2$ distance between $f$ and $f^{\mathrm{lin}}$, which in turn plays a central role in the proof of Theorem 3.4. In particular, the $L^2$ norm of this difference is controlled with a function of the Lipschitz constant of the Jacobian $\nabla_\theta f$ at initialization, and this Lipschitz constant is estimated with an expression in terms of $\|k_0 - k_\infty\|$ bounded by the left-hand side in Assumption 4. This is the content of Lemmas B.15 and B.16.

To measure how well the Gaussian process $G_t$ approximates the network $f$ at time $t$, we will use a well-known family of metrics between probability distributions, the Wasserstein distances:

**Definition 3.3.** Let $p \in [1, \infty]$ and let $\mu, \nu$ be two probability measures defined on a Polish space $(M, d_M)$ with finite $p$-moment. The $p$-Wasserstein distance between $\mu$ and $\nu$ is given by

$$\mathcal{W}_p(\mu, \nu) = \inf_{\gamma \in \Gamma(\mu,\nu), X \sim \mu, Y \sim \nu} \left( \mathbb{E}_\gamma \left[ d_M(X, Y)^p \right] \right)^{\frac{1}{p}},$$

where $\Gamma(\mu, \nu)$ denotes the set of joint probability measures $\gamma$ defined on $M \times M$ with marginal laws $\mu$ and $\nu$. With a slight abuse of notation, we will often write $\mathcal{W}_p(X, Y) = \mathcal{W}_p(\mu, \nu)$ for any $X \sim \mu$ and $Y \sim \nu$.

Now we can state our main theorem:

**Theorem 3.4.** *Under Assumptions 1, 2, 3 and 4, for each test point $x \in \mathbb{R}^{n_0}$ there exist positive constants $a_1$ and $a_2$ not depending on $n_0, n_1$ nor $t$ such that:*

$$\mathcal{W}_2^2(f(x; \theta_t), G_t(x)) \le r \left( \frac{a_1 \log n_1}{(\lambda_{\min}^\infty)^3 n_1 n_0} + \frac{a_2 n_0}{(\lambda_{\min}^\infty)^r n_1^{\frac{r}{4}}} (1 + t^8) \right).$$

*Here $r$ is the constant appearing in Assumption 4.*

*Remark* 3.5. Note that our result is not limited to fixed training time $t$, but holds for values of $t$ growing polynomially on $n_1$. Indeed, provided that $t$ grows at most polynomially in $n_1$, the constant $r$ can be chosen arbitrarily big making the term dependent on time negligible. In particular, as long as $t$ grows polynomially on $n_1$, a sufficiently big $r$ can be chosen so that the right-hand side in Theorem 3.4 tends to zero as $n_1$ diverges.

The term $t^8$ in the right hand side of the inequality is due to Lemma B.12 in Supplementary Material B. Lemma B.12 provides upper bounds of the entries $\theta_t^{(0)}$ and $\theta_t^{(1)}$ that account for perturbations that occur on tail events with respect to the initalization distribution (i.e. in the "bad event" $S^C$). A finer control is possible if one is interested in a result that holds $S$ on only, which has high probability, such as the ones in Bartlett et al. [2021], Chizat et al. [2019]. This finer control corresponds to Theorem B.9.

## 3.1 Sketch of proof of Theorem 3.4

The proof of our main theorem is as follows: we bound by triangle inequality

$$\mathcal{W}_2(f(x; \theta_t), G_t(x)) \le \mathcal{W}_2(f(x; \theta_t), f^{\text{lin}}(x; \overline{\theta}_t)) + \mathcal{W}_2(f^{\text{lin}}(x; \overline{\theta}_t), G_t(x)),$$

and proceed to control the two terms appearing in the right-hand side separately.

To bound the first summand, we partition $\mathbb{R}^N$ into a "good" event $S \subset \mathbb{R}^N$, in which the assumptions of Proposition B.9 hold, along some other concentration properties of the parameters, and a "bad" event $S^C$ in which they do not; so the estimation of the first summand reduces to the estimation of integrals over $S$ and $S^C$, respectively. Moreover, as $n_1$ diverges, $\mathbb{P}(S)$ converges to 1. Proposition B.9 consists on an upper bound of $\|f(x; \theta_t) - f^{\text{lin}}(x; \overline{\theta}_t)\|^2$ on this "good event"; which is a version of Theorem 5.1 in Bartlett et al. [2021] adapted to our setting. This allows to bound the first integral. In $S^C$ the strategy is to show that the measure of $S^C$ decreases faster than how the upper bounds in Theorem B.10 grow. Theorem B.10 provides estimations for $\|f(x; \theta_t) - f^{\text{lin}}(x; \overline{\theta}_t)\|^2$ that are rougher than the ones in Proposition B.9 in the sense that do not vanish in the wide limit, but hold in $S^C$. We estimate the second integral by partitioning $S^C$ into countably many discs parametrized by $\gamma \in \mathbb{N}$, and summing over $\gamma$ while exploiting concentration inequalities that hold on each disc. The result of this estimation is summarized in the following Theorem:

**Proposition 3.6.** *On the hypothesis of Theorem 3.4, there exist positive constants $a_1$ and $a_2$ not depending on $n_0, n_1$ nor $t$ such that:*

$$\mathcal{W}_2^2(f(x; \theta_t), f^{\text{lin}}(x; \overline{\theta}_t)) \le r \left( \frac{a_1 \log n_1}{(\lambda_{\min}^\infty)^3 n_1 n_0} + \frac{a_2 n_0}{(\lambda_{\min}^\infty)^r n_1^{\frac{r}{4}}} (1 + t^8) \right).$$

The dependence on time of the right-hand side of statement of the Theorem comes from Lemma B.12. In particular, in the "bad" event $S^C$ a lower bound of the smallest eigenvalue of the random matrix $k_0$ is not available, and hence by Definiton 2.2 the sharpest upper bound for $\|I_t(k_0)\|$ is $t$.

The second summand, instead, is estimated with the following result:

**Proposition 3.7.** *Let $f^{\text{lin}}$ be the linearization of $f$, and let $x \in \mathbb{R}^{n_0}$ be a test point. Then, under assumptions 1 and 3, there exist positive constants $C, \overline{C}, \overline{D}$ not depending on $n_1$ nor $t$ such that:*

$$\mathcal{W}_2^2(f^{\text{lin}}(\mathcal{X}; \overline{\theta}_t), G_t(\mathcal{X})) \leq \frac{1}{n_1} C(t+1)e^{-\lambda_{\min}^\infty t},$$

$$\mathcal{W}_2^2(f^{\text{lin}}(x; \overline{\theta}_t), G_t(x)) \leq \frac{1}{n_1}(\overline{C} + \overline{D}(t+1)e^{-\lambda_{\min}^\infty t}).$$

The first statement in Proposition 3.7 is proven by bounding $\frac{\partial}{\partial t}\|f^{\text{lin}}(\mathcal{X}; \overline{\theta}_t) - G_t(\mathcal{X})\|^2$ with Young's inequality and gradient flow equations. Next, we apply Grönwall and Hölder's inequalities to decompose the problem in some expected values of $L^2$ and $L^4$ norms of the differences between kernels and between $f^{\text{lin}}$ and $G$, both at initialization. Lastly, the main result in Basteri and Trevisan [2024] and Proposition B.4 providing $L^p$ estimations for the difference between the empirical NTK and the limiting kernel complete the proof. The proof for the second statement in Proposition 3.7 uses the bound for training points in a triangular system of differential inqualities, inherited by the solution of Equation (2.2).

Theorem 3.4 and Propositions 3.6 and 3.7 are proven in full detail in the Supplementary Material C and D.

## 4 Numerical Experiments

We conduct some numerical experiments in Figure 1 to support our theoretical results. In both experiments, $t$ is taken as the product of the learning rate and the number of iterations or epochs. For simplicity, we considered training and test inputs on the real line. The training set and test set were drawn from a uniform distribution on a fixed interval, and the labels $y$ of the training points correspond to a sine function with additive noise. The code is available at `https://github.com/emosig/quantitative_gaussian_trainedNN`.

### 4.1 Experiment 1: Gaussian approximation of $f$

The leftmost and center plots in Figure 1 represent 100 trained shallow neural networks with sigmoid activation of width $n_1 = 700$ on the leftmost plot and $n_1 = 1000$ in the central plot. The networks have been trained for $2 \cdot 10^4$ epochs with learning rates of $\frac{1}{700}$ and $\frac{7}{1000}$ (hence $t \approx 28.571$ on the leftmost plot and $t = 140$ in the central one) to fit two training points, corresponding to different random seeds on each case. Together with the networks, the plot depicts the mean (in black) of $G_t$ and a 95% confidence interval (in grey) over 200 equally spaced test points on the interval $[-10, 10]$. The networks were programmed with PyTorch 2.6.0 Paszke et al. [2019], and the Gaussian process $G_t$ was constructed using the library *neural tangents 0.6.5* Novak et al. [2020] for the kernels $\mathcal{K}$ and $k_\infty$, needed to construct $\mu_t$ and $\Sigma_t$ in (2.5). The operator $I_t(B)$ was programmed by solving the linear system of equations $BX = \mathbb{1}_n - e^{-Bt}$ with the *linalg* package of *NumPy* Harris et al. [2020].

### 4.2 Experiment 2: $\mathcal{W}_2(f(x; \theta_t), G_t(x))$ decays with $n_1$ for any $x$

In our second experiment (Figure 1, right), we compute the quadratic Wasserstein distance between the trained shallow network and $G_t$ for a variety of widths, ranging from 2 to 256. To do this we drew $10^4$ samples of $G_t$, given by the mean $\mu_t$ and covariance $\Sigma_t$ in (2.5), which were computed with the *neural tangents 0.6.5* library Novak et al. [2020]. Then, we trained indepently over a single training point, for 100 epochs and with a learning rate of 0.1 (hence $t = 10$), $10^4$ neural networks for each width and then calculated the empirical Wasserstein distance with the *Python Optimal Transport 0.9.5* library Flamary et al. [2021]. As in Experiment 1, *PyTorch* was used to construct the neural networks, and the activation chosen for $f$ and $G$ is once again the sigmoid.

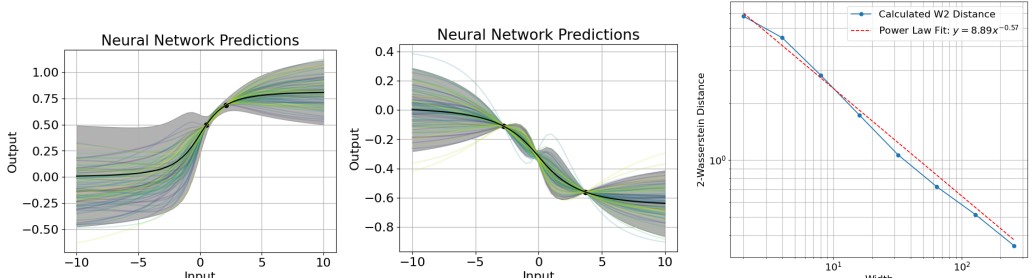

Figure 1: The Gaussian process approximates the neural networks during training (left and center images), and it converges in 2-Wasserstein space to $f_t$ (right image). On the rightmost image, the blue points represent the empirical Wasserstein distance between $f$ and $G$ for increasing widths, and the red plot is the power-law fit between the blue points.

**Discussion on the choice of $n_1$ and the number of samples**   The authors of Fournier and Guillin [2015] provide an estimation of the error between a probability measure $\mu$ and its empirical counterpart $\hat{\mu}_N = \frac{1}{N}\sum_{i=1}^{N}\delta_{X_i}$, given an i.i.d. sequence $(X_i)_{i=1}^{N}$, $X_i \sim \mu$. More precisely, Theorem 1 in Fournier and Guillin [2015], for $p = 2$ and $n_0 = 1$, provided that $\mu$ has finite third moment, reads:

$$\mathbb{E}[\mathcal{W}_2^2(\mu, \hat{\mu}_N)] \leq \frac{c_1}{\sqrt{N}},$$

for some constant $c_1$. This estimation applied to $f$ and $G$ can be combined with our main result to find a minimal ratio of samples per width needed for our numerical estimations to be noise-free. There exist constants $c_2, c_3$ not depending on $n_1, N$ nor $t$ such that:

$$\mathcal{W}_2^2(f(x;\theta_t), G_t(x)) \approx \mathcal{W}_2^2(\widehat{f(x;\theta_t)}_N, \widehat{G_t(x)}_N) + \frac{c_2}{\sqrt{N}} \leq \frac{c_3 \log n_1}{n_1}.$$

Hence our computations make sense when $N \gg \left(\frac{n_1}{\log n_1}\right)^2$. For $N = 10^4$, this means an upper bound for the widths for which our experiments make sense is, approximately, 650.

## 5   Discussion

In this paper we provided quantitative convergence rates in 2-Wasserstein distance between trained shallow neural networks with standard Gaussian initialization and an appropriate Gaussian process, for any positive training time. This was proven for Lipschitz, bounded activations with bounded derivative and for sufficiently large hidden layer width. We now address some limitations of our work and possible future research directions:

1. Our main result is not uniform in time. Although the dependence on time can be minimized at the price of including a sufficiently big multiplicative constant in the right-hand side of our inequality as discussed in Remark 3.5, a general result holding uniformly in $t > 0$, in the limit when $t$ tends to infinity exponentially on $n_1$ is not available.

   This dependence on time could be related to the transition from the NTK regime to a feature-learning regime, as suggested by the work of Huang and Yau [2020]. Their analysis, however, does not address the tails of the distributions, which in our proof correspond to the set $S^C$ and are responsible for the $t^8$ scaling. Moreover, Yang and Hu [2021] show that under standard and NTK parameterizations, wide networks cannot perform feature learning in the infinite-width limit.

   This suggests that our observed $t^8$ scaling might reflect the boundary of the NTK regime: in the "bad event" $S^C$ or for sufficiently large times, the training dynamics may drift into feature-learning, where purely kernel-based control breaks down. Our main result remains consistent with works such as Bartlett et al. [2021], Chizat et al. [2019], which hold with high probability, whereas our analysis explicitly incorporates the contribution of the event $S^C$. At present, it is unclear whether the $t^8$ scaling is sharp.

   We would like to address this problem in future work.

2. The bound in our Theorem 3.4 depends on the test point $x$. This dependence is explicitly stated on the proof of the auxiliary results Proposition 3.7 and Theorem B.10 in the Supplementary Material. Locally uniform bounds on the test point $x$ might follow from functional inequalities such as the ones found by Favaro et al. [2025] if extended to the NTK regime.

3. We conjecture that our main result remains valid even without Assumption 3, as suggested by our numerical experiments with the ReLU activation. In this work, we deliberately focused on a specialized setting with mild hypotheses to obtain a novel and technically precise result while maintaining a clear exposition. Future research will aim to relax the regularity assumptions on the activation and extend our analysis to a more general setting.

4. When Assumption 1 and 2 hold, $k_\infty(\mathcal{X}, \mathcal{X})$ is strictly positive definite and $\Phi$ is Lipschitz, the rate of convergence at initialization found in Trevisan [2023] for the squared 2-Wasserstein distance is of $n_1^{-2}$. This fact suggests that, in the proof of our main Theorem, a better estimation of $\mathcal{W}_2(f(x; \theta_t), f^{\text{lin}}(x; \overline{\theta}_t))$ can be found; either by improving Proposition 3.6 or by improving the estimation of the Lipschitz constant and the norm of the Jacobian $\nabla_\theta f(x; \theta_0)$ in the "good event" of the proof of Theorem 3.4, possibly by choosing a more restrictive "good event" that still makes the infinite sums in the proof of Theorem 3.4 converge. This last possibility calls for finer concentration inequalities for the parameters and the difference between the empirical NTK and its infinite-width limit.

5. Our results could be extended to deep, fully connected neural networks, as done for initialization in Trevisan [2023], Basteri and Trevisan [2024] and Favaro et al. [2025]. We hypothesize that Proposition 3.7 can be easily adapted to the deep setting exploiting recursive characterizations of $k_t$ and $k_\infty$ as the ones available in Jacot et al. [2018], Nguyen et al. [2021] or Lee et al. [2020]. The other half of our proof, though, relies in Proposition 3.6, which would need a new proof for the deep setting.

6. Another desirable step could be to study how well our result extends to other architectures, such as convolutional neural networks or the more modern attention-based architectures. This approach is present in Yang and Littwin [2021] but to the best of our knowledge no quantitative results in this direction are available.

# 6 Acknowledgements

E.M.G. and D.T. are members of GNAMPA group of the Istituto Nazionale di Alta Matematica (INdAM). All authors acknowledge the support of GNAMPA Project CUP E53C22001930001. E.M.G also gratefully acknowledges the hospitality and support of the Institute of Mathematical Statistics and Actuarial Sciences at University of Bern during part of this work. This research was supported by a Ph.D. fellowship funded by the Italian Ministry of University and Research (MUR) under the PNRR program "Transizioni digitali e ambientali (TDA)" (D.M. n. 118/2023, CUP I51J23000400007), within the project "Limiti di scala di dinamiche stocastiche."

D.T. acknowledges the MUR Excellence Department Project awarded to the Department of Mathematics, University of Pisa, CUP I57G22000700001, the HPC Italian National Centre for HPC, Big Data and Quantum Computing - Proposal code CN1 CN00000013, CUP I53C22000690001, the PRIN 2022 Italian grant 2022WHZ5XH - "understanding the LEarning process of QUantum Neural networks (LeQun)", CUP J53D23003890006, the project G24-202 "Variational methods for geometric and optimal matching problems" funded by Università Italo Francese. Research also partly funded by PNRR - M4C2 - Investimento 1.3, Partenariato Esteso PE00000013 - "FAIR - Future Artificial Intelligence Research" - Spoke 1 "Human-centered AI", funded by the European Commission under the NextGeneration EU programme. This research benefitted from the support of the FMJH Program Gaspard Monge for optimization and operations research and their interactions with data science.

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

# A    Gradient flow of the feature function

In this section we provide a closed analytical solution to (2.2) when $f = f^{\text{lin}}$.

Fix $x \in \mathbb{R}^{n_0}$ a test point. For the sake of clearness, we will use the following notation for each $t \geq 0$: $\overline{y}_t = f^{\text{lin}}(x; \overline{\theta}_t)$, $f_t^{\text{lin}} = f_t(\mathcal{X}; \overline{\theta}_t)$, $k_{\mathcal{X}\mathcal{X}} = k_0(\mathcal{X}, \mathcal{X})$ and $k_{x\mathcal{X}} = k_0(x, \mathcal{X})$. Note that $k_t = k_0$ for each $t$ when $f = f^{\text{lin}}$.

We begin by stating a lemma that shows that our definition of $I_t$ is consistent and commutes with the wide limit:

**Lemma A.1.** *For any real symmetric matrix $B \in \mathbb{R}^{n \times n}$ we have $I_t(B)B = BI_t(B) = \mathbb{1}_n - e^{-Bt}$; and for real symmetric matrix sequence $(B_n)_{n \in \mathbb{N}}$ with $B_n \to B$ we have $\lim_{n \to \infty} I_t(B_n) = I_t(B)$.*

The proof of this result follows from properties of the matrix exponential and is left to Supplementary Material E.

Consider the system of ODEs in (2.2) given by:

$$\frac{\partial}{\partial t} f_t^{\text{lin}} = -k_{\mathcal{X}\mathcal{X}}(f_t^{\text{lin}} - y), \tag{A.1}$$

$$\frac{\partial}{\partial t} \overline{y}_t = -k_{x\mathcal{X}}(f_t^{\text{lin}} - y). \tag{A.2}$$

Recall that, in general, the solution to the initial value problem $f'(t) = A(t)f(t)$, $f(0) = f_0$ can be written as $f(t) = \exp(\int_0^t A(s)ds)f_0$, where $f(t) \colon \mathbb{R} \to \mathbb{R}^m$, $A(t) \colon \mathbb{R} \to \mathbb{R}^{m \times m}$ are integrable functions, and $t > 0$. Therefore in our case, by letting $u_t = f_t^{\text{lin}} - y$:

$$u_t = \exp\left(-\int_0^t k_{\mathcal{X}\mathcal{X}} ds\right) u_0 = \exp(-k_{\mathcal{X}\mathcal{X}}t)u_0. \tag{A.3}$$

Moreover, by letting $v_t = \overline{y}_t - y$ and substituting the solution for $u_t$, we obtain the following expression for $v_t$:

$$\frac{\partial}{\partial t} v_t = -k_{x\mathcal{X}} \exp(-k_{\mathcal{X}\mathcal{X}}t)(f_0 - y), \quad v_0 = y_0,$$

which, by integrating and by using Definition 2.2, becomes

$$v_t - v_0 = \overline{y}_t - \overline{y}_0 \tag{A.4}$$

$$= -k_{x\mathcal{X}} \int_0^t \exp\left(-k_{\mathcal{X}\mathcal{X}}s\right) ds (f_0 - y) \tag{A.5}$$

$$= -k_{x\mathcal{X}} I_t(k_{\mathcal{X}\mathcal{X}})(f_0 - y). \tag{A.6}$$

Note that formulae (A.3) and (A.6) agree with the formulae found by the authors of Lee et al. [2020].

When $k_{\mathcal{X}\mathcal{X}}$ is not degenerate, taking the limit when $t$ tends to infinity, we get a prediction for the output of the linearized network at the end of the training:

$$f_\infty^{\text{lin}}(x) = \lim_{t \to \infty} \overline{y}_t = \overline{y}_0 - k_{x\mathcal{X}} k_{\mathcal{X}\mathcal{X}}^{-1}(f_0 - y).$$

## A.1    Proof of the characterization of $G_t$

Here we prove the formulae in (2.5).

Define $B_t = -k_{x\mathcal{X}} I_t(k_{\mathcal{X}\mathcal{X}})$ and $C_t = k_{x\mathcal{X}} I_t(k_{\mathcal{X}\mathcal{X}})$, so that $\overline{y}_t - \overline{y}_0 = B_t f_0 + C_t y$. Also recall that $k_t = k_0$ for all $t \geq 0$ since $f$ is linear on $\theta$. Note that $f_0$ and $\overline{y}_0$ are centered Gaussian processes and hence $\mathbb{E}[\overline{y}_0 + B_t f_0] = 0$. Therefore, taking the expected value of the wide limit yields:

$$\mathbb{E}[\lim_{n_1 \to \infty} \overline{y}_t] = \mathbb{E}[\lim_{n_1 \to \infty} C_t y] = k_\infty(x, \mathcal{X}) I_t(k_\infty)y, \tag{A.7}$$

where $k_\infty = k_\infty(\mathcal{X}, \mathcal{X})$. This limit is well defined thanks to Lemma A.1.

Now let $x' \in \mathbb{R}^{n_0}$ and put $y'_t = f_t(x')$ and $B'_t = -k_{x'\mathcal{X}} I_t(k_{\mathcal{X}\mathcal{X}})$ for each $t \geq 0$. Then,

$$\text{Cov}(\lim_{n_1 \to \infty} \bar{y}_t, \lim_{n_1 \to \infty} y'_t) = \mathbb{E}[\lim_{n_1 \to \infty} (\bar{y}_t - \mathbb{E}[\bar{y}_t])(y'_t - \mathbb{E}[y'_t])] \tag{A.8}$$

$$= \mathbb{E}[\lim_{n_1 \to \infty} (\bar{y}_0 + B_t f_0)(y'_0 + B'_t f_0)] \tag{A.9}$$

$$= \mathbb{E}[\lim_{n_1 \to \infty} \bar{y}_0 y'_0] + \mathbb{E}[\lim_{n_1 \to \infty} \bar{y}_0 f_0 B'_t] \tag{A.10}$$

$$+ \mathbb{E}[\lim_{n_1 \to \infty} y'_0 f_0 B_t] + \mathbb{E}[\lim_{n_1 \to \infty} f_0^2 B_t B'_t] \tag{A.11}$$

$$= \mathcal{K}(x, x') - \mathcal{K}(x, \mathcal{X}) I_t(k_\infty) k_\infty(\mathcal{X}, x') \tag{A.12}$$

$$- k_\infty(x, \mathcal{X}) I_t(k_\infty) \mathcal{K}(\mathcal{X}, x') \tag{A.13}$$

$$+ k_\infty(x, \mathcal{X}) I_t(k_\infty) \mathcal{K}(\mathcal{X}, \mathcal{X}) I_t(k_\infty) k_\infty(\mathcal{X}, x'). \tag{A.14}$$

Again, Lemma A.1 ensures the limit exists.

# B  Auxiliary and related results

In this Supplementary Material we state intermediate results in the proof of our main theorem and recall some useful results. Throughout this section we will use the following notation for each $t \geq 0$: $y_t = f(x; \theta_t)$, $f_t = f(\mathcal{X}; \theta_t)$, $k_t = k_t(\mathcal{X}, \mathcal{X})$ and $k_\infty = k_\infty(\mathcal{X}, \mathcal{X})$. All the proofs are deferred to Supplementary Material E.

In the next lemma we collect some well-known properties of the $p$-Wasserstein distance:

**Lemma B.1.** *Let $p \in [1, \infty[$ and let $X, Y$ be random variables with values in $\mathbb{R}^n$ and $Z$ be a random variable with values in $\mathbb{R}^m$. Let $\mathbb{P}_\xi$ denote the law of the random variable $\xi$ for each $\xi \in \{X, Y, Z\}$. Then*

1. *If $X, Y$ are defined on the same probability space, then $\mathcal{W}_p(X, Y) \leq \mathbb{E}[\|X - Y\|^p]^{\frac{1}{p}}$.*

2. *If $Z$ is independent from $X$ and $Y$ then $\mathcal{W}_p(X + Z, Y + Z) \leq \mathcal{W}_p(X, Y)$.*

3. *Convexity of $\mathcal{W}_p^p$: $\mathcal{W}_p^p(X, Y) \leq \int_{\mathbb{R}^m} \mathcal{W}_p^p(\mathbb{P}_{X|Z=z}, \mathbb{P}_Y) d\mathbb{P}_Z(z)$.*

4. *Let $\lambda \in \mathbb{R}^m$ be a constant vector and consider the joint random variables $\tilde{X} = (X, Z)$, $\tilde{Y} = (Y, \lambda)$. Then*

$$\mathcal{W}_p^p(\tilde{X}, \tilde{Y}) \leq \mathcal{W}_p^p(X, Y) + \mathcal{W}_p^p(Z, \lambda) = \mathcal{W}_p^p(X, Y) + \left( \int_{\mathbb{R}^m} \|z - \lambda\|^p d\mathbb{P}(z) \right)^{\frac{1}{p}},$$

5. *Let $V$ be a random variable with values in $\mathbb{R}^m$ and consider the joint random variables $\tilde{X} = (X, Z)$, $\tilde{Y} = (Y, V)$. Then, for $p \geq 2$,*

$$\mathcal{W}_p^p(\tilde{X}, \tilde{Y}) \leq 2^{\frac{p}{2}-1} \left( \mathcal{W}_{2p}^{2p}(X, Y) + \mathcal{W}_{2p}^{2p}(Z, V) \right).$$

*Moreover, for $p = 1$,*

$$\mathcal{W}_1(\tilde{X}, \tilde{Y}) \leq \mathcal{W}_1(X, Y) + \mathcal{W}_1(Z, V).$$

The following result provides explicit formulae for the components of the Jacobian of $f$ and the NTK:

**Lemma B.2** (Gradients $f$ and explicit formulae for $\tilde{k}$ and $k$). *The following hold for each $x, x' \in \mathbb{R}^{n_0}$:*

$$\nabla_{\theta^{(0)}} f(x, \theta) = \frac{1}{\sqrt{n_1 n_0}} x^\top \Phi'(\frac{1}{\sqrt{n_0}} x \theta^{(0)}) \theta^{(1)} \in \mathbb{R}^{n_0}, \tag{B.1}$$

$$\nabla_{\theta^{(1)}} f(x, \theta) = \frac{1}{\sqrt{n_1}} \Phi(\frac{1}{\sqrt{n_0}} x \theta^{(0)}) \in \mathbb{R}^{n_1}. \tag{B.2}$$

*Moreover, $\tilde{k}(x, x')$ is a diagonal $n_1 \times n_1$ matrix with*

$$\tilde{k}_{ii}(x, x') = \frac{1}{n_0} \sum_{u=1}^{n_0} x_u x'_u, \tag{B.3}$$

and $k(x, x')$ is a real function given by:

$$k(x, x') = \frac{1}{n_1 n_0} \sum_{u=1}^{n_0} x_u x'_u \sum_{v=1}^{n_1} \Phi'(h_v(x)) \Phi'(h_v(x')) (\theta_v^{(1)})^2 + \frac{1}{n_1} \sum_{v=1}^{n_1} \Phi(h_v(x)) \Phi(h_v(x')). \quad \text{(B.4)}$$

## B.1 Results at initialization

Throughout this subsection, we assume $t = 0$ and omit the subindex $t$ unless needed. Our results 3.7 and 3.4 aim to generalize Theorem 4.1 in Trevisan [2023] in different directions. For reference, we reproduce the main result from Trevisan [2023] here in a simplified version:

**Theorem B.3** (Trevisan). *Then, for each $p \in \mathbb{N}$ there exists a constant $c_p$ not depending on the network width $n_1$ such that:*

$$\mathcal{W}_p(f_0(\mathcal{X}), G_0(\mathcal{X})) \leq c_p \frac{1}{\sqrt{n_1}}. \quad \text{(B.5)}$$

*Furthermore, if $\varphi \colon \mathbb{R} \to \mathbb{R}$ is a Lipschitz function, then*

$$\mathcal{W}_p \left( \varphi((f_0(\mathcal{X}))^{\otimes 2}, \varphi(G_0(\mathcal{X}))^{\otimes 2} \right) \leq c_p \frac{(\mathrm{Lip}\varphi + \varphi(0))^2}{\sqrt{n_1}}. \quad \text{(B.6)}$$

Theorem B.3 provides a quantitative bound for the Gaussian approximation of the neural network $f_t$. This can be upgraded to a bound for the joint distribution of the empirical kernel and the output of the neural network.

From now to the end of this subsection assume $\Phi$ and $\Phi'$ are bounded and $x, x' \in \mathbb{R}^{n_0}$ are fixed. Also, we adopt the notation introduced in Supplementary Material A. We state a helpful estimation:

**Proposition B.4** ($L^p$ bound for the kernel difference). *Fix $x, x'$ in $\mathbb{R}^{n_0}$ and let $\tilde{k}_{11} = \tilde{k}_{11}(x, x')$, $k = k(x, x')$, $\mathcal{K} = \mathcal{K}(x, x')$ and $k_\infty = k_\infty(x, x')$. There exists a constant $C > 0$ independent of $n_1$ such that:*

$$\mathbb{E}[|\tilde{k}_{11} - \mathcal{K}|^p] = 0, \quad \text{(B.7)}$$

$$\mathbb{E}[|k - k_\infty|^p] \leq \frac{C}{n_1^{\frac{p}{2}}}. \quad \text{(B.8)}$$

*Remark* B.5. Note that since $k_\infty$ is deterministic, we have $\mathcal{W}_p^p(k, k_\infty) = \mathbb{E}[\|k - k_\infty\|^p]$. The constant $C$ in B.4 depends on the constants produced by applications of Theorem B.3.

The proof of Proposition B.4 goes by triangle inequality combined with Theorem B.3, and by exploiting the independence between the entries of $\theta^{(1)}$ and those of $\theta^{(0)}$. An auxiliary result to prove B.4 is the following:

**Proposition B.6** ($L^p$ bounds for the empirical kernel). *Let $\tilde{k}_{ij} = \tilde{k}_{ij}(x, x')$, for each $1 \leq i, j \leq n_1$, and $k = k(x, x')$. Then, the following inequalities hold:*

$$\mathbb{E}[|k|] = \|\Phi\|_\infty^2, \quad \text{(B.9)}$$

$$\mathbb{E}[|k|^p] \leq 2^{p-1}(2p-1)!! \|\Phi'\|_\infty^{2p} |\tilde{k}_{11}|^p + 2^{p-1} \|\Phi\|_\infty^{2p}. \quad \text{(B.10)}$$

Now we are ready to show the following result:

**Proposition B.7** (Joint distribution Basteri-Trevisan). *There exist a positive constant $C$ such that:*

$$\mathcal{W}_p \left( (k_0, f_0), (k_\infty, G_0) \right) \leq \frac{C}{\sqrt{n_1}}.$$

*with $C$ not depending on the width $n_1$.*

The proof is by using Dudley's lemma (also referred to as the gluing lemma from optimal transport) as outlined in Villani [2008]. This lemma is used to decompose the Wasserstein distance into two terms. The summand regarding the network and its Gaussian approximation is bounded with Theorem B.3, and the other summand is bounded by Proposition B.4.

Lastly, the following lemma is used in the proof of Proposition 3.7.

**Lemma B.8.** *The following inequalities hold:*

$$\mathbb{E}[\|f_0 - y\|] \leq \sqrt{n} \|\Phi\|_\infty + \|y\|, \quad \text{(B.11)}$$

$$\mathbb{E}[\|f_0 - y\|^4] \leq 32n^2 \|\Phi\|_\infty^4 + 8\|y\|^4. \quad \text{(B.12)}$$

## B.2 Approximation by linearization at training time $t > 0$

In this subsection we state two results paramount to prove Proposition 3.6, along with all their auxiliary lemmas. The following theorem resembles Theorem 5.4 in Bartlett et al. [2021], or Theorem 2.2 in Chizat et al. [2019], but we would like to remark that those result differ from ours since the first applies to neural networks in which the training is restricted to $\theta^{(0)}$ while keeping $\theta^{(1)}$ frozen, and in both of them the loss function used for training is different than the one considered in our work. The proof is similar to the one in Bartlett et al. [2021] and is carried in full detail in Supplementary Material E.

In this result we prove quenched estimations for the dynamics of the parameters, the linearization error both in the parameters and in the network valued on test points, and the convergence of the network to the labels over the training set.

**Assumption 5.** The smallest eigenvalue of $k_0$ is bounded from below by:

$$4L(\mathcal{X})\|f_0 - y\| < \lambda_{\min}(k_0),$$

where $L(\mathcal{X})$ the Lipschitz constant of $\nabla_\theta f_0$ seen as a function of $\theta$.

**Theorem B.9.** *Let* $\lambda_{\min} = \lambda_{\min}(k_0), \sigma_{\min} = \sigma_{\min}(k_0)$ *and* $\sigma_{\max} = \sigma_{\max}(k_0)$ *and let Assumption 5 hold. Then the following hold for* $t > 0$ *and for any test point* $x \in \mathbb{R}^{n_0}$:

$$\|\theta_t - \theta_0\| \leq \frac{2}{\sigma_{\min}}\|f_0 - y\|, \tag{B.13}$$

$$\|\theta_t - \overline{\theta}_t\| \leq \frac{(8 + 20\sigma_{\max}^2)L(\mathcal{X})}{\sigma_{\min}^3}\|f_0 - y\|^2, \tag{B.14}$$

$$\|f_t - y\|^2 \leq \|f_0 - y\|^2 \exp\left(-\frac{\lambda_{\min}}{2}t\right), \tag{B.15}$$

$$\|f(x;\theta_t) - f^{\text{lin}}(x;\overline{\theta}_t)\| \leq \frac{4L(x)}{\sigma_{\min}^2}\|y - f_0\|^2 \tag{B.16}$$

$$+ \frac{(8 + 20\sigma_{\max}^2)L(\mathcal{X})}{\sigma_{\min}^3}\|f_0 - y\|^2\|\nabla_\theta f_0(x)\|. \tag{B.17}$$

Proposition B.9 relies on a strong assumption involving the Lipschitz constant of the Jacobian of the network, the norm of the network at initialization and the positive-definiteness of the limiting kernel. The following rougher estimations depend on $t$, but they do not require Assumption 5 and will be key to prove our main theorem.

**Theorem B.10.** *Assume that* $\Phi$ *and* $\Phi'$ *are bounded. Let* $L(\mathcal{X})$ *be the Lipschitz constant of* $\nabla_\theta f_0$, *seen as a function of* $\theta$, *and let* $\psi(\theta_0) = \|f_0 - y\|$. *Then, for each* $t > 0$ *there exist positive constants* $A_1, \ldots, A_5, B_1, \ldots, B_9, C_1, \ldots, C_8$ *and* $C_9$ *not depending on* $n_1, n_0$ *nor* $t$ *such that,* $\mathbb{P}$-*almost everywhere:*

$$\|y_t - f_t\|^2 \leq \frac{A_0}{n_0}\|\theta_0^{(0)}\theta_0^{(1)}\|^2 + \frac{A_1 t^2}{n_0 n_1}\|\theta_0^{(0)}\|^2\psi(\theta_0)^2 + \frac{A_2\|\theta_0^{(1)}\|^2 t^4}{n_1^2 n_0}\psi(\theta_0)^4 \tag{B.18}$$

$$+ \frac{A_3 t^6}{n_1^2 n_0}\psi(\theta_0)^6 + \frac{A_4 t^2}{n_1 n_0}\|\theta_0^{(1)}\|^4\psi(\theta_0)^2 + \frac{A_5 t^4\|\theta_0^{(1)}\|^2}{n_1^2 n_0}\psi(\theta_0)^4, \tag{B.19}$$

$$\|f^{\text{lin}} - \overline{y}_t\|^2 \leq \frac{B_0}{n_1 n_0}\|\theta_0^{(0)}\theta_0^{(1)}\|^2 + \frac{B_1}{n_1^2 n_0^2}\|\theta_0^{(0)}\theta_0^{(1)}\|^2\psi(\theta_0)^4 t^4 \tag{B.20}$$

$$+ \frac{B_2}{n_1^2 n_0^2}\|\theta_0^{(0)}\|^2\|\theta_0^{(1)}\|^4\psi(\theta_0)^2 t^2 + \frac{B_3}{n_1 n_0^2}\|\theta_0^{(0)}\theta_0^{(1)}\|^2\psi(\theta_0)^2 t^2 \tag{B.21}$$

$$+ \frac{B_4}{n_1^2 n_0}\|\theta_0^{(1)}\|^2\psi(\theta_0)^4 t^4 + \frac{B_5}{n_1^2 n_0}\|\theta_0^{(1)}\|^4\psi(\theta_0)^2 t^2 \tag{B.22}$$

$$+ \frac{B_6}{n_1 n_0}\|\theta_0^{(1)}\|^2\psi(\theta_0)^2 t^2 + \frac{B_7}{n_1^2 n_0}\|\theta_0^{(0)}\|^2\psi(\theta_0)^4 t^4 \tag{B.23}$$

$$+ \frac{B_8}{n_1^2 n_0}\|\theta_0^{(0)}\|^2\|\theta_0^{(1)}\|^2\psi(\theta_0)^2 t^2 + \frac{B_9}{n_1 n_0}\|\theta_0^{(0)}\|^2\psi(\theta_0)^2 t^2, \tag{B.24}$$

$$\|f_t - f^{\text{lin}}\|^2 \le \frac{L(\mathcal{X})^2}{n_1^2} \left( \frac{C_1\psi(\theta_0)^4\|\theta_0^{(1)}\|^2 t^2}{n_1 n_0^2} + \frac{C_2\psi(\theta_0)^6 t^8}{n_1^2 n_0^2} + \frac{C_3\psi(\theta_0)^4 t^6}{n_1 n_0} \right. \tag{B.25}$$

$$+ \frac{C_4\|\theta_0^{(1)}\|^4 t^4}{n_0^2} + \frac{C_5\psi(\theta_0)^2\|\theta_0^{(1)}\|^2 t^6}{n_1 n_0^2} + \frac{C_6\|\theta_0^{(1)}\|^2 t^6}{n_0} \tag{B.26}$$

$$\left. + \frac{C_7\psi(\theta_0)^2\|\theta_0^{(1)}\|^2 t^4}{n_0} + \frac{C_8\psi(\theta_0)^4 t^6}{n_1 n_0} + C_9\psi(\theta_0)^2 t^4 \right), \tag{B.27}$$

where $\theta^{(0)}\theta^{(1)} \in \mathbb{R}^{n_0}$ denotes the usual product of the matrices $\theta^{(0)}$ and $\theta^{(1)}$.

*Remark* B.11. The dependence on time of the right-hand side of the above formulae comes from Lemma B.12. Indeed, by definition of the operator $I_t$, the sharpest upper bound for the matrices $I_t(k_t)$ and $I_t(k_0)$ when no lower bound for $\lambda_{\min}(k_t)$ and $\lambda_{\min}(k_0)$ is available is $\mathbb{1}_n t$.

The proof of the first two inequalities in this theorem is by exploiting an expression for the gradients of the network, contained in Lemma B.2; together with an integral result describing the behaviour of the parameters at time $t$ with respect to the parameters at initialization. This is the content of Lemma B.12. The third inequality uses an integral argument together with the semipositive-definiteness of $k_t$ to redirect the problem to studying $\|k_t - k_0\|$. All the constants in Theorem B.10 are multiples of the norms and Lipschitz constants of $\Phi$ and $\Phi'$, and of the norms of $x$ and $\mathcal{X}$.

Now we state Lemma B.12 along with some concentration inequalities and related auxiliary results.

**Lemma B.12** (Inequalities for $\theta_t^{(i)}$). *Fix $u$ and $v$ with $1 \le u \le n_0$ and $1 \le v \le n_1$ and put $\mathcal{X}_u = ((x_i)_u)_{i=1}^n \in \mathbb{R}^n$. Let $\lambda_{\min}$ and $\lambda_{\min}^0$ be the smallest eigenvalues of $k_t$ and $k_0$, respectively. Then the following inequalities hold:*

$$(\theta_v^{(1)})_t \le (\theta_v^{(1)})_0 + \frac{\|\Phi\|_\infty\|f_0 - y\|}{\sqrt{n_1}} I_t(\lambda_{\min}), \tag{B.28}$$

$$(\theta_{uv}^{(0)})_t \le (\theta_{uv}^{(0)})_0 + \frac{\|\Phi\|_\infty\|\Phi'\|_\infty\|f_0 - y\|^2\|\mathcal{X}_u\|}{2n_1\sqrt{n_0}} I_t(\lambda_{\min})^2 \tag{B.29}$$

$$+ \frac{\|\Phi'\|_\infty\|f_0 - y\|\|\mathcal{X}_u\|}{\sqrt{n_1 n_0}} (\theta_v^{(1)})_0 I_t(\lambda_{\min}), \tag{B.30}$$

$$(\bar{\theta}_v^{(1)})_t \le (\theta_v^{(1)})_0 + \frac{\|\Phi\|_\infty\|f_0 - y\|}{\sqrt{n_1}} I_t(\lambda_{\min}^0), \tag{B.31}$$

$$(\bar{\theta}_{uv}^{(0)})_t \le (\theta_{uv}^{(0)})_0 + \frac{\|\Phi\|_\infty\|\Phi'\|_\infty\|f_0 - y\|^2\|\mathcal{X}_u\|}{2n_1\sqrt{n_0}} I_t(\lambda_{\min}^0)^2 \tag{B.32}$$

$$+ \frac{\|\Phi'\|_\infty\|f_0 - y\|\|\mathcal{X}_u\|}{\sqrt{n_1 n_0}} (\theta_v^{(1)})_0 I_t(\lambda_{\min}^0). \tag{B.33}$$

*Remark* B.13. Recall from the definition of $I_t$ that for $t > 0$, $I_t(a) > 0$, even if $a = 0$. Moreover, $I_t(0) = t$ by definition. Hence $I_t(\lambda_{\min}^0) \le t$ and $I_t(\lambda_{\min}^0)^2 \le t^2$.

Recall the well known concentration inequality for $\chi^2$-distributed random variables (see, for example, Laurent and Massart [2000]). For each $\gamma > 0$:

$$\mathbb{P}(\|\theta_0^{(1)}\|^2 \ge 2\gamma + 2\sqrt{\gamma n_1} + n_1) \le \exp(-\gamma). \tag{B.34}$$

The following is a concentration inequality for the sup-norm of $\theta_0^{(1)}$; and as a consequence we get an estimation of the norm and Lipschitz constant of the Jacobian of $f$ at initialization.

**Lemma B.14.** *For any $\gamma > 0$:*

$$\|\theta_0^{(1)}\|_\infty \le \sqrt{r\gamma \log n_1}, \tag{B.35}$$

*with probability bigger or equal than $1 - \frac{1}{n_1^{\frac{r\gamma}{2}-1}}$.*

This concentration inequality is proven using the fact that $\|\theta_0^{(1)}\|_\infty$ is the supremum of $n_1$ Gaussian variables in absolute value.

**Lemma B.15** (Norm and Lipschitz constant of the Jacobian at $t = 0$). *Fix $r \geq 1$. Then for each $x \in \mathbb{R}^{n_0}$,*

$$\|\nabla_\theta f_0(x)\| \leq \frac{\|x\| \|\Phi'\|_\infty \sqrt{\gamma}}{\sqrt{n_0}} + \frac{\|\Phi\|_\infty}{\sqrt{n_1}}, \tag{B.36}$$

$$\mathrm{Lip}\nabla_\theta f_0(x) \leq \frac{\|x\|(\|\Phi'\|_\infty + \mathrm{Lip}\Phi)}{\sqrt{n_1 n_0}} + \frac{\|x\|^2 \mathrm{Lip}\Phi'}{\sqrt{n_1 n_0}} \sqrt{r\gamma \log n_1}. \tag{B.37}$$

*with probability greater or equal than $1 - \frac{1}{n_1^{\frac{r\gamma}{2}-1}} - \exp(-\gamma n_1)$, where $f_0(x) \colon \mathbb{R}^N \to \mathbb{R}$ is understood as a function of $\theta$.*

Now we state a concentration inequality controlling the norm of the difference between the NTK and its limit.

**Lemma B.16.** *Let $k = k_0(\mathcal{X}, \mathcal{X})$ and $k_\infty = k_\infty(\mathcal{X}, \mathcal{X})$ and let $\gamma \in \mathbb{N}$. Put $\lambda_{\min} = \lambda_{\min}(k)$ and $\lambda_{\min}^\infty = \lambda_{\min}(k_\infty)$. Then, for each $p \in \mathbb{N}$,*

$$\|k - k_\infty\| \leq \frac{\gamma \lambda_{\min}^\infty}{2}, \tag{B.38}$$

*with probability greater or equal than $1 - \left(\frac{2}{\gamma \lambda_{\min}^\infty}\right)^p \frac{C}{n_1^{\frac{p}{2}}}$, where $C$ is a positive constant not depending on $n_1$.*

*Remark* B.17. The previous lemma provides useful bounds for the smallest and largest eigenvalues of the empirical kernel at initialization, with arbitrarily high probability when the width diverges. Recall that the smallest eigenvalue of a matrix is a 1-Lipschitz function of the operator norm, which is bounded by the Frobenius norm:

$$|\lambda_{\min}(A) - \lambda_{\min}(B)| \leq \|A - B\|_{op} \leq \|A - B\|. \tag{B.39}$$

In particular, the previous Lemma implies, for $\gamma = 1$:

$$\lambda_{\min} \geq \lambda_{\min}^\infty - \|k - k_\infty\| \geq \frac{\lambda_{\min}^\infty}{2}. \tag{B.40}$$

Conversely, an upper bound for the largest eigenvalue of a matrix using the operator norm is given by:

$$\lambda_{\max}(A) \leq \lambda_{\max}(B) + \|A - B\|_{op} \leq \lambda_{\max}(B) + \|A - B\|. \tag{B.41}$$

Again, taking $\gamma = 1$ in the previous lemma yields:

$$\lambda_{\max} \leq \lambda_{\max}^\infty + \frac{\lambda_{\min}^\infty}{2}. \tag{B.42}$$

Both inequalities hold with probability greater or equal than $1 - \left(\frac{2}{\lambda_{\min}^\infty}\right)^p \frac{C}{n_1^{\frac{p}{2}}}$.

## C   Proof of Theorems 3.4 and 3.6

Here we prove Theorems 3.4 and 3.6, which share some auxiliary lemmas. Throughout this and the remaining appendices we will use the following notation for each $t \geq 0$: $y_t = f_t(x)$, $f_t = f_t(\mathcal{X})$, $\overline{y}_t = f^{\mathrm{lin}}(x; \overline{\theta}_t)$, $f_t^{\mathrm{lin}} = f^{\mathrm{lin}}(\mathcal{X}; \overline{\theta}_t)$ $k_t = k_t(\mathcal{X}, \mathcal{X})$ and $k_\infty = k_\infty(\mathcal{X}, \mathcal{X})$. The gradient $\nabla$ and the expectation $\mathbb{E}$ will always be taken with respect to the parameters $\theta$, unless otherwise indicated.

Let us begin by Proposition 3.6:

*Proof of Proposition 3.6.* Fix $p, r \in \mathbb{N}$. Consider the following subset of $\mathbb{R}^N$:

$$S = \{\theta \mid \frac{\|\theta\|^2}{n_1} \leq 5, \|\theta\|_\infty \leq \sqrt{r \log n_1}, \|k - k_\infty\| \leq \frac{\lambda_{\min}^\infty}{2}\}.$$

By the inequality (B.34) and Lemmas B.14 and B.16, the probability of $S$ is bounded from below by $1 - \exp(-n_1) - \frac{1}{\sqrt{n_1^{r-2}}} - \frac{\hat{c}}{(\lambda_{\min}^{\infty}\sqrt{n_1})^p}$, for a positive constant $\hat{c}$ not depending on $n_1, n_0$ nor $t$. Then,

$$\mathcal{W}_2^2(y_t, \overline{y}_t) \leq \inf_{\theta} \mathbb{E}_{\theta}[\|y_t - \overline{y}_t\|^2] \tag{C.1}$$

$$= \int_S \|y_t - \overline{y}_t\|^2 d\theta + \int_{S^C} \|y_t - \overline{y}_t\|^2 d\theta \tag{C.2}$$

$$\leq \sup_{\theta \in S} \|y_t - \overline{y}_t\|^2 + \left(\exp(-n_1) + \frac{1}{\sqrt{n_1^{r-2}}} + \frac{\hat{c}}{(\lambda_{\min}^{\infty}\sqrt{n_1})^p}\right) \sup_{\theta \in S^C} \|y_t - \overline{y}_t\|^2. \tag{C.3}$$

Let us begin by estimating the supremum over $S$. Note that by Lemma B.16, $\lambda_{\min} \geq \frac{\lambda_{\min}^{\infty}}{2} > 0$ in $S$. The hypothesis for Proposition B.9 are satisfied when $n_1, n_0$ are large enough. In particular, thanks to Lemma B.15, Assumption 4

$$\frac{4\|\mathcal{X}\|(\sqrt{5}\|\Phi\|_{\infty} + \|y\|)}{\sqrt{n_1 n_0}} \left(\|\Phi'\|_{\infty} + \mathrm{Lip}\Phi + \frac{\|\mathcal{X}\|\mathrm{Lip}\Phi'\sqrt{r\log n_1}}{\sqrt{n_0}}\right) < \lambda_{\min}^{\infty}$$

implies Assumption 5 for any $\theta$ in $S$. Indeed, by Lemmas B.2 and B.15:

$$4L(\mathcal{X})\|f_0 - y\| \tag{C.4}$$

$$\leq \frac{4\|\mathcal{X}\|}{\sqrt{n_1 n_0}} \left(\|\Phi'\|_{\infty} + \mathrm{Lip}\Phi + \frac{\|\mathcal{X}\|\mathrm{Lip}\Phi'\sqrt{r\log n_1}}{\sqrt{n_0}}\right)(\sqrt{5}\|\Phi\|_{\infty} + \|y\|). \tag{C.5}$$

Moreover, Lemma B.16 implies $\lambda_{\min} \geq \frac{\lambda_{\min}^{\infty}}{2}$ in $S$. These two inequalities together show that Assumption 4, which holds for sufficiently big $n_1$, is a sufficient condition for Proposition B.9 to hold.

On the other hand, by Lemmas B.16 and B.15, the following inequalities are satisfied in $S$, for $Z \in \{x, \mathcal{X}\}$:

$$L(Z) \leq \frac{\|Z\|}{\sqrt{n_1 n_0}} \left(\|\Phi'\|_{\infty} + \mathrm{Lip}\Phi + \frac{\|\mathcal{X}\|\mathrm{Lip}\Phi'\sqrt{r\log n_1}}{\sqrt{n_0}}\right), \tag{C.6}$$

$$\|\nabla_{\theta} f_0\| \leq \frac{\|\mathcal{X}\|\|\Phi'\|_{\infty}}{\sqrt{n_0}} + \frac{\|\Phi\|_{\infty}}{\sqrt{n_1}}, \tag{C.7}$$

$$\lambda_{\max} \leq \lambda_{\max}^{\infty} + \frac{\lambda_{\min}^{\infty}}{2}. \tag{C.8}$$

By substituting the estimations of $L(x), L(\mathcal{X}), \nabla f(x; \theta_0), \lambda_{\max}$ and $\lambda_{\min}$ above in Proposition B.9, for each $\theta \in S$:

$$\|y_t - \overline{y}_t\| \leq \frac{8(\sqrt{5}\|\Phi\|_{\infty} + \|y\|)^2}{\lambda_{\min}^{\infty}\sqrt{n_1 n_0}} \left(\|x\| + \frac{2\|\mathcal{X}\|}{\sqrt{\lambda_{\min}^{\infty}}}(\sqrt{2} + 15\lambda_{\max}^{\infty})\right. \tag{C.9}$$

$$\cdot \left(\|\Phi'\|_{\infty} + \mathrm{Lip}\Phi + \frac{\|\mathcal{X}\|\mathrm{Lip}\Phi'\sqrt{r\log n_1}}{\sqrt{n_0}}\right) \left.\left(\frac{\|\mathcal{X}\|\|\Phi'\|_{\infty}}{\sqrt{n_0}} + \frac{\|\Phi\|_{\infty}}{\sqrt{n_1}}\right)\right) \tag{C.10}$$

$$\leq c\sqrt{\frac{r\log n_1}{n_1 n_0 (\lambda_{\min}^{\infty})^3}}, \tag{C.11}$$

where the constant $c$ is independent of $r, t, n_0$ and $n_1$ and can be determined explicitly:

$$c = 384\|\mathcal{X}\|(\sqrt{5}\|\Phi\|_{\infty} + \|y\|)^2 \max\{\|x\|, \sqrt{2}, 15\lambda_{\max}^{\infty}, \|\Phi'\|_{\infty}, \mathrm{Lip}\Phi, \|\mathcal{X}\|\mathrm{Lip}\Phi', \|\mathcal{X}\|\|\Phi'\|_{\infty}, \|\Phi\|_{\infty}\}.$$

Hence,

$$\int_S \|y_t - \overline{y}_t\|^2 d\theta \leq \mathbb{P}(S) \sup_{\theta \in S} \|y_t - \overline{y}_t\|^2 \tag{C.12}$$

$$\leq \sup_{\theta \in S} \|y_t - \overline{y}_t\|^2 \tag{C.13}$$

$$\leq \frac{c^2 r \log n_1}{(\lambda_{\min}^{\infty})^3 n_1 n_0}. \tag{C.14}$$

Put $\alpha_1 = c^2$.

Now it only remains to estimate the second summand in (C.2). For each $\gamma \in \mathbb{N}$ let $\hat{\gamma} = 2\gamma + \sqrt{2\gamma} + 1$ and define the subsets:

$$\Omega_\gamma = \{\theta \mid \frac{\|\theta^{(1)}\|^2}{n_1} > \hat{\gamma}, \|\theta^{(1)}\|_\infty > \sqrt{r\gamma \log n_1}\}. \tag{C.15}$$

Also, define the subset

$$\Omega_* = \{\theta \mid \|k - k_\infty\| > \frac{\lambda_{\min}^\infty}{2}\}, \tag{C.16}$$

and let $\Omega = \Omega_* \setminus \bigcup_{\gamma \in \mathbb{N}} \Omega_\gamma$.

Intuitively, $\Omega_*$ and $\bigcup_{\gamma \in \mathbb{N}} \Omega_\gamma$ are the events in which the lower bound for the smallest eigenvalue of the empirical kernel and the upper bound of the Frobenius and sup-norm of the parameters at initialization, respectively, do not hold. Notice that $\mathbb{R}^N = S \sqcup \Omega \sqcup \bigcup_{\gamma \in \mathbb{N}} \Omega_\gamma$. We will use this partition of $\mathbb{R}^N$ to finish the proof.

By Lemma B.14 and by (B.34) we have $\mathbb{P}(\Omega_\gamma) \leq \exp(-\gamma n_1) + \frac{1}{n_1^{\frac{r\gamma}{2}-1}}$ and $\mathbb{P}(\Omega) \leq \mathbb{P}(\Omega_*) \leq \frac{\hat{c}}{(\lambda_{\min}^\infty \sqrt{n_1})^p}$. Moreover, the family $(\Omega_\gamma)_{\gamma \in \mathbb{N}}$ is a descending filtration of $S^C \setminus \Omega$. Let $D_\gamma = \Omega_\gamma \setminus \Omega_{\gamma+1}$, for each $\gamma \in \mathbb{N}$. This allows us to write:

$$\int_{S^C} \|y_t - \overline{y}_t\|^2 d\theta \leq \int_\Omega \|y_t - \overline{y}_t\|^2 d\theta + \sum_{\gamma \in \mathbb{N}} \int_{D_\gamma} \|y_t - \overline{y}_t\|^2 d\theta \tag{C.17}$$

$$\leq \left(\mathbb{P}(\Omega) + \sum_{\gamma \in \mathbb{N}} \mathbb{P}(D_\gamma)\right) \sup_{\theta \in D_\gamma} \|y_t - \overline{y}_t\|^2 \tag{C.18}$$

$$\leq 3\mathbb{P}(\Omega) \sup_{\theta \in \Omega} \left(\|y_t - f_t\|^2 + \|f_t - f_t^{\text{lin}}\|^2 + \|f_t^{\text{lin}} - \overline{y}_t\|^2\right) \tag{C.19}$$

$$+ 3 \sum_{\gamma \in \mathbb{N}} \mathbb{P}(D_\gamma) \sup_{\theta \in D_\gamma} \left(\|y_t - f_t\|^2 + \|f_t - f_t^{\text{lin}}\|^2 + \|f_t^{\text{lin}} - \overline{y}_t\|^2\right) \tag{C.20}$$

$$\leq 3 \sum_{\gamma \in \mathbb{N}} \exp(-\gamma n_1) \sup_{\theta \in D_\gamma} \|y_t - f_t\|^2 + 3 \sum_{\gamma \in \mathbb{N}} \exp(-\gamma n_1) \sup_{\theta \in D_\gamma} \|f_t - f_t^{\text{lin}}\|^2 \tag{C.21}$$

$$+ 3 \sum_{\gamma \in \mathbb{N}} \exp(-\gamma n_1) \sup_{\theta \in D_\gamma} \|f_t^{\text{lin}} - \overline{y}_t\|^2 + 3 \sum_{\gamma \in \mathbb{N}} \frac{1}{n_1^{\frac{r\gamma}{2}-1}} \sup_{\theta \in D_\gamma} \|y_t - f_t\|^2 \tag{C.22}$$

$$+ 3 \sum_{\gamma \in \mathbb{N}} \frac{1}{n_1^{\frac{r\gamma}{2}-1}} \sup_{\theta \in D_\gamma} \|f_t - f_t^{\text{lin}}\|^2 + 3 \sum_{\gamma \in \mathbb{N}} \frac{1}{n_1^{\frac{r\gamma}{2}-1}} \sup_{\theta \in D_\gamma} \|f_t^{\text{lin}} - \overline{y}_t\|^2 \tag{C.23}$$

$$+ 3 \frac{\hat{c}}{(\lambda_{\min}^\infty \sqrt{n_1})^p} \sup_{\theta \in \Omega} \|y_t - f_t\|^2 + 3 \frac{\hat{c}}{(\lambda_{\min}^\infty \sqrt{n_1})^p} \sup_{\theta \in \Omega} \|f_t - f_t^{\text{lin}}\|^2 \tag{C.24}$$

$$+ 3 \frac{\hat{c}}{(\lambda_{\min}^\infty \sqrt{n_1})^p} \sup_{\theta \in \Omega} \|f_t^{\text{lin}} - \overline{y}_t\|^2 \tag{C.25}$$

The 6 series in the previous expression are convergent. We compute an upper bound for each of the 6 series in (C.21) with the aid of Theorem B.10, Lemma B.14 and the inequality (B.34). Let $A = \max\{A_i\}$, $B = \max\{B_i\}$ and $C = \max\{C_i\}$ the maximums among the constants in Theorem B.10.

1. Let us estimate the first series. Observe that we can bound $\widehat{\gamma+1} \leq 7\gamma$. Moreover, since $A(\sqrt{\gamma+1} + \|y\|) \leq 2A\sqrt{\gamma}$ for $\gamma$ large enough, up to adding to the constant $A$ a multiple

of $\|y\|$ we can bound:

$$\sum_{\gamma \in \mathbb{N}} \exp(-\gamma n_1) \sup_{\theta \in D_\gamma} \|y_t - f_t\|^2 \tag{C.26}$$

$$\leq A \sum_{\gamma \in \mathbb{N}} \exp(-\gamma n_1) \left( (\widehat{\gamma+1})^2 n_1^2 n_0 + \widehat{\gamma+1}(\sqrt{\gamma+1} + \|y\|)^2 t^2 \right. \tag{C.27}$$

$$+ \frac{\widehat{\gamma+1}(\sqrt{\gamma+1} + \|y\|)^4 t^4}{n_1 n_0} + \frac{(\sqrt{\gamma+1} + \|y\|)^6 t^6}{n_1^2 n_0} \tag{C.28}$$

$$\left. + \frac{(\widehat{\gamma+1})^2(\sqrt{\gamma+1} + \|y\|)^2 n_1 t^2}{n_0} + \frac{\widehat{\gamma+1}(\sqrt{\gamma+1} + \|y\|)^4 t^4}{n_1 n_0} \right) \tag{C.29}$$

$$\leq 7A \sum_{\gamma \in \mathbb{N}} \exp(-\gamma n_1) \left( 7\gamma^3 n_1^2 n_0 + 4\gamma^2 t^2 \right. \tag{C.30}$$

$$\left. + \frac{16\gamma^3 t^4}{n_1 n_0} + \frac{64\gamma^3 t^6}{7 n_1^2 n_0} + \frac{28\gamma^3 n_1 t^2}{n_0} + \frac{16\gamma^3 t^4}{n_1 n_0} \right). \tag{C.31}$$

Since the negative exponential function decreaeses faster than any polynomial, for each $p \in \mathbb{N}$ there exists a positive constant $N_p$ such that $x^p \leq \exp(-\frac{x}{2})$ for each $x \geq N_p$. Therefore, up to a multiplicative constant not depending on $n_1, n_0, t$ nor $\gamma$:

$$\sum_{\gamma \in \mathbb{N}} \exp(-\gamma n_1) \sup_{\theta \in D_\gamma} \|y_t - f_t\|^2 \leq 7 \cdot 64 \cdot 6A n_0 (1 + t^6) \sum_{\gamma \in \mathbb{N}} \exp(-\frac{\gamma n_1}{2}) \tag{C.32}$$

$$\leq \frac{7 \cdot 64 \cdot 6A n_0 (1 + t^6) e^{-\frac{n_1}{2}}}{1 - e^{-\frac{n_1}{2}}}. \tag{C.33}$$

2. Now we estimate the second series. Using the bounds from the first series combined with Theorem B.10:

$$\sum_{\gamma \in \mathbb{N}} \exp(-\gamma n_1) \sup_{\theta \in D_\gamma} \|f_t^{\text{lin}} - \overline{y}_t\|^2 \tag{C.34}$$

$$\leq 7B n_1 \sum_{\gamma \in \mathbb{N}} \exp(-\gamma n_1) \left( 7\gamma^2 n_1 + \frac{112\gamma^4}{n_1} + \frac{196\gamma^4 n_1 t^2}{n_0} + \frac{28\gamma^3 n_1 t^2}{n_0} \right. \tag{C.35}$$

$$\left. + \frac{16\gamma^3 t^4}{n_1 n_0} + \frac{28\gamma^3 t^2}{n_0} + \frac{4\gamma^2 t^2}{n_0} + 16\gamma^3 t^4 n_0 + 28\gamma^3 t^2 + 4\gamma^2 t^2 \right) \tag{C.36}$$

$$\leq 7 \cdot 196 \cdot 9B n_1 (1 + t^4) \sum_{\gamma \in \mathbb{N}} \exp(-\frac{\gamma n_1}{2}) \tag{C.37}$$

$$= \frac{7 \cdot 196 \cdot 9B n_1 (1 + t^4) \exp(-\frac{n_1}{2})}{1 - \exp(-\frac{n_1}{2})}. \tag{C.38}$$

3. Now we estimate the third series in the same fashion as we did with the preceding series. For an upper bound of $L(\mathcal{X})$, recall Lemma B.15, which reads

$$L(\mathcal{X}) \leq \frac{d}{\sqrt{n_1 n_0}} \left( 1 + \frac{\sqrt{r(\gamma+1) \log n_1}}{\sqrt{n_0}} \right),$$

for $\theta \in D_\gamma$ and $d$ a constant not depending on $n_1, n_0, t$ nor $\gamma$. For $n_1$ large enough, we can suppose $1 + \sqrt{\frac{r(\gamma+1)\log n_1}{n_0}} \leq 2\sqrt{\frac{r\gamma \log n_1}{n_0}}$. Then,

$$\sum_{\gamma \in \mathbb{N}} \exp(-\gamma n_1) \sup_{\theta \in D_\gamma} \|f_t - f_t^{\text{lin}}\|^2 \tag{C.39}$$

$$\leq \frac{4Cdr \log n_1}{n_1^3 n_0^2} \sum_{\gamma \in \mathbb{N}} \gamma \exp(-\gamma n_1) \left( \frac{112 \gamma^3 t^2}{n_0^2} + \frac{64 \gamma^3 t^8}{n_1^2 n_0^2} + \frac{16 \gamma^2 t^6}{n_1 n_0} \right. \tag{C.40}$$

$$\left. + \frac{49 \gamma^2 n_1^2 t^4}{n_0^2} + \frac{28 \gamma^2 t^6}{n_0^2} + \frac{7 \gamma n_1 t^6}{n_0} + \frac{28 \gamma^2 n_1 t^4}{n_0} + \frac{16 \gamma^2 t^6}{n_1 n_0} + 4\gamma t^4 \right) \tag{C.41}$$

$$\leq \frac{4 \cdot 112 \cdot 9 Cdr \log n_1 (1 + t^8)}{n_1^3 n_0^2} \sum_{\gamma \in \mathbb{N}} \exp\left(-\frac{\gamma n_1}{2}\right) \tag{C.42}$$

$$= \frac{4 \cdot 112 \cdot 9 Cdr(1 + t^8) e^{-\frac{n_1}{2}}}{n_1^2 n_0^2 (1 - e^{-\frac{n_1}{2}})}. \tag{C.43}$$

4. Now we estimate the fourth series. Following the reasoning from the first series:

$$\sum_{\gamma \in \mathbb{N}} \frac{1}{n_1^{\frac{r\gamma}{2}-1}} \sup_{\theta \in D_\gamma} \|y_t - f_t\|^2 \leq 7A \sum_{\gamma \in \mathbb{N}} \frac{1}{n_1^{\frac{r\gamma}{2}-1}} \left( 7\gamma^3 n_1^2 n_0 + 4\gamma^2 t^2 + \frac{16\gamma^3 t^4}{n_1 n_0} \right. \tag{C.44}$$

$$\left. + \frac{64\gamma^3 t^6}{7 n_1^2 n_0} + \frac{28\gamma^3 n_1 t^2}{n_0} + \frac{16\gamma^3 t^4}{n_1 n_0} \right). \tag{C.45}$$

Recall that we can choose $r$ large enough so that all the summands have $n_1$ on the denominator. In particular, it is enough to choose $r \geq 5$ in this case. Moreover, by reasoning like in the proof of the first series, up to a multiplicative constant the terms of the form $\gamma^p n_1^{-\frac{r\gamma}{2}}$ are bounded from above by $n_1^{-\frac{r\gamma}{4}}$. Therefore, up to a multiplicative constant not depending on $n_1, n_0, t$ nor $\gamma$:

$$\sum_{\gamma \in \mathbb{N}} \exp(-\gamma n_1) \sup_{\theta \in D_\gamma} \|y_t - f_t\|^2 \leq 7 \cdot 64 \cdot 6 A n_0 (1 + t^6) \sum_{\gamma \in \mathbb{N}} \frac{1}{n_1^{\frac{r\gamma}{4}}} \tag{C.46}$$

$$= \frac{7 \cdot 64 \cdot 6 A n_0 (1 + t^6) n_1^{-\frac{r}{4}}}{1 - n_1^{-\frac{r}{4}}}. \tag{C.47}$$

5. Now we estimate the fifth series. Using the bounds from the first series combined with Theorem B.10, up to a multiplicative constant:

$$\sum_{\gamma \in \mathbb{N}} \frac{1}{n_1^{\frac{r\gamma}{2}-1}} \sup_{\theta \in D_\gamma} \|f_t^{\text{lin}} - \bar{y}_t\|^2 \tag{C.48}$$

$$\leq 7Bn_1 \sum_{\gamma \in \mathbb{N}} \frac{1}{n_1^{\frac{r\gamma}{2}}} \left( 7\gamma^2 n_1 + \frac{112\gamma^4 t^4}{n_1} + \frac{196\gamma^4 n_1 t^2}{n_0} + \frac{28\gamma^3 n_1 t^2}{n_0} \right. \tag{C.49}$$

$$\left. + \frac{16\gamma^3 t^4}{n_1 n_0} + \frac{28\gamma^3 t^2}{n_0} + \frac{4\gamma^2 t^2}{n_0} + 16\gamma^3 t^4 n_0 + 28\gamma^3 t^2 + 4\gamma^2 t^2 \right). \tag{C.50}$$

As we did for the fourth series, we can choose $r$ large enough so that the previous series is convergent. $r \geq 5$ is sufficient. Up to a multiplicative constant:

$$\sum_{\gamma \in \mathbb{N}} \frac{1}{n_1^{\frac{r\gamma}{2}-1}} \sup_{\theta \in D_\gamma} \|y_t - f_t\|^2 \leq 196 \cdot 7 \cdot 9 B n_0 (1 + t^4) \sum_{\gamma \in \mathbb{N}} \frac{1}{n_1^{\frac{r\gamma}{4}}} \tag{C.51}$$

$$\leq \frac{196 \cdot 7 \cdot 9 B n_0 (1 + t^4) n_1^{-\frac{r}{4}}}{1 - n_1^{-\frac{r}{4}}}. \tag{C.52}$$

6. Now we estimate the sixth and last series. Following the reasoning in the third and fourth series, up to a multiplicative constant:

$$\sum_{\gamma \in \mathbb{N}} \frac{1}{n_1^{\frac{r\gamma}{2}-1}} \sup_{\theta \in D_\gamma} \|f_t - f_t^{\text{lin}}\|^2 \tag{C.53}$$

$$\leq \frac{4Cdr \log n_1}{n_1^2 n_0^2} \sum_{\gamma \in \mathbb{N}} \frac{\gamma}{n_1^{\frac{r\gamma}{2}}} \left( \frac{112\gamma^3 t^2}{n_0^2} + \frac{64\gamma^3 t^8}{n_1^2 n_0^2} + \frac{16\gamma^2 t^7}{n_1 n_0} \right. \tag{C.54}$$

$$\left. + \frac{49\gamma^2 n_1^2 t^4}{n_0^2} + \frac{28\gamma^2 t^6}{n_0^2} + \frac{7\gamma n_1 t^6}{n_0} + \frac{28\gamma^2 n_1 t^4}{n_0} + \frac{16\gamma^2 t^6}{n_1 n_0} + 4\gamma t^4 \right). \tag{C.55}$$

In this case, any $r \geq 3$ makes the series converge.

$$\sum_{\gamma \in \mathbb{N}} \frac{1}{n_1^{\frac{r\gamma}{2}-1}} \sup_{\theta \in D_\gamma} \|f_t - f_t^{\text{lin}}\|^2 \tag{C.56}$$

$$\leq \frac{4 \cdot 112 \cdot 9 Cdr(1+t^8)}{n_1 n_0^2} \sum_{\gamma \in \mathbb{N}} \frac{\gamma}{n_1^{\frac{r\gamma}{4}}} \tag{C.57}$$

$$= \frac{4 \cdot 112 \cdot 9 Cdr(1+t^8) n_1^{-\frac{r}{4}}}{n_1 n_0^2 (1 - n_1^{-\frac{r}{4}})}. \tag{C.58}$$

Now we finish the proof by estimating the last 3 terms in (C.21). Recall that, by definition of $\Omega$, the bounds for the Frobenius and the sup-norms of the parameters at initialization used in the estimation of $\int_S \|y_t - \overline{y}_t\| d\theta$ hold also for $\theta \in \Omega$; and recall that $\mathbb{P}(\Omega) \leq \frac{\hat{c}}{(\lambda_{\min}^\infty \sqrt{n_1})^p}$. Let $R_1 = \frac{\hat{c}}{(\lambda_{\min}^\infty)^p}$. Then it suffices to choose $p$ large enough to counterattack the biggest exponent of $n_1$ in each of the bounds given by Theorem B.10. In particular, as seen while choosing $r$ when computing the six series, it is enough to take $p = r \geq 5$. Then, up to multiplicative constants,

$$\frac{R_1}{n_1^{\frac{p}{2}}} \sup_{\theta \in \Omega} \|y_t - f_t\|^2 \leq AR_1 n_0 (1+t^6) \frac{1}{\sqrt{n_1}}, \tag{C.59}$$

$$\frac{R_1}{n_1^{\frac{p}{2}}} \sup_{\theta \in \Omega} \|f_t\|^2 \leq R_1 B n_0 (1+t^4) \frac{1}{\sqrt{n_1}}, \tag{C.60}$$

$$\frac{R_1}{n_1^{\frac{p}{2}}} \sup_{\theta \in \Omega} \|y_t - f_t\|^2 \leq Cdr R_1 (1+t^8) \frac{\log n_1}{n_1 \sqrt{n_1} n_0^2} \tag{C.61}$$

$$\leq Cdr R_1 (1+t^8) \frac{1}{\sqrt{n_1} n_0^2}. \tag{C.62}$$

Grouping the estimations for the nine summands in (C.21) and taking $\alpha_2 = R_1 \max\{A, B, Cd\}$, up to a multiplicative constant,

$$\int_{S^C} \|y_t - \overline{y}_t\|^2 d\theta \leq \frac{\alpha_2 r n_0}{(\lambda_{\min}^\infty)^r n_1^{\frac{r}{4}}} (1+t^8). \tag{C.63}$$

This concludes the proof. □

Then, our main result, Theorem 3.4, is a direct application of Propositions 3.6 and 3.7:

*Proof of Theorem 3.4.* By triangle inequality and the elementary inequality $(a+b)^2 \leq 2a^2 + 2b^2$ for $a, b \geq 0$ decompose:

$$\mathcal{W}_2^2(y_t, G_t) \leq 2\mathcal{W}_2^2(y_t, \overline{y}_t) + 2\mathcal{W}_2^2(\overline{y}_t, G_t). \tag{C.64}$$

Then the thesis follows estimating the first summand with Proposition 3.6 and the second one with Proposition 3.7. For large enough $n_1$ we can take the constants $a_1$ and $a_2$ to be a multiple of $\alpha_1$ and $\alpha_2$ in Proposition 3.6; since the right-hand side in the statement in that result decreases as $\frac{\log n_1}{n_1} + \frac{1}{n_1^{\frac{r}{4}}}$, which is strictly slower than the right-hand side of the statement in Proposition 3.7 for any $n_1 \geq 2$. □

# D   Proof of Proposition 3.7

*Proof of Proposition 3.7.* Fix $x \in \mathbb{R}^{n_0}$ a test point. Let $G_t^x = G_t(x)$ and $G_t^{\mathcal{X}} = G_t(\mathcal{X})$.

First we show the result for the training set. With the aid of Equation (2.2), we derive a closed ODE for $|f_t - G_t|^2$. By Cauchy-Schwarz's inquality and Young's inequality

$$\frac{1}{2}\frac{\partial}{\partial t}\|f_t - G_t^{\mathcal{X}}\|^2 = \langle k_{\mathcal{XX}}(y - f_t) - k_\infty(y - G_t^{\mathcal{X}}), (f_t - G_t^{\mathcal{X}})\rangle \tag{D.1}$$

$$= (k_{\mathcal{XX}} - k_\infty)(y - f_t)(f_t - G_t^{\mathcal{X}}) - k_\infty\|f_t - G_t^{\mathcal{X}}\|^2 \tag{D.2}$$

$$\leq \|k_{\mathcal{XX}} - k_\infty\|\|y - f_t\|\|f_t - G_t^{\mathcal{X}}\| - \lambda_{\min}^\infty\|f_t - G_t^{\mathcal{X}}\|^2 \tag{D.3}$$

$$\leq \frac{1}{2\varepsilon}\|k_{\mathcal{XX}} - k_\infty\|^2\|y - f_t\|^2 + \frac{\varepsilon}{2}\|f_t - G_t^{\mathcal{X}}\|^2 - \lambda_{\min}^\infty\|f_t - G_t^{\mathcal{X}}\|^2. \tag{D.4}$$

Note that, by gradient flow equation we have $\|f_t - y\| \leq e^{-\frac{\lambda_{\min}^\infty}{2}t}\|f_0 - y\|$ for each $t \geq 0$. Choosing $\varepsilon = \lambda_{\min}^\infty$ and putting $b_t = \frac{e^{-\lambda_{\min}^\infty t}}{\lambda_{\min}^\infty}\|k_{\mathcal{XX}} - k_\infty\|^2\|y - f_0\|^2$ yields:

$$\frac{\partial}{\partial t}\|f_t - G_t^{\mathcal{X}}\|^2 \leq b_t - \lambda_{\min}^\infty\|f_t - G_t^{\mathcal{X}}\|^2. \tag{D.5}$$

Grönwall's inequality applied to (D.5) implies:

$$\|f_t - G_t\|^2 \leq e^{-\lambda_{\min}^\infty t}\left(\|f_0 - G_0\|^2 + \int_0^t e^{\lambda_{\min}^\infty s}b_s ds\right) \tag{D.6}$$

$$= e^{-\lambda_{\min}^\infty t}\left(\frac{\|k_{\mathcal{XX}} - k_\infty\|^2\|f_0 - y\|^2 t}{\lambda_{\min}^\infty} + \|f_0 - G_0\|^2\right). \tag{D.7}$$

Recall the definition of 2-Wasserstein distance. Taking the expected value of the previous equation and taking the infimum on all the couplings between $f_t$ and $G_t$ we can bound by Hölder's inequality:

$$\mathcal{W}_2^2(f_t, G_t) \leq \mathbb{E}[\|f_t - G_t\|^2] \tag{D.8}$$

$$\leq e^{-\lambda_{\min}^\infty t}\left(\frac{t}{\lambda_{\min}^\infty}\mathbb{E}[\|k_{\mathcal{XX}} - k_\infty\|^2\|f_0 - y\|^2] + \mathbb{E}[\|f_0 - G_0\|^2]\right) \tag{D.9}$$

$$\leq e^{-\lambda_{\min}^\infty t}\left(\frac{t}{\lambda_{\min}^\infty}\mathbb{E}[\|k_{\mathcal{XX}} - k_\infty\|^4]^{\frac{1}{2}}\mathbb{E}[\|f_0 - y\|^4]^{\frac{1}{2}} + \mathbb{E}[\|f_0 - G_0\|^2]\right). \tag{D.10}$$

Now we can take the infimum on the couplings between $f_0$ and $G_0$ to apply Theorem B.3, Proposition B.4 and Lemma B.8 to estimate the right-hand side of (D.10). There are positive constants $c_1$ and $c_2$ not depending on $n_1$ such that:

$$\mathcal{W}_2^2(f_t, G_t) \leq e^{-\lambda_{\min}^\infty t}\left(\frac{c_1 t}{\lambda_{\min}^\infty n_1}\sqrt{32n^2\|\Phi\|_\infty^4 + 8\|y\|^4} + \frac{c_2}{n_1}\right) \tag{D.11}$$

$$\leq \frac{Ce^{-\lambda_{\min}^\infty t}}{n_1}(t + 1), \tag{D.12}$$

with $C = \max\{\frac{2c_1}{\lambda_{\min}^\infty}\sqrt{8n^2\|\Phi\|_\infty^4 + 2\|y\|^4}, c_2\}$.

Now we show the result for an arbitrary test point $x \in \mathbb{R}^{n_0}$. Let $k_\infty^x = k_\infty(x, \mathcal{X})$. We can bound, by Equation (2.2),

$$\frac{\partial}{\partial t}(y_t - G_t^x)^2 = 2(k_{x\mathcal{X}}(y - f_t) - k_\infty^x(y - G_t^{\mathcal{X}}))(y_t - G_t^x).$$

By the formula for the derivative of the product and Cauchy-Schwarz's inequality, the preceding equation implies:

$$\frac{\partial}{\partial t}(y_t - G_t^x) \leq k_{x\mathcal{X}}(y - f_t) - k_\infty^x(y - G_t^{\mathcal{X}}) = \|k_{x\mathcal{X}} - k_\infty^x\|\|y - f_t\| - k_\infty^x(f_t - G_t^{\mathcal{X}}). \tag{D.13}$$

Put $\overline{\lambda} = \min_{1 \le i \le n} k_\infty(x, x_i)$. The last summand can be further estimated with the first result in this theorem. There exists a positive constants $C$ not depending on $n_1$ such that:

$$\frac{\partial}{\partial t}(y_t - G_t^x) \le \|k_{x\mathcal{X}} - k_\infty^x\|\|y - f_t\| + \frac{\overline{\lambda}Ce^{-\frac{\lambda_{\min}^\infty}{2}t}}{\sqrt{n_1}}\sqrt{t+1}. \tag{D.14}$$

Again, we use $\|f_t - y\| \le e^{-\frac{\lambda_{\min}^\infty}{2}t}\|f_0 - y\|$. Integrating we obtain:

$$(y_t - G_t^x) \le (y_0 - G_0^x) + \|k_{x\mathcal{X}} - k_\infty^x\|\|y - f_0\|(1 - e^{-\lambda_{\min}^\infty t}) + \frac{\overline{\lambda}C}{\sqrt{n_1}}\int_0^t \sqrt{s+1}e^{-\frac{\lambda_{\min}^\infty}{2}s}ds \tag{D.15}$$

$$\le (y_0 - G_0^x) + \|k_{x\mathcal{X}} - k_\infty^x\|\|y - f_0\|(1 - e^{-\lambda_{\min}^\infty t}) + \frac{\overline{\lambda}CD}{\sqrt{n_1}}(2 - \sqrt{t+1}e^{-\frac{\lambda_{\min}^\infty t}{2}}), \tag{D.16}$$

for a positive constant $D$, which explicit computation we now show separately. First we compute an antiderivative of $\sqrt{t+1}e^{-\frac{\lambda_{\min}^\infty}{2}t}$:

$$\int \sqrt{s+1}e^{-\frac{\lambda_{\min}^\infty}{2}s}ds \tag{D.17}$$

$$= 2e^{\frac{\lambda_{\min}^\infty}{2}}\int u^2 e^{-\frac{\lambda_{\min}^\infty}{2}u^2}du \tag{D.18}$$

$$= 2e^{\frac{\lambda_{\min}^\infty}{2}}\left(-\frac{ue^{-\frac{\lambda_{\min}^\infty}{2}u^2}}{\lambda_{\min}^\infty} + \int \frac{ue^{-\frac{\lambda_{\min}^\infty}{2}u^2}}{\lambda_{\min}^\infty}du\right) + C \tag{D.19}$$

$$= 2e^{\frac{\lambda_{\min}^\infty}{2}}\left(-\frac{ue^{-\frac{\lambda_{\min}^\infty}{2}u^2}}{\lambda_{\min}^\infty} + \frac{\sqrt{\pi}}{\sqrt{2(\lambda_{\min}^\infty)^3}}\int \frac{2e^{-v^2}}{\sqrt{\pi}}dv\right) + C \tag{D.20}$$

$$= 2e^{\frac{\lambda_{\min}^\infty}{2}}\left(-\frac{ue^{-\frac{\lambda_{\min}^\infty}{2}u^2}}{\lambda_{\min}^\infty} + \frac{\sqrt{\pi}\operatorname{erf}v}{\sqrt{2(\lambda_{\min}^\infty)^3}}\right) + C \tag{D.21}$$

$$= 2e^{\frac{\lambda_{\min}^\infty}{2}}\left(-\frac{\sqrt{s+1}e^{-\frac{\lambda_{\min}^\infty}{2}(s+1)}}{\lambda_{\min}^\infty} + \frac{\sqrt{\pi}\operatorname{erf}\left(\frac{\sqrt{\lambda_{\min}^\infty(s+1)}}{\sqrt{2}}\right)}{\sqrt{2(\lambda_{\min}^\infty)^3}}\right) + C \tag{D.22}$$

$$= -\frac{2\sqrt{s+1}e^{-\frac{\lambda_{\min}^\infty s}{2}}}{\lambda_{\min}^\infty} + \frac{\sqrt{2\pi}e^{\frac{\lambda_{\min}^\infty}{2}}\operatorname{erf}\left(\frac{\sqrt{\lambda_{\min}^\infty(s+1)}}{\sqrt{2}}\right)}{(\lambda_{\min}^\infty)^{\frac{3}{2}}} + C. \tag{D.23}$$

where in the first step we substituted $u = \sqrt{s+1}$, in the third step we substituted $v = \sqrt{\frac{\lambda_{\min}^\infty}{2}}u$ and in the fourth step we used the definition of Gauss error function $\operatorname{erf}(x)$; and $C$ denotes the integration constant. Therefore,

$$\int_0^t \sqrt{s+1}e^{-\frac{\lambda_{\min}^\infty}{2}s}ds = \frac{2 - 2\sqrt{t+1}e^{-\frac{\lambda_{\min}^\infty t}{2}}}{\lambda_{\min}^\infty} \tag{D.24}$$

$$+ \frac{\sqrt{2\pi}e^{\frac{\lambda_{\min}^\infty}{2}}\left(\operatorname{erf}\left(\frac{\sqrt{\lambda_{\min}^\infty(t+1)}}{\sqrt{2}}\right) - \operatorname{erf}\left(\frac{\sqrt{\lambda_{\min}^\infty}}{\sqrt{2}}\right)\right)}{(\lambda_{\min}^\infty)^{\frac{3}{2}}}. \tag{D.25}$$

Since $\operatorname{erf}(t) \in ]-1, 1[$, we can bound:

$$\int_0^t \sqrt{s+1}e^{-\frac{\lambda_{\min}^\infty}{2}s}ds \le \frac{2}{\lambda_{\min}^\infty}\left(1 - \sqrt{t+1}e^{-\frac{\lambda_{\min}^\infty t}{2}} + \sqrt{\frac{2\pi}{\lambda_{\min}^\infty}}e^{-\frac{\lambda_{\min}^\infty}{2}}\right) \tag{D.26}$$

$$\le D(2 - \sqrt{t+1}e^{-\frac{\lambda_{\min}^\infty t}{2}}), \tag{D.27}$$

with $D = \frac{2}{\lambda_{\min}^\infty} \max\{1, \sqrt{\frac{2\pi}{\lambda_{\min}^\infty}} e^{-\frac{\lambda_{\min}^\infty}{2}}\}$.

Turning back to (D.16), we can finish by applying the elementary inequality $(a + b + c)^2 \leq 3a^2 + 3b^2 + 3c^2$ for $a, b, c \geq 1$ and Hölder's inequality. After that, Theorem B.3, Proposition B.4 and Lemma B.8 can be applied in the same fashion as in the proof of the training case, yielding positive constants $d_1, d_2$ and $d_3$ such that:

$$\mathcal{W}_2^2(y_t, G_t^x) = \mathbb{E}[|y_t - G_t^x|^2] \tag{D.28}$$

$$\leq 3\mathbb{E}[|y_0 - G_0^x|^2] + 3\mathbb{E}[\|k_{x\mathcal{X}} - k_\infty^x\|^4]^{\frac{1}{2}} \mathbb{E}[\|y - f_0\|^4]^{\frac{1}{2}} (1 - e^{-\lambda_{\min}^\infty t})^2 \tag{D.29}$$

$$+ \frac{3(\bar{\lambda}CD)^2}{n_1}(2 - \sqrt{t+1}e^{-\frac{\lambda_{\min}^\infty t}{2}})^2 \tag{D.30}$$

$$\leq \frac{3d_1}{n_1} + \frac{6d_2}{n_1}\sqrt{8n^2\|\Phi\|_\infty^4 + 2\|y\|^4}(1 + e^{-2\lambda_{\min}^\infty t}) \tag{D.31}$$

$$+ \frac{3(\bar{\lambda}CD)^2}{n_1}(4 + (t+1)e^{-\lambda_{\min}^\infty t}). \tag{D.32}$$

By putting $\overline{C} = \max\{3d_1, 12(\bar{\lambda}CD)^2\}$ and $\overline{D} = \max\{6d_2\sqrt{8n^2\|\Phi\|_\infty^4 + 2\|y\|^4}, 3(\bar{\lambda}CD)^2\}$ we obtain the thesis.

Note that $C, \overline{C}$ and $\overline{D}$ do not depend neither on $n_1$ nor $t$. $\qquad\square$

# E   Proofs of the auxiliary results

We present here all the remaining proofs. For clearness, we will use the following notation: $X_v = h(x)_v$ and $X_v' = h(x')_v$ for each $1 \leq v \leq n_1$, for any $x, x' \in \mathbb{R}^{n_0}$.

*Proof of Lemma A.1.* It is trivial when $B$ is nonsingular. Let $\lambda_1, \ldots, \lambda_n$ be the (possible repeated) ordered eigenvalues of $B$ and suppose $\lambda_j = 0$. Then, by using the eigenvalue decomposition of $B$ and elementary properties of the matrix exponential:

$$I_t(B)B = U\begin{pmatrix} \frac{1-e^{-\lambda_1 t}}{\lambda_1} & & & & \\ & \ddots & & & \\ & & t & & \\ & & & \ddots & \\ & & & & \frac{1-e^{-\lambda_n t}}{\lambda_n} \end{pmatrix} U^\top U \begin{pmatrix} \lambda_1 & & & \\ & \ddots & & \\ & & 0 & \\ & & & \ddots & \\ & & & & \lambda_n \end{pmatrix} U^\top \tag{E.1}$$

$$= U\begin{pmatrix} 1-e^{-\lambda_1 t} & & & & \\ & \ddots & & & \\ & & 0 & & \\ & & & \ddots & \\ & & & & 1-e^{-\lambda_n t} \end{pmatrix} U^\top \tag{E.2}$$

$$= UU^\top - U\begin{pmatrix} e^{-\lambda_1 t} & & & & \\ & \ddots & & & \\ & & 0 & & \\ & & & \ddots & \\ & & & & e^{-\lambda_n t} \end{pmatrix} U^\top \tag{E.3}$$

$$= \mathbb{1}_n - e^{-Bt}. \tag{E.4}$$

The converse equality is proven in an analogous way.

As for the limit property, it is enough to show it for real numbers. Let $(a_n)_{n\in\mathbb{N}}$ be a real sequence converging to $a \in \mathbb{R}$. If $a \neq 0$, of if $a = a_n = 0$ for each $n$ the result is trivial. Thus, assume $a = 0$

and, up to taking a subsequence, that $a_n \neq 0$. Then

$$\lim_{n \to \infty} I_t(a_n) = \lim_{n \to \infty} \frac{1 - e^{-a_n t}}{a_n} = t = I_t(0).$$

$\square$

*Proof of Lemma B.1.* For the proof of the first three points we refer to any monograph on the Wasserstein distance such as Villani [2008]. In order to show (4), let $\pi^{XY}$ be an optimal transport plan between $X$ and $Y$, let $\pi^{XZ}$ be the law of $\tilde{X}$ and let $\mu^X = \mathbb{P}_X$ be the marginal law of $X$. Applying the gluing lemma of optimal transport produces a probability measure $\pi$ on $\mathbb{R}^n \times \mathbb{R}^m \times \mathbb{R}^n$ given by

$$\pi(x, z, y) = \frac{\pi^{XY}(x, y)}{\mu^X(x)} \mu^X(x) \frac{\pi^{XZ}(x, z)}{\mu^X(x)} = \frac{\pi^{XY}(x, y) \pi^{XZ}(x, z)}{\mu^X(x)}.$$

Note that integrating with respect to $y$ yields $\pi^{XZ}$ and integrating with respect to $z$ yields $\pi^{XY}$. Note also that there is a unique coupling between $Y$ and $\lambda$, which is given by $\mathbb{P}_Y \times \delta_\lambda$, hence,

$$\mathcal{W}_p^p\left(\tilde{X}, \tilde{Y}\right) \leq \int_{\mathbb{R}^{n+m+n}} \int_{\mathbb{R}^m} \|\tilde{X}(x, z) - \tilde{Y}(y, w)\|^p \delta_\lambda(dw) d\pi(dx, dz, dy) \tag{E.5}$$

$$= \int_{\mathbb{R}^{n+m+n}} \|\tilde{X}(x, z) - \tilde{Y}(y, \lambda)\|^p d\pi(dx, dz, dy) \tag{E.6}$$

$$\leq \int_{\mathbb{R}^{n+n}} \|X - Y\|^p d\pi^{XY}(dx, dy) + \int_{\mathbb{R}^m} \|Z - \lambda\|^p d\mu^Z(dz) \tag{E.7}$$

$$= \mathcal{W}_p^p(X, Y) + \mathcal{W}_p^p(Z, \lambda), \tag{E.8}$$

where $\mu^Z$ is the marginal probability on $Z$.

On the other hand, to show (5) fix $\mu \in \mathcal{P}(\mathbb{R}^n)$ and $\nu \in \mathcal{P}(\mathbb{R}^m)$ two probability measures, and denote $\mu \times \nu$ the product measure on $\mathbb{R}^{n+m}$ with the tensor product $\sigma$-algebra. Then

$$\mathcal{W}_p^p(\tilde{X}, \tilde{Y}) \leq \mathbb{E}_{\mu \times \nu}[\|\tilde{X} - \tilde{Y}\|^p] \tag{E.9}$$

$$\leq \mathbb{E}_{\mu \times \nu}[(\|X - Y\|^2 + \|Z - V\|^2)^{\frac{p}{2}}] \tag{E.10}$$

$$\leq 2^{\frac{p}{2} - 1} \left(\mathbb{E}_\mu[\|X - Y\|^{2p}] + \mathbb{E}_\nu[\|Z - V\|^{2p}]\right), \tag{E.11}$$

where in the last step we applied the elementary inequality $(a + b)^p \leq 2^{p-1}(a^p + b^p)$, for $a, b \geq 0$ and $p \geq 1$. For the 1-Wasserstein distance instead, we apply the elementary inequality $\sqrt{a + b} \leq \sqrt{a} + \sqrt{b}$:

$$\mathcal{W}_1(\tilde{X}, \tilde{Y}) \leq \mathbb{E}_{\mu \times \nu}[\|\tilde{X} - \tilde{Y}\|] \tag{E.12}$$

$$\leq \mathbb{E}_{\mu \times \nu}[(\|X - Y\|^2 + \|Z - V\|^2)^{\frac{1}{2}}] \tag{E.13}$$

$$\leq \mathbb{E}_\mu[\|X - Y\|] + \mathbb{E}_\nu[\|Z - V\|]. \tag{E.14}$$

Lastly, taking the infimum over $(\mu, \nu) \in \mathcal{P}(\mathbb{R}^n) \times \mathcal{P}(\mathbb{R}^m)$ finishes the proof. $\square$

*Proof of Lemma B.2.* The computation of the gradients is by chain rule and definition of $f$. The claims about the kernels follow from taking the dot product on the gradients we just calculated, for each $1 \leq i, j \leq n_1$:

$$\tilde{k}_{ij}(x, x') = \left(\nabla_{\theta^{(0)}} X_i\right)\left(\nabla_{\theta^{(0)}} X'_j\right) \tag{E.15}$$

$$= \frac{1}{n_0} \sum_{\substack{u=1,\ldots,n_0 \\ v=1,\ldots,n_1}} \frac{\partial}{\partial \theta_{uv}^{(0)}}(x\theta_{-i}^{(0)}) \frac{\partial}{\partial \theta_{uv}^{(0)}}(x'\theta_{-j}^{(0)}) \tag{E.16}$$

$$= \frac{1}{n_0} \sum_{\substack{u=1,\ldots,n_0 \\ v=1,\ldots,n_1}} x_u \delta_{iv} x'_u \delta_{jv} \tag{E.17}$$

$$= \frac{1}{n_0} \sum_{u=1}^{n_0} x_u x'_u \delta_{ij}. \tag{E.18}$$

Lastly,

$$k(x,x') = \left(\nabla_{\theta^{(0)}} f^{(2)}(x)\right)\left(\nabla_{\theta^{(0)}} f^{(2)}(x')\right) + \left(\nabla_{\theta^{(1)}} f^{(2)}(x)\right)\left(\nabla_{\theta^{(1)}} f^{(2)}(x')\right) \tag{E.19}$$

$$= \frac{1}{n_1} \sum_{\substack{u=1,\ldots,n_0 \\ v=1,\ldots,n_1}} \frac{\partial}{\partial \theta^{(0)}_{uv}}\left(\Phi\left(\frac{1}{\sqrt{n_0}}x\theta^{(0)}\right)\theta^{(1)}\right)\frac{\partial}{\partial \theta^{(0)}_{uv}}\left(\Phi\left(\frac{1}{\sqrt{n_0}}x'\theta^{(0)}\right)\theta^{(1)}\right) \tag{E.20}$$

$$+ \frac{1}{n_1}\sum_{z=1}^{n_1} \frac{\partial}{\partial \theta^{(1)}_z}\left(\Phi\left(\frac{1}{\sqrt{n_0}}x\theta^{(0)}\right)\theta^{(1)}\right)\frac{\partial}{\partial \theta^{(1)}_z}\left(\Phi\left(\frac{1}{\sqrt{n_0}}x'\theta^{(0)}\right)\theta^{(1)}\right) \tag{E.21}$$

$$= \frac{1}{n_1 n_0} \sum_{\substack{u=1,\ldots,n_0 \\ v=1,\ldots,n_1}} x_u x'_u \Phi'(X_v)\Phi'(X_v)(\theta^{(1)}_v)^2 + \frac{1}{n_1}\sum_{z=1}^{n_1} \Phi(X_z)\Phi(X'_z). \tag{E.22}$$

$\square$

## E.1 Proof of results at initialization

In this subsection we prove the useful Proposition B.4, which generalises the second part of Theorem B.3. This step is paramount for the proof of the rest of our results. From now to the end of this subsection assume $\Phi$ and $\Phi'$ are bounded and $x, x' \in \mathbb{R}^{n_0}$ are fixed.

*Proof of Proposition B.6.* Note that, from Lemma B.2, $k$ can be written as:

$$k = \frac{1}{n_1}\tilde{k}_{11}\sum_{v=1}^{n_1} \Phi'(X_v)\Phi'(X'_v)(\theta^{(1)}_v)^2 + \frac{1}{n_1}\sum_{v=1}^{n_1} \Phi(X_v)\Phi(X'_v),$$

and recall that $\tilde{k}$ is deterministic, for any fixed inputs $x, x'$. Hence, by boundedness of $\Phi$ and $\Phi'$, and independence of $\theta^{(1)}$ and $\theta^{(0)}$,

$$\mathbb{E}[|k|] \leq (\|\Phi'\|_\infty^2 \mathbb{E}[|\tilde{k}_{11}|] + \|\Phi\|_\infty^2) = \|\Phi\|_\infty^2, \tag{E.23}$$

$$\mathbb{E}[|k|^p] = \mathbb{E}[|\frac{1}{n_1}\tilde{k}_{11}\sum_{v=1}^{n_1} \Phi'(X_v)\Phi'(X'_v)(\theta^{(1)}_v)^2 + \frac{1}{n_1}\sum_{v=1}^{n_1} \Phi(X_v)\Phi(X'_v)|^p] \tag{E.24}$$

$$\leq 2^{p-1}\|\Phi'\|_\infty^{2p}\mathbb{E}[|\tilde{k}_{11}|^p]\mathbb{E}[(\theta^{(1)}_1)^{2p}] + 2^{p-1}\|\Phi\|_\infty^{2p} \tag{E.25}$$

$$\leq 2^{p-1}(2p-1)!!\|\Phi'\|_\infty^{2p}|\tilde{k}_{11}|^p + 2^{p-1}\|\Phi\|_\infty^{2p}, \tag{E.26}$$

where in the last inequality we used that the $2p$-th moment of a standard Gaussian variable is equal to $(2p-1)!!$.

$\square$

*Proof of Proposition B.4.* We will use the notation instroduced in Proposition B.6. The first claim is trivial since $\tilde{k}$ coincides with $\mathcal{K}$. To show the second claim, we split:

$$\mathbb{E}[|k_{11} - k_\infty|^p] = \mathbb{E}[|\frac{1}{n_1}\tilde{k}_{11}\sum_{v=1}^{n_1} \Phi'(X_v)\Phi'(X'_v)(\theta^{(1)}_v)^2 + \frac{1}{n_1}\sum_{v=1}^{n_1} \Phi(X_v)\Phi(X'_v) \tag{E.27}$$

$$- \mathcal{K}\mathbb{E}_G[\Phi'(G(x))\Phi'(G(x'))] - \mathbb{E}_G[\Phi(G(x))\Phi(G(x'))]|^p] \tag{E.28}$$

$$\leq 2^{p-1}\mathbb{E}[|\frac{1}{n_1}\tilde{k}_{11}\sum_{v=1}^{n_1} \Phi'(X_v)\Phi'(X'_v)(\theta^{(1)}_v)^2 - \mathcal{K}\mathbb{E}_G[\Phi'(G(x))\Phi'(G(x'))]|^p] \tag{E.29}$$

$$+ 2^{p-1}\mathbb{E}[|\frac{1}{n_1}\sum_{v=1}^{n_1} \Phi(X_v)\Phi(X'_v) - \mathbb{E}_G[\Phi(G(x))\Phi(G(x'))]|^p]. \tag{E.30}$$

To bound the first summand in (E.29) we split again by adding and substracting an auxiliary term:

$$\mathbb{E}[|\frac{1}{n_1}\tilde{k}_{11}\sum_{v=1}^{n_1}\Phi'(X_v)\Phi'(X_v')(\theta_v^{(1)})^2 - \mathcal{K}\mathbb{E}_G[\Phi'(G(x))\Phi'(G(x'))]|^p] \tag{E.31}$$

$$\leq 2^{p-1}\mathbb{E}[|\frac{1}{n_1}\tilde{k}_{11}\sum_{v=1}^{n_1}\Phi'(X_v)\Phi'(X_v')(\theta_v^{(1)})^2 - \frac{1}{n_1}\mathcal{K}\sum_{v=1}^{n_1}\Phi'(X_v)\Phi'(X_v')(\theta_v^{(1)})^2|^p] \tag{E.32}$$

$$+ 2^{p-1}\mathbb{E}[|\frac{1}{n_1}\mathcal{K}\sum_{v=1}^{n_1}\Phi'(X_v)\Phi'(X_v')(\theta_v^{(1)})^2 - \mathcal{K}\mathbb{E}_G[\Phi'(G(x))\Phi'(G(x'))]|^p] \tag{E.33}$$

$$= 2^{p-1}\mathbb{E}[|\frac{1}{n_1}(\tilde{k}_{11} - \mathcal{K})\sum_{v=1}^{n_1}\Phi'(X_v)\Phi'(X_v')(\theta_v^{(1)})^2|^p] \tag{E.34}$$

$$+ 2^{p-1}(\mathcal{K})^p\mathbb{E}[|\frac{1}{n_1}\sum_{v=1}^{n_1}\Phi'(X_v)\Phi'(X_v')(\theta_v^{(1)})^2 - \mathbb{E}_G[\Phi'(G(x))\Phi'(G(x'))]|^p]. \tag{E.35}$$

The first summand in (E.34) vanishes since $\tilde{k}$ equals $\mathcal{K}$. We estimate the second summand in (E.34), once again by adding and substracting an auxiliary term:

$$\mathbb{E}[|\frac{1}{n_1}\sum_{v=1}^{n_1}\Phi'(X_v)\Phi'(X_v')(\theta_v^{(1)})^2 - \mathbb{E}_G[\Phi'(G(x))\Phi'(G(x'))]|^p] \tag{E.36}$$

$$\leq 2^{p-1}\mathbb{E}[|\frac{1}{n_1}\sum_{v=1}^{n_1}\Phi'(X_v)\Phi'(X_v')((\theta_v^{(1)})^2 - 1)|^p] \tag{E.37}$$

$$+ 2^{p-1}\mathbb{E}[|\frac{1}{n_1}\sum_{v=1}^{n_1}\Phi'(X_v)\Phi'(X_v') - \mathbb{E}_G[\Phi'(G(x))\Phi'(G(x'))]|^p]. \tag{E.38}$$

The first summand in (E.37) vanishes, by boundedness of $\Phi'$ and independence of the parameters $\theta_v^{(1)}$:

$$\mathbb{E}[|\frac{1}{n_1}\sum_{v=1}^{n_1}\Phi'(X_v)\Phi'(X_v')((\theta_v^{(1)})^2 - 1)|^p] \leq \frac{1}{n_1^p}\|\Phi'\|_\infty^{2p}\mathbb{E}[|\sum_{v=1}^{n_1}(\theta_v^{(1)})^2 - 1|^p] \tag{E.39}$$

$$= \frac{1}{n_1^p}\|\Phi'\|_\infty^{2p}\sum_{\alpha_1,\ldots,\alpha_p=1}^{n_1}\prod_{j=1}^{p}\mathbb{E}[(\theta_{\alpha_j}^{(1)})^2 - 1] \tag{E.40}$$

$$= 0. \tag{E.41}$$

As for the second summand in (E.37), by Theorem B.3 there exists a constant $c_1$ not depending on $n_1$ such that:

$$\mathcal{W}_p^p(\frac{1}{n_1}\sum_{v=1}^{n_1}\Phi'(X_v)\Phi'(X_v'), \mathbb{E}_G[\Phi'(G(x))\Phi'(G(x'))]) \tag{E.42}$$

$$= \mathbb{E}[|\frac{1}{n_1}\sum_{v=1}^{n_1}\Phi'(X_v)\Phi'(X_v') - \mathbb{E}_G[\Phi'(G(x))\Phi'(G(x'))]|^p] \tag{E.43}$$

$$\leq c_1\left(\frac{\text{Lip}\Phi' + \Phi'(0)}{\sqrt{n_1}}\right)^p. \tag{E.44}$$

It remains only to bound the second summand in (E.29). This is done by using again Theorem B.3. There exists a constant $c_2$ not depending on $n_1$ such that:

$$\mathcal{W}_p^p(\frac{1}{n_1}\sum_{v=1}^{n_1}\Phi(X_v)\Phi(X_v'), \mathbb{E}_G[\Phi(G(x))\Phi(G(x'))]) \tag{E.45}$$

$$= \mathbb{E}[\frac{1}{n_1}\sum_{v=1}^{n_1}\Phi(X_v)\Phi(X_v') - \mathbb{E}_G[\Phi(G(x))\Phi(G(x'))]|^p] \tag{E.46}$$

$$\leq c_2\left(\frac{\mathrm{Lip}\Phi + \Phi(0)}{\sqrt{n_1}}\right)^p. \tag{E.47}$$

Putting together all the preceding estimations we obtain:

$$\mathbb{E}[|k_{11} - k_\infty|^p] \leq C\frac{1}{n_1^{\frac{p}{2}}}, \tag{E.48}$$

with $C = 2^{p-1}\max\{2^{2p-2}c_1(\mathrm{Lip}\Phi' + \Phi'(0)), c_2(\mathrm{Lip}\Phi + \Phi(0))\}$. $\qquad\square$

These results suffice to prove Proposition B.7.

*Proof of Proposition B.7.* Consider the joint random variables $\tilde{X} = (k_{\mathcal{X}\mathcal{X}}, \hat{f}(\mathcal{X}))$ and $\tilde{Y} = (k_\infty(\mathcal{X}, \mathcal{X}), G(\mathcal{X}))$. Then Lemma B.1.4 together with Proposition B.4 and Theorem B.3 yield

$$\mathcal{W}_p(\tilde{X}, \tilde{Y}) \leq \mathcal{W}_p(k_{\mathcal{X}\mathcal{X}}, k_\infty(\mathcal{X}, \mathcal{X})) + \mathcal{W}_p(f(\mathcal{X}), G(\mathcal{X})) \leq \frac{C + D}{\sqrt{n_1}}, \tag{E.49}$$

where $C$ is the constant in Proposition B.4 and $D$ the one in Theorem B.3. Both constants do not depend on $n_1$. $\qquad\square$

Lastly, we prove Lemma B.8.

*Proof of Lemma B.8.* Let $\theta_{ij}^{(0)}$ denote the $ij$-th entry of $\theta_0^{(0)} \in \mathbb{R}^{n_0 \times n_1}$, and let $\theta_j^{(1)}$ denote the $j$-th component of $\theta_0^{(1)} \in \mathbb{R}^{n_1}$. By Jensen's inequality and independence of the parameters and $x_1, \ldots x_n$:

$$\mathbb{E}[\|f_0\|] \leq \sqrt{\mathbb{E}[\|\frac{1}{\sqrt{n_1}}\Phi(\mathcal{X}\theta_0^{(0)})\theta_0^{(1)}\|^2]} \tag{E.50}$$

$$\leq \sqrt{n}\sqrt{\mathbb{E}[|\Phi(x_1\theta_{-1}^{(0)})|^2]\mathbb{E}[|\theta_1^{(1)}|^2]} \tag{E.51}$$

$$\leq \sqrt{n}\|\Phi\|_\infty. \tag{E.52}$$

As for the fourth moment,

$$\mathbb{E}[\|f_0\|^4] = \mathbb{E}\left[\left(\sum_{i=1}^{n}\frac{1}{n_1}\left(\sum_{j=1}^{n_1}\Phi(x_i\theta_{-j}^{(0)})\theta_j^{(1)}\right)^2\right)^2\right] \tag{E.53}$$

$$\leq \frac{n^2}{n_1^2}\mathbb{E}\left[\left(\sum_{j=1}^{n_1}\Phi(x_1\theta_{-j}^{(0)})\theta_j^{(1)}\right)^4\right] \tag{E.54}$$

$$\leq \frac{\|\Phi\|_\infty^4 n^2}{n_1^2}(3n_1 + n_1^2) \tag{E.55}$$

$$\leq 4n^2\|\Phi\|_\infty^4. \tag{E.56}$$

Triangular inequality finishes the proof. $\qquad\square$

## E.2 Approximation of the network by linearization

In this subsection we prove the results involved in the proof of Proposition 3.6.

With a slight abuse of notation, we will denote by $\|x - \mathcal{X}\|$ the positive quantity $\sup_{1 \leq i \leq n} \|x - x_i\|$. Also, given any matrix $A = (a_{ij})_{\substack{1 \leq i \leq n \\ 1 \leq j \leq m}}$ we will denote by $\frac{\partial}{\partial A} f$ the matrix $\nabla_A f = (\frac{\partial}{\partial A_{ij}})_{\substack{1 \leq i \leq n \\ 1 \leq j \leq m}}$. We will consider the *linearized gradient flow*, given by

$$\frac{\partial}{\partial t}\overline{\theta}_t = -\nabla_\theta f_0(f^{\text{lin}}(\mathcal{X}; \overline{\theta}_t) - y).$$

For this subsection introduce the following notations: $f_t^{\text{lin}} = f^{\text{lin}}(\mathcal{X}; \overline{\theta}_t)$ and $\overline{y}_t = f^{\text{lin}}(x; \overline{\theta}_t)$.

*Proof of Lemma B.12.* Let $\lambda_{\min}$ be the smallest eigenvalue of $k_t$. By gradient flow equations for the parameters $\theta_t$ and $\overline{\theta}_t$:

$$\|f_t - y\| \leq e^{-\lambda_{\min}t}\|f_0 - y\|, \tag{E.57}$$

$$\|f_t^{\text{lin}} - y\| \leq e^{-\lambda_{\min}^0 t}\|f_0 - y\|. \tag{E.58}$$

On the other hand, Lemma B.2 combined with Cauchy-Schwarz's inequality and the gradient flow equations produces the following system of differential inequalities:

$$\frac{\partial}{\partial t}(\theta_v^{(1)})_t \leq \frac{1}{\sqrt{n_1}}\|\Phi\|_\infty\|f_0 - y\|e^{-\lambda_{\min}t}, \tag{E.59}$$

$$\frac{\partial}{\partial t}(\theta_{uv}^{(0)})_t \leq \frac{1}{\sqrt{n_1 n_0}}\|\Phi'\|_\infty\|f_0 - y\|\|\mathcal{X}_u\|(\theta_v^{(1)})_t e^{-\lambda_{\min}t}. \tag{E.60}$$

The previous is a triangular system of differential inequalities of the form

$$\begin{cases} \frac{\partial}{\partial t}(\theta_v^{(1)})_t & \leq B_1 e^{-\lambda_{\min}t} \\ \frac{\partial}{\partial t}(\theta_{uv}^{(0)})_t & \leq B_0(\theta_v^{(1)})_t e^{-\lambda_{\min}t}, \end{cases}$$

with $B_1 = \frac{1}{\sqrt{n_1}}\|\Phi\|_\infty\|f_0 - y\|$ and $B_0 = \frac{1}{\sqrt{n_1 n_0}}\|\Phi'\|_\infty\|f_0 - y\|\|\mathcal{X}_u\|$.

By integration on $[0, t]$ and substitution we get:,

$$(\theta_v^{(1)})_t \leq (\theta_v^{(1)})_0 + B_1 I_t(\lambda_{\min}) \tag{E.61}$$

$$\leq (\theta_v^{(1)})_0 + \frac{\|\Phi\|_\infty\|f_0 - y\|}{\sqrt{n_1}}I_t(\lambda_{\min}), \tag{E.62}$$

$$(\theta_{uv}^{(0)})_t \leq (\theta_{uv}^{(0)})_0 + B_0 B_1 \int_0^t I_s(\lambda_{\min})ds + B_0\|\theta_0^{(1)}\|I_t(\lambda_{\min}) \tag{E.63}$$

$$\leq (\theta_{uv}^{(0)})_0 + \frac{\|\Phi\|_\infty\|\Phi'\|_\infty\|f_0 - y\|^2\|\mathcal{X}_u\|}{2n_1\sqrt{n_0}}I_t(\lambda_{\min})^2 \tag{E.64}$$

$$+ \frac{\|\Phi'\|_\infty\|f_0 - y\|\|\mathcal{X}_u\|}{\sqrt{n_1 n_0}}I_t(\lambda_{\min})(\theta_v^{(1)})_0. \tag{E.65}$$

Note that in the last inequality, we used $\int_0^t I_s(b)ds \leq \frac{I_t(b)^2}{2}$, for any $b \geq 0$.

Thanks to (E.57), the linearised parameters $\overline{\theta}_t$ also satisfy the preceding inequalities, and hence the thesis holds. $\square$

*Proof of Lemma B.14.* Let $\Phi$ denote the CDF of a standard Gaussian variable. For each $a > 0$, since the entries of $\theta_0^{(1)}$ are $n_1$ i.i.d. standard Gaussian variables,

$$\mathbb{P}(\|\theta_0^{(1)}\|_\infty \leq a) = (1 - 2(1 - \Phi(a)))^{n_1}. \tag{E.66}$$

Bernouilli's inequality and standard estimations for Gaussian tails yield

$$\mathbb{P}(\|\theta_0^{(1)}\|_\infty \leq a) \geq 1 - 2n_1(1 - \Phi(a)) \tag{E.67}$$

$$\geq 1 - n_1 \exp\left(-\frac{a^2}{2}\right). \tag{E.68}$$

Let $r \geq 1$ and put $a = \sqrt{r\gamma \log n_1}$. Then:

$$\mathbb{P}(\|\theta_0^{(1)}\|_\infty \leq a) \geq 1 - n_1 \exp\left(-\frac{r\gamma \log n_1}{2}\right) \tag{E.69}$$

$$= 1 - n_1 \exp\left(\log n_1^{-\frac{r\gamma}{2}}\right) \tag{E.70}$$

$$= 1 - \frac{1}{n_1^{\frac{r\gamma}{2}-1}}. \tag{E.71}$$

$$\square$$

*Proof of Lemma B.15.* We will write $f$ for short of $f_0(x)(\theta)$. By Lemma B.2 and Cauchy-Schwarz's inequality,

$$\|\nabla_\theta f\|^2 = \|\frac{\partial}{\partial \theta^{(0)}} f\|^2 + \|\frac{\partial}{\partial \theta^{(1)}} f\|^2 \tag{E.72}$$

$$\leq \frac{1}{n_0 n_1} \|x\|^2 \|\Phi'\|_\infty^2 \|\theta^{(1)}\|^2 + \frac{1}{n_1} \|\Phi\|_\infty^2. \tag{E.73}$$

Then the first claim follows by (B.34) and the elementary inequality $\sqrt{a+b} \leq \sqrt{a} + \sqrt{b}$, for $a, b \geq 0$.

Now we prove the second inequality. Let $\theta, \tilde{\theta} \in \mathbb{R}^N$, then,

$$\|\nabla_\theta f(\theta) - \nabla_\theta f(\tilde{\theta})\|^2 = \|\frac{\partial}{\partial \theta^{(0)}} f(\theta) - \frac{\partial}{\partial \theta^{(0)}} f(\tilde{\theta})\|^2 + \|\frac{\partial}{\partial \theta^{(1)}} f(\theta) - \frac{\partial}{\partial \theta^{(1)}} f(\tilde{\theta})\|^2. \tag{E.74}$$

Let us estimate the first summand in the previous expression. By Lemma B.2, for each $1 \leq u \leq n_0, 1 \leq v \leq n_1$,

$$\left|\frac{\partial}{\partial \theta_{uv}^{(0)}} f(\theta) - \frac{\partial}{\partial \theta_{uv}^{(0)}} f(\tilde{\theta})\right| \tag{E.75}$$

$$\leq \frac{1}{\sqrt{n_1 n_0}} x_u \left(\Phi'\left(\frac{1}{\sqrt{n_0}} \sum_{j=1}^{n_0} x_j \theta_{jv}^{(0)}\right)\theta_v^{(1)} - \Phi'\left(\frac{1}{\sqrt{n_0}} \sum_{j=1}^{n_0} x_j \tilde{\theta}_{jv}^{(0)}\right)\tilde{\theta}_v^{(1)}\right) \tag{E.76}$$

$$\leq \frac{x_u}{\sqrt{n_1 n_0}} \Phi'\left(\frac{1}{\sqrt{n_0}} \sum_{j=1}^{n_0} x_j \theta_{jv}^{(0)}\right)(\theta_v^{(1)} - \tilde{\theta}_v^{(1)}) \tag{E.77}$$

$$+ \frac{x_u \tilde{\theta}_v^{(1)}}{\sqrt{n_1 n_0}} x_u \left(\Phi'\left(\frac{1}{\sqrt{n_0}} \sum_{j=1}^{n_0} x_j \theta_{jv}^{(0)}\right) - \Phi'\left(\frac{1}{\sqrt{n_0}} \sum_{j=1}^{n_0} x_j \tilde{\theta}_{jv}^{(0)}\right)\right) \tag{E.78}$$

$$\leq \frac{x_u \|\Phi'\|_\infty (\theta_v^{(1)} - \tilde{\theta}_v^{(1)})}{\sqrt{n_1 n_0}} + \frac{x_u \mathrm{Lip}\Phi' \tilde{\theta}_v^{(1)}}{\sqrt{n_1 n_0}} \sum_{j=1}^{n_0} x_j (\theta_{jv}^{(0)} - \tilde{\theta}_{jv}^{(0)}). \tag{E.79}$$

Hence,

$$\|\frac{\partial}{\partial \theta^{(0)}} f(\theta) - \frac{\partial}{\partial \theta^{(0)}} f(\tilde{\theta})\|^2 \tag{E.80}$$

$$= \sum_{\substack{u=1,\ldots,n_0 \\ v=1,\ldots,n_1}} \left|\frac{\partial}{\partial \theta_{uv}^{(0)}} f(\theta) - \frac{\partial}{\partial \theta_{uv}^{(0)}} f(\tilde{\theta})\right|^2 \tag{E.81}$$

$$\leq \frac{\|x\|^2 \|\Phi'\|_\infty^2 \|\theta^{(1)} - \tilde{\theta}^{(1)}\|^2}{n_1 n_0} + \frac{\|x\|^4 (\mathrm{Lip}\Phi')^2 \|\tilde{\theta}^{(1)}\|_\infty^2 \|\theta^{(0)} - \tilde{\theta}^{(0)}\|^2}{n_1 n_0^2} \tag{E.82}$$

$$\leq \frac{\|x\|^2 \|\Phi'\|_\infty^2 \|\theta^{(1)} - \tilde{\theta}^{(1)}\|^2}{n_1 n_0} + \frac{\|x\|^4 (\mathrm{Lip}\Phi')^2 \|\theta^{(0)} - \tilde{\theta}^{(0)}\|^2}{n_1 n_0^2} r\gamma \log n_1, \tag{E.83}$$

with probability greater or equal than $1 - \frac{1}{n^{\frac{r\gamma}{2}-1}}$, where in the last step we used Lemma B.14.

Similarly, the second summand can be estimated as follows. First compute the partial derivatives by using Lemma B.2, for each $1 \le v \le n_1$:

$$|\frac{\partial}{\partial\theta_v^{(1)}}f(\theta) - \frac{\partial}{\partial\theta_v^{(1)}}f(\tilde{\theta})| \tag{E.84}$$

$$\le \frac{1}{\sqrt{n_1}}(\Phi(\frac{1}{\sqrt{n_0}}\sum_{j=1}^{n_0}x_j\theta_{jv}^{(0)}) - \Phi(\frac{1}{\sqrt{n_0}}\sum_{j=1}^{n_0}x_j\tilde{\theta}_{jv}^{(0)})) \tag{E.85}$$

$$\le \frac{\text{Lip}\Phi}{\sqrt{n_1 n_0}}\sum_{j=1}^{n_0}x_j(\theta_{jv}^{(0)} - \tilde{\theta}_{jv}^{(0)}). \tag{E.86}$$

Therefore,

$$\|\frac{\partial}{\partial\theta^{(1)}}f(\theta) - \frac{\partial}{\partial\theta^{(1)}}f(\tilde{\theta})\|^2 \tag{E.87}$$

$$= \sum_{v=1,\ldots,n_1}|\frac{\partial}{\partial\theta_v^{(1)}}f(\theta) - \frac{\partial}{\partial\theta_v^{(1)}}f(\tilde{\theta})|^2 \tag{E.88}$$

$$\le \frac{(\text{Lip}\Phi)^2\|x\|^2}{n_1 n_0}\|\theta^{(0)} - \tilde{\theta}^{(0)}\|^2. \tag{E.89}$$

The preceding estimations, together with $\frac{\|\theta^{(i)}-\tilde{\theta}^{(i)}\|^2}{\|\theta-\tilde{\theta}\|^2} \le 1$ for $i = 0, 1$, yield:

$$\frac{\|\nabla_\theta f(\theta) - \nabla_\theta f(\tilde{\theta})\|^2}{\|\theta - \tilde{\theta}\|^2} \tag{E.90}$$

$$\le \frac{\|x\|^2\|\Phi'\|_\infty^2}{n_1 n_0} + \frac{\|x\|^4(\text{Lip}\Phi')^2}{n_1 n_0^2}r\gamma\log n_1 + \frac{(\text{Lip}\Phi)^2\|x\|^2}{n_1 n_0}. \tag{E.91}$$

Taking the square root in the last inequality yields the thisis. $\square$

*Proof of Lemma B.16.* Fix $\gamma \in \mathbb{N}$. The probability of $Z = \|k - k_\infty\| > \frac{\gamma\lambda_{\min}^\infty}{2}$ can be estimated with Markov's inequality and Proposition B.4. There exists a constant $C > 0$ not depending on $n_1$ such that:

$$\mathbb{P}(Z > \frac{\gamma\lambda_{\min}^\infty}{2}) = \mathbb{P}\left(Z^p > \left(\frac{\gamma\lambda_{\min}^\infty}{2}\right)^p\right) \tag{E.92}$$

$$\le \left(\frac{2}{\gamma\lambda_{\min}^\infty}\right)^p \mathbb{E}[\|Z\|^p] \tag{E.93}$$

$$= \left(\frac{2}{\gamma\lambda_{\min}^\infty}\right)^p \mathcal{W}_p^p(k, k_\infty) \tag{E.94}$$

$$\le \left(\frac{2}{\gamma\lambda_{\min}^\infty}\right)^p \frac{C}{n_1^{\frac{p}{2}}}. \tag{E.95}$$

Note that Proposition B.4 holds for every natural $p$. This concludes the proof. $\square$

Now are ready to prove Proposition B.9:

*Proof of Proposition B.9.* For the sake of clearness we introduce the following abbreviations for the remainder of the proof. Let $y_t = f(x; \theta_t), \bar{y}_t = f^{\text{lin}}(x; \bar{\theta}_t), f_t = f(\mathcal{X}; \theta_t)$ and $f_t^{\text{lin}} = f^{\text{lin}}(\mathcal{X}; \bar{\theta}_t)$. Also, let $k_t = k(\mathcal{X}, \mathcal{X}; \theta_t), \nabla = \nabla_\theta$ and let $L(\mathcal{X})$ denote the Lipschitz constant of $\nabla f$, seen as a function of $\theta$.

Consider the empirical risk for the quadratic loss $\mathcal{R}_\mathcal{D}(\theta_t) = \frac{1}{2}\sum_{i=1}^n(f^{(L)}(x_i; \theta_t) - y)^2$.

From gradient flow equations we have:

$$\frac{\partial}{\partial t} f_t = -k_t(f_t - y), \tag{E.96}$$

$$\frac{\partial}{\partial t} \|f_t - y\|^2 = -2\langle f_t - y, k_t(f_t - y)\rangle. \tag{E.97}$$

Let $t_* = \inf\{t \mid \|\theta_t - \theta_0\| > \frac{\sigma_{\min}}{2L(\mathcal{X})}\}$ Then for each $t \le t_*$, by 1-Lipschitzianity of the smallest eigenvalue with respect to the operator norm, and by definition of $t_*$, we obtain an upper bound for $\lambda_{\min}(k_t)$:

$$|\lambda_{\min}(k_t) - \lambda_{\min}| \le \|k_t - k_0\|_{op} \le \|k_t - k_0\| \le L(\mathcal{X})\|\theta_t - \theta_0\| \le \frac{\sigma_{\min}}{2},$$

which implies:

$$\lambda_{\min}(k_t) \ge \lambda_{\min} - \frac{\sigma_{\min}}{2} \ge \frac{\lambda_{\min}}{4}.$$

This estimation combined with Grönwall's inequality applied to (E.97) yield:

$$\|f_t - y\|^2 \le \|f_0 - y\|^2 \exp\left(-\frac{\lambda_{\min}}{2}t\right). \tag{E.98}$$

From (E.97) and Cauchy-Schwarz we deduce:

$$\frac{\partial}{\partial t}\|f_t - y\| = -\frac{\|\nabla f_t(f_t - y)\|^2}{\|f_t - y\|} \tag{E.99}$$

$$\le -\frac{\sigma_{\min}}{2}\|\nabla f_t(f_t - y)\|. \tag{E.100}$$

Hence,

$$\frac{\partial}{\partial t}\left(\|f_t - y\| + \frac{\sigma_{\min}}{2}\|\theta_t - \theta_0\|\right) \le \frac{\partial}{\partial t}\|f_t - y\| + \frac{\sigma_{\min}}{2}\|\frac{\partial}{\partial t}\theta_t\| \le 0. \tag{E.101}$$

for all $t \le t_*$.

Thus, for all $t \le t_*$:

$$\|\theta_t - \theta_0\| \le \frac{2}{\sigma_{\min}}\|f_0 - y\|. \tag{E.102}$$

Let us show that this property holds for all $t > 0$. By contradiction assume $t_* < \infty$. (E.102) with Assumption 5 implies

$$\|\theta_{t_*} - \theta_0\| < \frac{2}{\sigma_{\min}}\frac{\sigma_{\min}^2}{4L(\mathcal{X})} \tag{E.103}$$

$$= \frac{\sigma_{\min}}{2L(\mathcal{X})}. \tag{E.104}$$

In particular the last inequality holds for $t_*$, which contradicts its definition. Hence $t_* = \infty$.

Let us now prove the rest of the inequalities in the theorem.

The gradient flow equation for the linearised network reads:

$$\frac{\partial}{\partial t} f_t^{\text{lin}} = -k_0(f_t^{\text{lin}} - y). \tag{E.105}$$

Define the difference $r_t = f_t - f_t^{\text{lin}}$. Then

$$\frac{\partial}{\partial t} r_t = -k_t(f_t - y) + k_0(f_t^{\text{lin}} - y) \tag{E.106}$$

$$= -k_t r_t - (k_t - k_0)(f_t^{\text{lin}} - y). \tag{E.107}$$

Then, by Cauchy-Schwarz and (E.98) combined with (E.105),

$$\frac{1}{2}\frac{\partial}{\partial t}\|r_t\|^2 = -\langle r_t, k_t r_t\rangle - \langle r_t, (k_t - k_0)(f_t^{\text{lin}} - y)\rangle \tag{E.108}$$

$$\leq -\lambda_{\min}(k_t)\|r_t\|^2 + \|r_t\|\|k_t - k_0\|\|f_t^{\text{lin}} - y\| \tag{E.109}$$

$$\leq -\frac{\lambda_{\min}}{4}\|r_t\|^2 + \|r_t\|\|k_t - k_0\|\|f_0 - y\|\exp\left(-\frac{\lambda_{\min}t}{4}\right). \tag{E.110}$$

Hence,

$$\frac{\partial}{\partial t}\|r_t\| \leq -\frac{\lambda_{\min}}{4}\|r_t\| + \|k_t - k_0\|\|f_0 - y\|\exp\left(-\frac{\lambda_{\min}t}{4}\right). \tag{E.111}$$

Now let us bound separately the different factors in the previous equation. The norm of the difference between the kernels can be estimated as:

$$\|k_t - k_0\| \leq \|\nabla f_t \nabla f_t^\top - \nabla f_0 \nabla f_0^\top\| \tag{E.112}$$

$$\leq 2\|\nabla f_0\|\|\nabla f_t - \nabla f_0\| + \|\nabla f_t - \nabla f_0\|^2 \tag{E.113}$$

$$\leq 2\sigma_{\max}L(\mathcal{X})\|\theta_t - \theta_0\| + L(\mathcal{X})^2\|\theta_t - \theta_0\|^2 \tag{E.114}$$

$$\leq 2\sigma_{\max}L(\mathcal{X})\|\theta_t - \theta_0\| + L(\mathcal{X})\|\theta_t - \theta_0\|\frac{\sigma_{\min}}{2} \tag{E.115}$$

$$\leq \frac{5}{2}\sigma_{\max}L(\mathcal{X})\|\theta_t - \theta_0\|, \tag{E.116}$$

where in (E.115) we applied the definition of $t_*$.

Moreover, by Grönwall and Cauchy-Schwarz inequalities we have

$$\|r_t\| \leq \exp\left(-\frac{\lambda_{\min}t}{4}\right)\|f_0 - y\|\int_0^t \|k_s - k_0\|ds \tag{E.117}$$

$$\leq \exp\left(-\frac{\lambda_{\min}t}{4}\right)\|f_0 - y\|\sup_{s\geq 0}\|k_s - k_0\| \tag{E.118}$$

$$\leq \exp\left(-\frac{\lambda_{\min}t}{4}\right)\frac{5}{2}\sigma_{\max}L(\mathcal{X})\|f_0 - y\|\sup_{s\geq 0}\|\theta_s - \theta_0\| \tag{E.119}$$

$$\leq \exp\left(-\frac{\lambda_{\min}t}{4}\right)\frac{5}{2}\sigma_{\max}L(\mathcal{X})\|f_0 - y\|\sup_{s\geq 0}\frac{2}{\sigma_{\min}}\|f_0 - y\| \tag{E.120}$$

$$\leq \exp\left(-\frac{\lambda_{\min}t}{4}\right)\frac{5\sigma_{\max}}{\sigma_{\min}}L(\mathcal{X})\|f_0 - y\|^2 \tag{E.121}$$

$$\tag{E.122}$$

Moreover, by taking the difference of the gradient flow equations for $\theta_t$ and $\overline{\theta}_t$ we obtain:

$$\frac{\partial}{\partial t}\|\theta_t - \overline{\theta}_t\| \leq \|\nabla f_t - \nabla f_0\|\|f_t - y\| + \|\nabla f_0\|\|f_t - f_t^{\text{lin}}\| \tag{E.123}$$

$$\leq L(\mathcal{X})\|\theta_t - \theta_0\|\|f_t - y\| + \sigma_{\max}\|f_t - f_t^{\text{lin}}\| \tag{E.124}$$

$$\leq \frac{2L(\mathcal{X})}{\sigma_{\min}}\|f_0 - y\|^2\exp\left(-\frac{\lambda_{\min}t}{4}\right) \tag{E.125}$$

$$+ \frac{5\sigma_{\max}^2}{\sigma_{\min}}L(\mathcal{X})\|f_0 - y\|^2\exp\left(-\frac{\lambda_{\min}t}{4}\right) \tag{E.126}$$

$$\leq \frac{(2 + 5\sigma_{\max}^2)L(\mathcal{X})}{\sigma_{\min}}\|f_0 - y\|^2\exp\left(-\frac{\lambda_{\min}t}{4}\right). \tag{E.127}$$

where in (E.125) we used (E.102), (E.98) and E.121.

Integrating the previous inequality we obtain:

$$\|\theta_t - \overline{\theta}_t\| \leq \frac{(2 + 5\sigma_{\max}^2)L(\mathcal{X})}{\sigma_{\min}}\|f_0 - y\|^2 \int_0^t \exp\left(-\frac{\lambda_{\min}s}{4}\right) ds \tag{E.128}$$

$$\leq \frac{4(2 + 5\sigma_{\max}^2)L(\mathcal{X})}{\sigma_{\min}^3}\|f_0 - y\|^2 \left(1 - \exp\left(-\frac{\lambda_{\min}s}{4}\right)\right) \tag{E.129}$$

$$\leq \frac{(8 + 20\sigma_{\max}^2)L(\mathcal{X})}{\sigma_{\min}^3}\|f_0 - y\|^2. \tag{E.130}$$

Now we are ready to prove the last inequality in the thesis. Decompose by triangle inequality:

$$\|y_t - \overline{y}_t\| \leq \|y_t - f^{\lin}(x; \theta_t)\| + \|f^{\lin}(x; \theta_t) - \overline{y}_t\|. \tag{E.131}$$

First, let us focus on the first summand of (E.131). Denote by $L(x)$ the Lipschitz constant of $\nabla y_0$ seen as a function of $\theta$. Then, by Lemma B.15,

$$\|y_t - f^{\lin}(x; \theta_t)\| = \|\int_0^t (\nabla f(x; \theta_s) - \nabla f(x; \theta_0))\frac{\partial}{\partial t}\theta_s ds\| \tag{E.132}$$

$$\leq L(x)\sup_{t \geq 0}\|\theta_t - \theta_0\| \int_0^t \|\frac{\partial}{\partial t}\theta_s\| ds \tag{E.133}$$

$$\leq L(x)\sup_{t \geq 0}\|\theta_t - \theta_0\| \cdot \frac{2}{\sigma_{\min}}\|y - f_0\| \tag{E.134}$$

$$\leq L(x)\frac{4\|y - f_0\|^2}{\lambda_{\min}}, \tag{E.135}$$

where in the third inequality we used (E.101) and (E.102) on the last one.

As for the second summand of (E.131), by (E.130) and Lemma B.15:

$$\|f^{\lin}(x; \theta_t) - \overline{y}_t\| = \|\nabla f(x; \theta_0)(\theta_t - \overline{\theta}_t)\| \tag{E.136}$$

$$\leq \frac{(8 + 20\sigma_{\max}^2)L(\mathcal{X})}{\sigma_{\min}^3}\|f_0 - y\|^2\|\nabla f(x; \theta_0)\|. \tag{E.137}$$

Combining the two preceding estimations, we obtain the thesis.

$$\square$$

Lastly, we prove Theorem B.10.

*Proof of Theorem B.10.* We prove the three inequalities separately. Let $\lambda_{\min}$ denote the smallest eigenvalue of $k_t$.

- By Lemma B.2 and Cauchy-Schwarz's inequality,

$$\|y_t - f_t\| \leq \frac{1}{\sqrt{n_1 n_0}}\text{Lip}\Phi\|x - \mathcal{X}\|\|\theta_t^{(0)}\theta_t^{(1)}\|. \tag{E.138}$$

Recall that $I_t(\lambda_{\min}) \leq t$. Then the norm $\|\theta_t^{(0)}\theta_t^{(1)}\|^2$ can be estimated with the aid of Lemma B.12:

$$\|\theta_t^{(0)}\theta_t^{(1)}\|^2 = \sum_{u=1}^{n_0}\left(\sum_{v=1}^{n_1}(\theta_{uv}^{(0)})_t(\theta_v^{(1)})_t\right)^2 \tag{E.139}$$

$$\leq \sum_{u=1}^{n_0}\left(\sum_{v=1}^{n_1}(\theta_v^{(1)})_0(\theta_{uv}^{(0)})_0 + \frac{a_1(\theta_{uv}^{(0)})_0}{\sqrt{n_1}}\psi(\theta_0)t + \frac{a_0(\theta_v^{(1)})_0}{n_1\sqrt{n_0}}\psi(\theta_0)^2t^2\right. \tag{E.140}$$

$$\left.+ \frac{a_0a_1}{n_1^{\frac{3}{2}}\sqrt{n_0}}\psi(\theta_0)^3t^3 + \frac{a_0'(\theta_v^{(1)})_0^2}{\sqrt{n_1 n_0}}\psi(\theta_0)t + \frac{a_0'a_1(\theta_v^{(1)})_0}{n_1\sqrt{n_0}}\psi(\theta_0)^2t^2\right)^2 \tag{E.141}$$

$$\leq \sum_{u,v} n_1(\theta_v^{(1)})_0^2(\theta_{uv}^{(0)})_0^2 + a_1^2(\theta_{uv}^{(0)})_0^2\psi(\theta_0)^2t^2 + \frac{a_0^2(\theta_v^{(1)})_0^2}{n_1 n_0}\psi(\theta_0)^4t^4 \tag{E.142}$$

$$+ \frac{a_0^2a_1^2}{n_1^2 n_0}\psi(\theta_0)^6t^6 + \frac{a_0'^2(\theta_v^{(1)})_0^4}{n_0}\psi(\theta_0)^2t^2 + \frac{a_0'^2a_1^2(\theta_v^{(1)})_0^2}{n_1 n_0}\psi(\theta_0)^4t^4 \tag{E.143}$$

$$\leq n_1\|\theta_0^{(0)}\theta_0^{(1)}\|^2 + a_1^2\|\theta_0^{(0)}\|^2\psi(\theta_0)^2t^2 + \frac{a_0^2\|\theta_0^{(1)}\|^2}{n_1}\psi(\theta_0)^4t^4 \tag{E.144}$$

$$+ \frac{a_0^2a_1^2}{n_1}\psi(\theta_0)^6t^6 + a_0'^2\|\theta_0^{(1)}\|^4\psi(\theta_0)^2t^2 + \frac{a_0'^2a_1^2\|\theta_0^{(1)}\|^2}{n_1}\psi(\theta_0)^4t^4. \tag{E.145}$$

with $a_0 = \frac{1}{2}\|\Phi\|_\infty\|\Phi'\|_\infty\|\mathcal{X}_u\|$, $a_0' = \|\Phi'\|_\infty\|\mathcal{X}_u\|$ and $a_1 = \|\Phi\|_\infty$.

Hence,

$$\|y_t - f_t\|^2 \leq \frac{(\mathrm{Lip}\Phi)^2\|x - \mathcal{X}\|}{n_0 n_1}\|\theta_t^{(0)}\theta_t^{(1)}\|^2 \tag{E.146}$$

$$\leq \frac{A_0}{n_0}\|\theta_0^{(0)}\theta_0^{(1)}\|^2 + \frac{A_1t^2}{n_0 n_1}\|\theta_0^{(0)}\|^2\psi(\theta_0)^2 + \frac{A_2\|\theta_0^{(1)}\|^2t^4}{n_1^2 n_0}\psi(\theta_0)^4 \tag{E.147}$$

$$+ \frac{A_3t^6}{n_1^2 n_0}\psi(\theta_0)^6 + \frac{A_4t^2}{n_1 n_0}\|\theta_0^{(1)}\|^4\psi(\theta_0)^2 + \frac{A_5t^4\|\theta_0^{(1)}\|^2}{n_1^2 n_0}\psi(\theta_0)^4. \tag{E.148}$$

with $A_0 = (\mathrm{Lip}\Phi)^2\|x - \mathcal{X}\|^2$, $A_1 = a_1^2$, $A_2 = a_0^2$, $A_3 = a_1^2a_0^2$, $A_4 = a_0'^2$ and $A_5 = a_0'^2a_1^2$.

- We follow a similar strategy to prove the second inequality in the Theorem. Put $\overline{w}_t = \overline{\theta}_t - \theta_0$. By the triangle inequality and Cauchy-Schwarz we decompose:

$$\|f^{\mathrm{lin}} - \overline{y}_t\|^2 \leq 2\|f_0 - y_0\|^2 + 2\|\nabla_\theta f_0 - \nabla_\theta y_0\|^2\|\overline{w}_t\|^2. \tag{E.149}$$

The first summand in (E.149) is bounded exactly as the first summand in (E.138) by setting $t = 0$:

$$\|f_0 - y_0\|^2 \leq \frac{(\mathrm{Lip}\Phi)^2\|x - \mathcal{X}\|^2}{n_1 n_0}\|\theta_0^{(0)}\theta_0^{(1)}\|^2 \leq \frac{A_0}{n_1 n_0}\|\theta_0^{(0)}\theta_0^{(1)}\|^2. \tag{E.150}$$

As for the second summand in (E.149), we decompose by Lemma B.2. Factoring out $\max_i\{(\theta_0^{(1)})_i\} = \|\theta_0^{(1)}\|_\infty$ permits us to write:

$$\|\nabla_\theta f_0 - \nabla_\theta y_0\|^2 \leq \|\frac{\partial}{\partial\theta^{(0)}}(f_0 - y_0)\|^2 + \|\frac{\partial}{\partial\theta^{(1)}}(f_0 - y_0)\|^2 \tag{E.151}$$

$$\leq \frac{1}{n_1 n_0}\|(\mathcal{X}^\top\Phi'(\frac{1}{\sqrt{n_0}}\mathcal{X}\theta_0^{(0)}) - x^\top\Phi'(\frac{1}{\sqrt{n_0}}x\theta_0^{(0)}))\theta_0^{(1)}\|^2 \tag{E.152}$$

$$+ \frac{1}{n_1}\|\Phi(\frac{1}{\sqrt{n_0}}\mathcal{X}\theta_0^{(0)}) - \Phi(\frac{1}{\sqrt{n_0}}x\theta_0^{(0)})\|^2 \tag{E.153}$$

$$\leq \frac{2}{n_1 n_0}\|(\mathcal{X}^\top\Phi'(\frac{1}{\sqrt{n_0}}\mathcal{X}\theta_0^{(0)}) - \mathcal{X}^\top\Phi'(\frac{1}{\sqrt{n_0}}x\theta_0^{(0)}))\theta_0^{(1)}\|^2 \tag{E.154}$$

$$+ \frac{2}{n_1 n_0}\|(\mathcal{X}^\top\Phi'(\frac{1}{\sqrt{n_0}}x\theta_0^{(0)}) - x^\top\Phi'(\frac{1}{\sqrt{n_0}}x\theta_0^{(0)}))\theta_0^{(1)}\|^2 \tag{E.155}$$

$$+ \frac{A_0}{n_1 n_0}\|\theta_0^{(0)}\|^2 \tag{E.156}$$

$$\leq \frac{2}{n_1 n_0^2}(\text{Lip}\Phi')^2\|\mathcal{X}\|^2\|x - \mathcal{X}\|^2\|\theta_0^{(0)}\theta_0^{(1)}\|^2 \tag{E.157}$$

$$+ \frac{2}{n_1 n_0}\|\Phi'\|_\infty^2\|x - \mathcal{X}\|^2\|\theta_0^{(1)}\|^2 + \frac{A_0}{n_1 n_0}\|\theta_0^{(0)}\|^2. \tag{E.158}$$

Moreover we can bound the norm of $\overline{w}_t$ with Lemma B.12:

$$\|\overline{w}_t\|^2 \leq \|\overline{\theta}_t^{(0)} - \theta_0^{(0)}\|^2 + \|\overline{\theta}_t^{(1)} - \theta_0^{(1)}\|^2 \tag{E.159}$$

$$\leq \sum_{u,v}\frac{2a_0^2}{n_1^2 n_0}\psi(\theta_0)^4 t^4 + \frac{2a_0'^2(\theta_v^{(1)})_0^2}{n_1 n_0}\psi(\theta_0)^2 t^2 + \sum_v\frac{a_1^2}{n_1}\psi(\theta_0)^2 t^2 \tag{E.160}$$

$$\leq \frac{2a_0^2}{n_1}\psi(\theta_0)^4 t^4 + \frac{2a_0'^2\|\theta^{(1)}\|^2}{n_1}\psi(\theta_0)^2 t^2 + a_1^2\psi(\theta_0)^2 t^2. \tag{E.161}$$

Hence, (E.149) can be written as:

$$\|f^{\text{lin}} - \overline{y}_t\|^2 \leq \frac{B_0}{n_1 n_0}\|\theta_0^{(0)}\theta_0^{(1)}\|^2 + \frac{B_1}{n_1^2 n_0^2}\|\theta_0^{(0)}\theta_0^{(1)}\|^2\psi(\theta_0)^4 t^4 \tag{E.162}$$

$$+ \frac{B_2}{n_1^2 n_0^2}\|\theta_0^{(0)}\|^2\|\theta_0^{(1)}\|^4\psi(\theta_0)^2 t^2 + \frac{B_3}{n_1 n_0^2}\|\theta_0^{(0)}\theta_0^{(1)}\|^2\psi(\theta_0)^2 t^2 \tag{E.163}$$

$$+ \frac{B_4}{n_1^2 n_0}\|\theta_0^{(1)}\|^2\psi(\theta_0)^4 t^4 + \frac{B_5}{n_1^2 n_0}\|\theta_0^{(1)}\|^4\psi(\theta_0)^2 t^2 \tag{E.164}$$

$$+ \frac{B_6}{n_1 n_0}\|\theta_0^{(1)}\|^2\psi(\theta_0)^2 t^2 + \frac{B_7}{n_1^2 n_0}\|\theta_0^{(0)}\|^2\psi(\theta_0)^4 t^4 \tag{E.165}$$

$$+ \frac{B_8}{n_1^2 n_0}\|\theta_0^{(0)}\|^2\|\theta_0^{(1)}\|^2\psi(\theta_0)^2 t^2 + \frac{B_9}{n_1 n_0}\|\theta_0^{(0)}\|^2\psi(\theta_0)^2 t^2, \tag{E.166}$$

where the constants in the last inequality are, explicitly, $B_0 = 2A_0$, $B_1 = 8(\text{Lip}\Phi'\|\mathcal{X}\|\|x - \mathcal{X}\|a_0)^2$, $B_2 = 8(\text{Lip}\Phi'\|\mathcal{X}\|\|x - \mathcal{X}\|a_0')^2$, $B_3 = 4(\text{Lip}\Phi'\|\mathcal{X}\|\|x - \mathcal{X}\|a_1)^2$, $B_4 = 8(\|\Phi'\|_\infty\|x - \mathcal{X}\|a_0)^2$, $B_5 = 8(\|\Phi'\|_\infty\|x - \mathcal{X}\|a_0')^2$, $B_6 = 4(\|\Phi'\|_\infty\|x - \mathcal{X}\|a_1)^2$, $B_7 = 4A_0 a_0^2$, $B_8 = 4A_0 a_0'^2$, and $B_9 = 2A_0 a_1^2$.

- It remains to estimate the last inequality. Consider $\Delta(t) = \|f_t - f_t^{\text{lin}}\|$ Then by gradient flow equations and Cauchy-Schwarz,

$$\frac{\partial}{\partial t}(\Delta(t)^2) = \langle k_t(f_t - y) - k_0(f_t^{\text{lin}} - y), f_t - f_t^{\text{lin}} \rangle \tag{E.167}$$

$$= \sum_{i=1}^{n} (k_t(x_i, \mathcal{X})(f_t - y) - k_0(x_i, \mathcal{X})(f_t^{\text{lin}} - y))(f_t(x_i) - f_t^{\text{lin}}(x_i)) \tag{E.168}$$

$$= \sum_{i=1}^{n} (k_t(x_i, \mathcal{X}) - k_0(x_i, \mathcal{X}))(f_t - y)(f_t(x_i) - f_t^{\text{lin}}(x_i)) \tag{E.169}$$

$$- k_0(x_i, \mathcal{X})(f_t - f_t^{\text{lin}})(f_t(x_i) - f_t^{\text{lin}}(x_i)) \tag{E.170}$$

$$= \|(k_t - k_0)(f_t - y)(f_t - f_t^{\text{lin}})^\top\|_1 - \|k_0(f_t - f_t^{\text{lin}})(f_t - f_t^{\text{lin}})^\top\|_1 \tag{E.171}$$

By equivalence of the 1-norm and the euclidean norm for $v \in \mathbb{R}^d$ we have $\|v\| \le \|v\|_1 \le \sqrt{d}\|v\|$. Then, by Cauchy-Schwarz's inequality,

$$\frac{\partial}{\partial t}\Delta(t) = n\|k_t - k_0\|\|f_t - y\| - \lambda_{\min}^0 \Delta(t) \tag{E.172}$$

$$\le n\|k_t - k_0\|\psi(\theta_0)e^{-\lambda_{\min}t} - \lambda_{\min}^0 \Delta(t) \tag{E.173}$$

Let us bound the norm of $k_t - k_0$:

$$\|k_t - k_0\| = \|\nabla_\theta f_t(\mathcal{X})\nabla_\theta f_t(\mathcal{X})^\top - \nabla_\theta f_0(\mathcal{X})\nabla_\theta f_0(\mathcal{X})^\top\| \tag{E.174}$$

$$\le \|\nabla_\theta f_t(\mathcal{X}) + \nabla_\theta f_0(\mathcal{X})\|L(\mathcal{X})\|\theta_t - \theta_0\| \tag{E.175}$$

$$\tag{E.176}$$

From Lemmas B.2 and B.12, we have:

$$\|\nabla_\theta f_t(\mathcal{X})\|^2 = \|\nabla_{\theta^{(0)}} f_t(\mathcal{X})\|^2 + \|\nabla_{\theta^{(1)}} f_t(\mathcal{X})\|^2 \tag{E.177}$$

$$\le \frac{\|\mathcal{X}\|^2\|\Phi'\|_\infty^2\|\theta_t^{(1)}\|^2}{n_1 n_0} + \frac{\|\Phi\|_\infty^2}{n_1} \tag{E.178}$$

$$\le \frac{\|\mathcal{X}\|^2\|\Phi'\|_\infty^2}{n_1 n_0}\left(\|\theta_0^{(1)}\| + \frac{\|\Phi\|_\infty \psi(\theta^0)}{\sqrt{n_1}}t\right)^2 + \frac{\|\Phi\|_\infty^2}{n_1}. \tag{E.179}$$

Analogously,

$$\|\nabla_\theta f_0(\mathcal{X})\|^2 = \|\nabla_{\theta^{(0)}} f_0(\mathcal{X})\|^2 + \|\nabla_{\theta^{(1)}} f_0(\mathcal{X})\|^2 \tag{E.180}$$

$$\le \frac{\|\mathcal{X}\|^2\|\Phi'\|_\infty^2\|\theta_0^{(1)}\|^2}{n_1 n_0} + \frac{\|\Phi\|_\infty^2}{n_1}. \tag{E.181}$$

Moreover, again by Lemma B.12,

$$\|\theta_t - \theta_0\|^2 = \|\theta_t^{(0)} - \theta_0^{(0)}\|^2 + \|\theta_t^{(1)} - \theta_0^{(1)}\|^2 \tag{E.182}$$

$$\le \frac{2a_0^2}{n_1^2 n_0}\psi(\theta_0)^4 t^4 + \frac{2a_0'^2}{n_1 n_0}\|\theta_0^{(1)}\|^2 t^2 + \frac{a_1^2\psi(\theta^0)^2}{n_1}t^2 \tag{E.183}$$

$$\tag{E.184}$$

Inequalities (E.179),(E.181) and (E.183) allow us to estimate:

$$\|k_t - k_0\| \le \frac{L(\mathcal{X})}{n_1}\left(\frac{c_1\psi(\theta_0)^2\|\theta_0^{(1)}\|}{\sqrt{n_1}n_0} + \frac{c_2\psi(\theta_0)^3 t^3}{n_1 n_0} + \frac{c_3\psi(\theta_0)^2 t^2}{\sqrt{n_1 n_0}}\right. \tag{E.185}$$

$$+ \frac{c_4\|\theta_0^{(1)}\|^2 t}{n_0} + \frac{c_5\psi(\theta_0)\|\theta_0^{(1)}\|t^2}{\sqrt{n_1}n_0} + \frac{c_6\|\theta_0^{(1)}\|t}{\sqrt{n_0}} \tag{E.186}$$

$$\left. + \frac{c_7\psi(\theta_0)\|\theta_0^{(1)}\|t}{\sqrt{n_0}} + \frac{c_8\psi(\theta_0)^2 t^2}{\sqrt{n_1 n_0}} + c_9\psi(\theta_0)t\right), \tag{E.187}$$

with $c_1 = 2\sqrt{2}a_0\|\mathcal{X}\|\|\Phi'\|_\infty$, $c_2 = \sqrt{2}a_0\|\Phi\|_\infty$, $c_3 = 2\sqrt{2}a_0\|\Phi\|_\infty$, $c_4 = 2\sqrt{2}a_0'\|\mathcal{X}\|\|\Phi'\|_\infty$, $c_5 = \sqrt{2}a_0'\|\Phi\|_\infty$, $c_6 = 2\sqrt{2}a_0'\|\Phi\|_\infty$, $c_7 = 2a_1\|\mathcal{X}\|\|\Phi'\|_\infty$, $c_8 = a_1\|\Phi\|_\infty$ and $c_9 = 2a_1\|\Phi\|_\infty$. Let $C(n_1, n_0, t, \theta_0)$ be the right hand side of (E.185). Then, the reight-hand side of (E.173) can be $\mathbb{P}$-almost surely bounded from above with:

$$\frac{\partial}{\partial t}\Delta(t) \leq n\|k_t - k_0\|\psi(\theta_0)e^{-\lambda_{\min}t} - \lambda_{\min}^0\Delta(t) \tag{E.188}$$

$$\leq nC(n_1, n_0, t, \theta_0). \tag{E.189}$$

In the previous inequality we used that the event $\lambda_{\min} = 0$ has null measure. Integrating, and using that $\Delta(0) = 0$:

$$\Delta(t) \leq \frac{nL(\mathcal{X})}{n_1}\left(\frac{c_1\psi(\theta_0)^2\|\theta_0^{(1)}\|t}{\sqrt{n_1}n_0} + \frac{c_2\psi(\theta_0)^3t^4}{2n_1n_0} + \frac{c_3\psi(\theta_0)^2t^3}{\sqrt{n_1}n_0}\right. \tag{E.190}$$

$$+\frac{c_4\|\theta_0^{(1)}\|^2t^2}{2n_0} + \frac{c_5\psi(\theta_0)\|\theta_0^{(1)}\|t^3}{3\sqrt{n_1}n_0} + \frac{c_6\|\theta_0^{(1)}\|t^3}{2\sqrt{n_0}} \tag{E.191}$$

$$\left.+\frac{c_7\psi(\theta_0)\|\theta_0^{(1)}\|t^2}{2\sqrt{n_0}} + \frac{c_8\psi(\theta_0)^2t^3}{3\sqrt{n_1}n_0} + \frac{c_9\psi(\theta_0)t^2}{2}\right) \tag{E.192}$$

Taking the square, applying the elementary inequality $(\sum_{i=1}^n a_i)^2 \leq n\sum_{i=1}^n a_i^2$, for $a_i \geq 0$, and adjusting the constants yields the desired result.

$\square$

