# OpenReview forum: "Quantitative convergence of trained neural networks to Gaussian processes"
_NeurIPS.cc/2025/Conference — NeurIPS 2025 poster_

### Official Review · Reviewer_3AEa · 2025-06-30

**Clarity:** 3
**Significance:** 3
**Originality:** 3
**Rating:** 5
**Confidence:** 3

**Summary:**

This paper studies the Gaussian approximation of shallow neural networks for arbitrary test points and at any time during gradient-descent training. The  approximation is quantified in terms of the quadratic Wasserstein distance between the two random variables $f_t(x)$ and $G_t(x)$ where $G_t$ is the Gaussian approximation. This result can be extended to training times that scale at most polynomially with the network’s width.  This is done by first controlling the distance between the true and the linearized evolutions, and then by bounding the distance between the linearized dynamics driven by the empirical NTK and that driven by the deterministic limiting NTK.

**Questions:**

- Would it be possible to include a figure showing the best power-law fit to illustrate the bound on the rate of convergence of the Wasserstein distance?
- Could you add a discussion on the dependence of your bound on the test point? For example, does this Gaussian approximation hold uniformly in $x$?

**Ethical Concerns:**

["NO or VERY MINOR ethics concerns only"]

**Final Justification:**

The authors addressed my points (adding a power-law fit and discussion on the x-dependence). After reading the other reviews and answers from the authors, I confirm my score.

**Limitations:**

Yes

**Quality:**

3

**Strengths And Weaknesses:**

Strengths:
- The paper is clearly written, results are in line with expectations and nicely extend previous works, which focused either on the intialization, or provided qualitative results on the training (or results in probability).
- The strategy of the proof given in Section 3.1. provides a good summary of the key steps, and enables readers to grasp the essential ideas.
- The experiments address practical consideration. In particular, the authors take into account the fact that the Wasserstein distance is estimated by considering that of the empirical measures. Section 4.2 explicitly discusses the number of samples required to obtain a good approximation.


Weaknesses:
- While the qualitative convergence was previously know, explicit quantitative rates were not derived for the training dynamics. The experiments could have been more illuminating if they explicitly showed the best power-law fits capturing the the decay of the Wasserstein distance.
- The quantitative bounds shed light the effects of the network width, input dimension, and training time. However, a discussion on the dependence on the test point $x$ would have been useful.

---

> ### Author Rebuttal · Authors · 2025-07-30
>
> We thank Reviewer 3AEa for their careful reading of our manuscript and for the constructive comments.
> We now address each of the weaknesses and questions raised by the reviewer:
>
> 1. Would it be possible to include a figure showing the best power-law fit to illustrate the bound on the rate of convergence of the Wasserstein distance?:
>
> Due to the reviewing policies of Neurips we cannot upload pictures or .pdf files to the responses, but we modified
> Experiment 2 to include the power-law fit between the points in the plot, in a different color.
> This is also acknowledged in the body of the numerical experiments section.
>
> 2. Could you add a discussion on the dependence of your bound on the test point? For example, does this Gaussian approximation hold uniformly in x?
>
> We thank the reviewer for pointing out an interesting question.
> The dependence of our bounds in the choice of the test point comes from Theorem B.10 and Proposition 3.7, and this dependence was previously not discussed in our draft.
> A new paragraph in the discussion section has been added to address this:
>
> The bound in our Theorem 3.4 depends on the test point $x$.
> This dependence is explicitly stated on the proof of the auxiliary results Proposition 3.8 and Theorem 3.10 in the Supplementary Material.
> Locally uniform bounds on the test point $x$ might follow from functional inequalities such as the ones found by [Favaro et al. '25] if extended to the NTK regime.

---

> > ### Comment · Reviewer_3AEa · 2025-08-05
> >
> > Thank you for this detailed response, the proposed changes, and clarifications.
> >
> > The authors have answered to all of the comments raised. After also reading the other reviews and rebuttals, I confirm the rating.

---

> > > ### Author Response · Authors · 2025-08-07
> > >
> > > We thank you for your helpful feedback and for engaging with our rebuttal. We are glad that our clarifications and additions addressed the concerns raised. Your input has been valuable in improving the paper.

---

### Official Review · Reviewer_yCLR · 2025-06-30

**Clarity:** 3
**Significance:** 2
**Originality:** 2
**Rating:** 4
**Confidence:** 3

**Summary:**

This paper provides quantitative upper bounds on the Wasserstein-2 distance between the output distribution of large-width shallow neural networks, trained via continuous-time gradient flow, and their associated Gaussian processes at any given test point. Building on prior works that identify the limiting Gaussian processes in the infinite-width regime, the authors derive polynomial convergence rates in network width.

**Questions:**

- In the experiments, it is unclear how the authors determine the value of $t$ corresponding to the number of gradient steps.
- Suggestion: Include more experiments aimed at quantifying the quality of the theoretical bound. For instance, in the rightmost plot of Figure 1, the decay rate appears closer to $1/\sqrt{\text{width}}$ than the expected $1/\text{width}$, suggesting the empirical rate may differ from the theoretical one. Clarifying this discrepancy would strengthen the paper.
- Suggestion: Add a discussion of the related line of work on optimization and generalization dynamics in overparameterized neural networks and clarify how the present contributions differ or connect to those results.

**Ethical Concerns:**

["NO or VERY MINOR ethics concerns only"]

**Final Justification:**

Thank you for the clarifications and the additional discussion of related work. The new paragraph and citations satisfactorily address my concerns about prior literature coverage. I have therefore revised my overall score to reflect these improvements.

**Limitations:**

yes

**Paper Formatting Concerns:**

OK

**Quality:**

2

**Strengths And Weaknesses:**

Strengths:
The paper presents novel quantitative results that extend known convergence properties of wide neural networks to the full training trajectory, with explicit polynomial bounds in Wasserstein-2 distance. The writing is clear and well-organized, making the topic accessible. In particular, the proof sketch provides valuable insight into the structure of the argument and highlights the roles of key components, such as linearization and kernel concentration.

Weaknesses:
- There is a rich, yet unmentioned, line of work studying the convergence properties and training dynamics of shallow neural networks trained with gradient flow or gradient descent, including extensions to deep architectures and various nonlinearities. Notable examples include Arora et al. (Fine-grained analysis of optimization and generalization for overparameterized two-layer neural networks) and Du et al. (Gradient descent provably optimizes overparameterized neural networks), among others. A discussion of the similarities and differences with these works would improve the positioning of this paper.
- The experiments, while appreciated, are quite limited. In particular, there is no empirical assessment of the accuracy of the derived upper bounds, or at least of the convergence rates as a function of the width or input dimension, for example. Such experiments would help evaluate how well the theoretical insights align with observed behavior.

---

> ### Author Rebuttal · Authors · 2025-07-30
>
> We thank Reviewer yCLR for their careful reading of our manuscript and for the constructive comments.
> We now address each of the weaknesses and questions raised by the reviewer:
>
> 1. In the experiments, it is unclear how the authors determine the value of t corresponding to the number of gradient steps:
>
> We thank the reviewer for pointing this out.
> To clarify, as stated in line 263 of the manuscript, in both experiments we define $t$ as the product of the learning rate and the number of iterations.
> A new sentence to emphasize this has been added to the caption of the plots.
>
> 2. About Experiments:
>
> We thank the reviewer for this observation.
> We would like to clarify that, while the reviewer is correct that the relation between $n_1$ and $\mathcal{W}_2^2$ is $\frac{1}{n_1}$, the plot in question corresponds to $\mathcal{W}_2$.
> Therefore, a slope of $-\frac{1}{2}$ in logarithmic scale is indeed expected.
>
> 3. Discussion on the works of Arora et al, Du et al:
>
> We thank the reviewer for this valuable suggestion and fully agree that the line of work initiated by Arora et al. and Du et al. is central to our topic. In response, we have added a new paragraph to the Related Work subsection of the introduction, where we now cite these seminal papers along with several subsequent contributions that build upon them.
> We hope that this addition provides an adequate overview of the relevant literature. Should we have overlooked any important reference, we would be happy to include it in a revised version.
>
> Here is the paragraph we propose:
>
> A foundational stream of research has shown that, under sufficient overparameterization, gradient-based training of neural networks converges to a global minimum.
> Seminal results by [Du et al. '19] and [Arora et al. '19] established that for wide two‑layer networks with ReLU activation, the empirical NTK remains well-conditioned, enabling convergence via kernel regression.
> Subsequent advances generalized these results to deep architectures in different directions, such as [Allen‑Zhu et al. '19,
> Zou and Gu '19,
> Sankararaman et al. '20,
> Wu et al. '19, Wei et al. '19, Zou et al. '20],
> which provide guarantees that hold with high probability over parameter initalization.
> These works reinforce that in the NTK regime, the network trajectory stays close to its linearization around initialization.
> Our contributions align with this body of work
> and further extend this literature by providing novel finite-sample quantitative bounds on the Wasserstein‑2 distance between neural network outputs and their Gaussian process approximations.

---

> > ### Comment · Reviewer_yCLR · 2025-08-01
> >
> > Thank you for the clarifications and the additional discussion of related work. The new paragraph and citations satisfactorily address my concerns about prior literature coverage. I have therefore revised my overall score to reflect these improvements.

---

> > > ### Author Response · Authors · 2025-08-07
> > >
> > > We thank you for your thoughtful feedback and for taking the time to reconsider your evaluation after our rebuttal. We are glad that our clarifications and the additional discussion of related work addressed your concerns. We greatly appreciate your updated assessment and the constructive comments that helped strengthen the paper.

---

### Official Review · Reviewer_R9Qg · 2025-07-01

**Clarity:** 3
**Significance:** 2
**Originality:** 2
**Rating:** 4
**Confidence:** 2

**Summary:**

This work investigates the rate of which a wide, two-layer neural network trained under gradient flow converge to a Gaussian process, in Wasserstein 2 distance. In particular, the rates allow to characterize the relative scaling between data dimension, network width and training horizon required for the network to stay close to the corresponding linearized kernel predictor.

**Questions:**

- In L24-26:
> "*In this limit, the network evolves approximately linearly around its initialization, and training can be understood as kernel regression with a fixed data-dependent kernel.*".

What do you mean exactly by "data dependent kernel" in this sentence? In the strict infinite width limit, the NTK kernel remains context, and corresponds to the kernel of the network at initialization.

- Although this is something ultimately personal, I find the choice of notation confusing. For instance, from the equation below L114, I understand that since the output is a scalar, $x\theta^{(0)}\in\mathbb{R}^{n_{1}}$ denote the standard (left) matrix multiplication between the matrix $\theta^{(0)}\in\mathbb{R}^{n_{0}\times n_{1}}$ and the vector $x\in\mathbb{R}^{n_{0}}$. Then, to produce a scalar (after applying the non-linearity component-wise), should I understand the product with $\theta^{(1)}\in\mathbb{R}^{n_{1}}$ as a scalar product? But then the eq. below L125 would suggest that $h_{i}\in\mathbb{R}$ and not $\mathbb{R}^{n_{1}}$ since $\theta^{(0)}_{- i}\in\mathbb{R}^{n{0}}$ is a vector? Can you clarify what I am missing or if this is a typo?

- In L182, what is the definition of the mentioned Gaussian process $G_{t}$? The one appearing after in eq. (2.5)?

- What is the main challenging of extending the result for unbounded activation $\Phi$, e.g. ReLU?

- How tight are these rates? In other words, can one turn this result around and use it as an estimate of the training timescale over which a wide network with NTK scaling start to learn features?

**Small typos**:

- L224: polinomially -> polynomially

**Ethical Concerns:**

["NO or VERY MINOR ethics concerns only"]

**Final Justification:**

This is a well-written and technically sound paper with new results concerning the convergence rate of wide neural networks to Gaussian processes. As stated in my review, I believe the main weakness is the significance of the scope of the results to the NeurIPS ML theory community.

**Limitations:**

The authors discuss the limitations of their result in Section 5.

**Paper Formatting Concerns:**

No paper formatting concerns.

**Quality:**

3

**Strengths And Weaknesses:**

**Strengths**: The paper is clearly writen: the motivation and goals are clearly stated, and the result is compared with the relevant related literature. The better control over the rates allowing to consider longer timescales on gradient flow is interesting.


**Weaknesses**: The main weakness of this works concerns the significance. Although there is still some interest on the NTK regime in the NeurIPS ML theory community, I believe it is now consensual its limited relevance to the theory of neural networks. Nevertheless, the refined NTK rates could be of interest to a subset of the community.

---

> ### Author Rebuttal · Authors · 2025-07-30
>
> We thank Reviewer R9Qg for their careful reading of our manuscript and for the constructive comments.
> We now address each of the weaknesses and questions raised by the reviewer:
>
> 1. Clarification on "In this limit, the network evolves approximately linearly around its initialization, and training can be understood as kernel regression with a fixed data-dependent kernel.":
>
> We thank the reviewer for raising this point. With the original phrase we intended to emphasize that the infinite-width kernel is a function of $(x,x')$.
> We agree that, in the infinite-width limit, the kernel is independent of the network parameters $\theta^{(0)}, \theta^{(1)}$.
> To avoid ambiguity, we have revised the sentence to:
> "In this limit, the network evolves approximately linearly around its initialization, and training can be understood as kernel regression with a fixed kernel which depends on the the architecture only, and is a function of $(x,x')".
>
> 2. Issue with scalar products:
>
> We agree with the reviewer. The i-th preactivation $h_i(x;\theta)$ belongs to $\mathbb R$.
> We corrected this typo and deleted the subsequent definition of $h_{ij}$.
>
> 3. In L182, what is the definition of the mentioned Gaussian process ? The one appearing after in eq. (2.5):
>
> We thank the reviewer for noticing this incoherence in the order of exposition.
> We moved equation (2.5) before that paragraph in order to fix this.
>
> 3. What is the main challenging of extending the result for unbounded activation, e.g. ReLU?
>
> For unbounded activation, the inequalities in Lemma B.12, which are paramount to showing our main result, become empty.
> Although our proof does not hold for unbounded activations for this reason, we believe the result is still true in this setting and that the proof can be improved.
> This is now acknowledged in a new paragraph in the Discussion section:
>
> We conjecture that our main result remains valid even without Assumption 3, as suggested by our numerical experiments with the ReLU activation.
> In this work, we deliberately focused on a specialized setting with mild hypotheses to obtain a novel and technically precise result while maintaining a clear exposition.
> Future research will aim to relax the regularity assumptions on the activation and extend our analysis to a more general setting.
>
> 4. How tight are these rates? In other words, can one turn this result around and use it as an estimate of the training timescale over which a wide network with NTK scaling start to learn features?
>
> As far as we know, our bounds are not tight.
> We rephrased the first item in the discussion session, which discusses limitations of our main result, in order to include a brief discussion on this matter and potential connections to feature learning:
>
> Our main result is not uniform in time.
> Although the dependence on time can be minimized at the price of including a sufficiently big multiplicative constant in the right-hand side of our inequality as discussed in Remark 3.5,
> a general result holding uniformly in $t>0$, in the limit when $t$ tends to infinity exponentially on $n_1$ is not available.
>
> This dependence on time could be related to the transition from the NTK regime to a feature-learning regime, as suggested by the work of Huang and Yau (2020).
> Their analysis, however, does not address the tails of the distributions, which in our proof correspond to the set $S^C$ and are responsible for the $t^6$ scaling.
> Moreover, Yang and Hu (2021) show that under standard and NTK parameterizations, wide networks cannot perform feature learning in the infinite-width limit.
>
> This suggests that our observed $t^6$ scaling might reflect the boundary of the NTK regime:
> in the ``bad event'' $S^C$ or for sufficiently large times, the training dynamics may drift into feature-learning, where purely kernel-based control breaks down.
> Our main result remains consistent with works such as [Bartlett, Montanari and Rakhlin '21, Chizat, Oyallon and Bach '19], which hold with high probability, whereas our analysis explicitly incorporates the contribution of the event $S^C$.
> At present, it is unclear whether the $t^6$ scaling is sharp.
> We would like to address this problem in future work.

---

> > ### Comment · Reviewer_R9Qg · 2025-08-01
> >
> > I thank the authors for welcoming my suggestions and for their rebuttal. My questions have been answered, and I am retaining my score.

---

> > > ### Author Response · Authors · 2025-08-07
> > >
> > > We thank you for your helpful feedback and for engaging with our rebuttal. We are glad that our clarifications and additions addressed the concerns raised. Your input has been valuable in improving the paper.

---

### Official Review · Reviewer_9rHw · 2025-07-02

**Clarity:** 2
**Significance:** 4
**Originality:** 3
**Rating:** 5
**Confidence:** 3

**Summary:**

The authors addresses a gap in the current literature on infinite width neural networks trained by gradient descent by providing a quantitative characterization of the distance between a finite-width neural network approximation and its infinite counterpart. The authors specifically focus on shallow (single-hidden-layer) networks and demonstrate that the Wasserstein distance decays polynomially with increasing network width (Theorem 3.4).

This is validated experimentally in Section 4, where the authors empirically compute the Wasserstein distance between a collection of finite-width networks and the samples from the infinite-width network.

**Questions:**

On Theorem 3.4, the bound's polynomial dependence on $t^6$ is noted as becoming slack for fixed $r$ and $n\_1$ as $t$ grows unboundedly, even though the authors state that a sufficiently large $r$ can make the time-dependent term negligible for $t$ growing polynomially with $n\_1$. Could the authors provide a more detailed intuition behind this $t^6$ term? After all, one would expect training to converge as time goes on.

Additionally, the authors briefly mention the work by de G. Matthews et al. (2018), in that paper, the authors prove the weak convergence of arbitrary width BNNs at initialization under the metric:

$$
\rho(f,f') = \sum\_{i=1}^\infty \min(1, |f(x\_i)-f'(x\_i)|),
$$

where $\\{x\_i\\}\subset\mathcal{X}$ is a countably infinite input set. The same paper also empirically compare the distance of these distributions under the  maximum mean discrepancy (MMD) under a RBF kernel. As the bound of the authors holds for $t=0$, I believe the authors should briefly discuss the relationship, if any, between these two metrics ($\rho$ and MMD) with the Wasserstein metric. For example, does convergence in one implies convergence in another? Do the bounds relate? Why pick one over the other?

**Ethical Concerns:**

["NO or VERY MINOR ethics concerns only"]

**Final Justification:**

The authors have addressed all of my issues in a thorough manner, I believe these additions and corrections will increase the connections of this paper with the existing literature and also point to readers where improvements in current results can happen.

**Limitations:**

Yes

**Quality:**

3

**Strengths And Weaknesses:**

I believe this paper offers an interesting initial step into the evolution of the convergence of infinite-width neural networks. It is also commendable that the authors address the limitations and future directions of their work in clear terms.

In terms of clarity, currently, assumptions are stated across two different sections (”Notation” and “Assumptions and Main Result”), I believe keeping all assumptions in one section would improve the flow of the paper. Additionally, the authors use, at the same time, notations where the dependency of $f$ on the parameters is implicit (i.e., $\dot{f}\_t(x)$) and where it's explicit (i.e., $\nabla\_{\theta}\mathcal{R\_D}[f,\theta]$). Given that there is still space available, I would suggest the authors keep the notation on the explicit side and write $\frac{\partial}{\partial t}f(x;\theta\_t)$ to streamline reading by others.

---

> ### Author Rebuttal · Authors · 2025-07-30
>
> We thank Reviewer 9rHw for their careful reading of our manuscript and for the constructive comments.
> We now address each of the weaknesses and questions raised by the reviewer:
>
> 1. Intuition on term $t^6$ in the main theorem:
>
> We added the following remark to the main theorem to try to give an intuition of this term:
>
> The term $t^6$ in the right hand side of the inequality is due to Lemma B.12 and Theorem B.10 in the Supplementary Material. Lemma B.12 provides upper bounds of the entries $\theta^{(0)}_t$ and $\theta^{(1)}_t$ that account for perturbations that occur on tail events with respect to the initalization distribution (i.e. in the ``bad event" $S^C$).
> A finer control is possible if one is interested in a result that holds on $S$ only, which has high probability, such as the ones in [Bartlett, Montanari and Rakhlin '21, Chizat, Oyallon and Bach '19].
> This finer control corresponds to Theorem B.9.
>
> 2. Relation to $\rho$ and MMD metrics from [Matthews et al. '18]:
>
> We thank the reviewer for raising this important point. Our result connects with the findings of [Matthews et al. '18], which prove the convergence of fully connected neural networks to a Gaussian process under the following metric:
> $$
> \rho(f,f') = \sum_{i\in \mathbb N}\frac{1}{2^i} \min (1,\vert f(x_i)-f'(x_i)\vert),
> $$
> where the $x_i$ form a countable set of inputs.
> Our setting, as opposed to the setting of [Matthews et al. '18] considers  a finite set of test and training points.
> However, it is natural to consider in our setting the version of $\rho$ restricted to finite input, which we shall denote $\rho_F$.
> Then the expected value of $\rho_F$ can be related to $\mathcal W_2$:
> $$
> \rho_F(f,f') \le \sum_{i=1}^n \frac{1}{2^i} \vert f(x_i)-f'(x_i)\vert < \| f-f'\|_2
> $$
> By taking expected value with respect to a coupling $\pi$ and then taking the infimum on the couplings $\pi$ between $f\sim \mu$ and $f' \sim \nu$ we obtain:
> $$
> \mathbb E[\rho_F(f,f')] < \mathcal W_2(\mu,\nu).
> $$
>
> Furthermore, [Matthews et al. '18] also studies convergence under the maximum mean discrepancy (MMD) with respect to a RKHS $(H,k_H)$.
> Even though the connection between MMD and regularized versions of the Wasserstein distance, such as Sinkhorn divergences ([Feydy et al. '19]) or Gaussian-smoothed OT ([Nietert et al. '21]), it is known that in general the MMD cannot be bounded bounded from above by the $p$-Wasserstein distance.
> The authors of [Vayer,Gribonval '23] found sufficient and necessary conditions on $k_H$ under which, up to a multiplicative constant:
> $$MMD(f,f') \le \mathcal W_2(f,f').$$
>
> Hence convergence in $2$-Wasserstein distance implies convergence in the $\rho_F$ metric, and convergence with respect to MMD under the hypothesis of Proposition 2 in [Vayer,Gribonval '23].
> Note that, while the metric $\rho_F$ offers a notion of pointwise convergence, it lacks the geometric interpretability and sensitivity to scaling that Wasserstein-2 captures.
> On the other hand, also note that the setting of [Matthews et al. '18] is more general since it covers deep networks, while we restrict our study to the shallow case.
>
>
> We decided to add to the Related Work subsection of the introduction the following paragraph summarizing the above discussion:
>
>
> Our results are closely related to the work of [Matthews et al. '18], who proved weak convergence of fully-connected BNNs at initialization to a Gaussian process under the metric
> $\rho(f,f') = \sum_{i \in \mathbb{N}} 2^{-i} \min(1, |f(x_i)-f'(x_i)|)$, defined on a countable input set.
> In our setting, the input set is finite; considering the restriction $\rho_F$, it follows that convergence in $\mathcal{W}_2$ implies convergence in $\rho_F$.
> [Matthews et al. '18] also analyzed convergence under the maximum mean discrepancy (MMD).
> While MMD is not generally controlled by Wasserstein distances, connections have been established via regularized OT divergences [Feydy et al. '19],[Nietert et al. '21].
> Moreover, [Vayer,Gribonval '23] identified conditions on the RKHS kernel $k_H$ under which $MMD \lesssim \mathcal{W}_2$.
> Consequently, our bounds also imply MMD convergence under these conditions.
> The metric $\rho_F$ which offers a notion of pointwise convergence and, by taking the minimum, is oblivious of the tails of the distributions, which helps stablish the results in [Matthews et al. '18].
> On the other hand, $\mathcal{W}_2$ captures the geometric structure and scaling of the output space.
> Finally, while [Matthews et al. '18] address the more general setting deep networks, our analysis focuses on the shallow case, yielding new quantitative rates which improve previous ones in our setting.
>
>
> 3. Weakness: keep all the assumptions (1-4) in the same place?
>
>
> We agree that the exposition benefits from having all the assumptions collected in one place. Accordingly, we put all the assumptions on a list before Theorem 3.4, which we believe improves the readability of the manuscript.
>
>
> 4. Weakness: keep $\theta$ explicitly in notation.
>
>
> We agree with the reviewer that it's best to stick with one notation only. This has been now changed to show at all times the dependence on $\theta$ and $\bar \theta$ explicitly.

---

> > ### Comment · Reviewer_9rHw · 2025-08-06
> >
> > Thank you for the reply. I believe the discussion about the choice of metric for convergence is a great addition and provides helpful pointers for readers interested in these topics. Specifically, for these two papers, it is quite relevant that Vayer and Gribonval (2023) show that $W\_2$ bounds MMDs based on RBFs.
> >
> > Regarding the $t^6$ term, the reason for its presence is clearer now, as it arises from the $\\mathcal{W}\_2(f\_t, f\_t^{\\text{lin}})$ term via the triangle inequality. However, my main concern is what happens when $n\_1$ is held constant. A naive reading of the theorem suggests that either $f\_t$ is diverging as $t \\to \\infty$ or the bound is becoming too loose to be useful. While I understand that we can gain more control by increasing $n\_1$ or imposing stronger assumptions, have you observed in practice how $\\mathcal{W}\_2(f\_t, f\_t^{\\text{lin}})$, or an estimate of it, compares to the bound containing $t^6$?

---

> > > ### Author Response · Authors · 2025-08-07
> > >
> > > Thank you again for your thoughtful follow-up.
> > > You are right that the $t^6$ dependence arises from approximating a tail event $S^C$ when bounding  $\mathcal{W}_2(f_t, f_t^{\text{lin}})$.
> > > This term reflects a worst-case contribution from rare trajectories and is not observed in results that hold with high probability or use truncated metrics, where the constants become uniform in time (see for example [Bartlett, Montanari, Rakhlin '21], [Chizat, Oyallon, Bach '19] or [Arora et al. '19]).
> > >
> > > We agree that it is unclear whether this $t^6$ scaling is sharp.
> > > Estimating $\mathcal{W}_2(f_t, f_t^{\text{lin}})$ empirically is difficult in this regime, precisely because the growth is driven by rare initializations (corresponding to tails of Gaussians) that are hard to capture numerically.
> > > Even at fixed width low-width settings, $S^C$ has low probability, and at present, we do not have a matching lower bound.
> > >
> > > We thank the reviewer for raising this subtle point and believe it highlights an important distinction between our full-distributional bound and previous high-probability results.

---

> > > > ### Comment · Reviewer_9rHw · 2025-08-08
> > > >
> > > > Thank you; this discussion has been very helpful and clarifying. I believe these final comments would be well-suited for the paper to give readers greater context and show areas where further improvements could be made. I have no further concerns and look forward to seeing the paper with all the improvements in place.

---

### Decision · Program_Chairs · 2025-09-17

**Decision:**

Accept (poster)

**Comment:**

This paper focuses on the rate of which a wide and two-layer neural network with gradient descent converge to a Gaussian process in Wasserstein 2 distance. After the discussion phase, almost all responses seem to be fixed well. The novelty and improvements of this work have received praise from reviewers. However, there still are some doubts on the formal expression of the main theorems and the readability of this paper.